# Persistent T cell unresponsiveness associated with chronic visceral leishmaniasis in HIV-coinfected patients
Nicky de Vrij [1,2], Julia Pollmann [3], Antonio M. Rezende [4], Ana V. Ibarra-Meneses[5], Thao-Thy Pham[1], Wasihun Hailemichael [6], Mekibib Kassa[7], Tadfe Bogale[7], Roma Melkamu[7], Arega Yeshanew[7], Rezika Mohammed[7], Ermias Diro [7], Ilse Maes[8], Malgorzata A. Domagalska [8], Hanne Landuyt[9], Florian Vogt[10,11,12], Saskia van Henten [12], Kris Laukens [2], Bart Cuypers [2], Pieter Meysman [2], Hailemariam Beyene[13], Kasaye Sisay[13], Aderajew Kibret[13], Dagnew Mersha[13], Koert Ritmeijer[14], Johan van Griensven[12] & Wim Adriaensen [1] ✉

A large proportion of HIV-coinfected visceral leishmaniasis (VL-HIV) patients exhibit chronic disease with frequent VL recurrence. However, knowledge on immunological determinants underlying the disease course is scarce. We longitudinally profiled the circulatory cellular immunity of an Ethiopian HIV cohort that included VL developers. We show that chronic VL-HIV patients exhibit high and persistent levels of TIGIT and PD-1 on CD8+/CD8- T cells, in addition to a lower frequency of IFN-γ+ TIGIT- CD8+/CD8- T cells, suggestive of impaired T cell functionality. At single T cell transcriptome and clonal resolution, the patients show CD4+ T cell anergy, characterised by a lack of T cell activation and lymphoproliferative response. These findings suggest that PD-1 and TIGIT play a pivotal role in VL-HIV chronicity, and may be further explored for patient risk stratification. Our findings provide a strong rationale for adjunctive immunotherapy for the treatment of chronic VL-HIV patients to break the recurrent disease cycle.

Visceral leishmaniasis (VL) or kala-azar, caused by species of the *Leishmania donovani* complex, is one of the most severe parasitic diseases. It is a vector-borne disease, as *Leishmania* parasites are transmitted by the bite of an infected female phlebotomine sand fly. In the human host, *Leishmania* predominantly replicates within the phagolysosomes of myeloid host cells residing in the liver, spleen, and bone marrow. Leishmaniasis is classified by the World Health Organization (WHO) as one of the twenty neglected tropical diseases, disproportionally affecting poor populations living in low-to-middle-income countries with a tropical climate[1]. The global burden is substantial, accounting for an estimated 50,000 to 90,000 new cases annually[2,3]. Ethiopia, together with Brazil, China, Eritrea, India, Kenya, Somalia, South Sudan, Sudan and Yemen account for more than 90% of all VL cases reported to the WHO in 2020[3].

Although VL is fatal if left untreated, around 90–95% of immunocompetent patients exhibit a good clinical response if treated[4-6]. However, co-infection with the Human Immunodeficiency Virus (HIV) presents a substantial challenge in patient management, particularly in North-West Ethiopia where up to 30% of individuals with VL are co-infected with HIV[7,8].

The risks for initial treatment failure and VL relapse 3–12 months after treatment are substantially increased, ranging up to 20% and 70% in VL-HIV patients, respectively[9-12]. Even after apparent initial parasitological cure with VL treatment, many patients with VL-HIV co-infection will still develop a chronic form of VL characterised by frequent recurrent VL episodes over a period of several years[13]. These patients demonstrate a persistent lack of CD4+ T cell reconstitution despite successful HIV suppression with anti-retroviral therapy (ART)[10]. The recurrent VL episodes in these patients with chronic VL-HIV have been shown to be caused by the recrudescence of the initial population of infecting parasites, and not by (re) infections[14]. Moreover, a progressive loss of the host immune response needed to control the parasite is believed to drive this parasite recrudescence, rather than the development of treatment resistance by the parasite[14,15].

Therefore, the persistent dysregulation of host immunity in VL-HIV disease is thought to be caused by synergistic immunopathological mechanisms of the two infections[16-19]. For example, both *Leishmania* and HIV have been observed to reduce the antigen presentation capacity of antigen presenting cells[20-22]. In addition, HIV causes marked depletion of

CD4[+] T cells and promotes a CD4[+] T helper 2 (Th2) polarised immune response[23]. In experimental models, a Th2 polarised immune response has been shown to be permissive for parasite replication, while protective immunity to VL has been linked to a CD4[+] T helper 1 (Th1) polarised immune response with high interferon gamma (IFN-γ) production, although this Th1/Th2 dichotomy has been challenged in human VL[23–25]. CD8[+] T cells typically play a key role in viral clearance, and may adopt a similar protective role during VL in a CD4[+] T cell-deprived environment[26]. Yet, HIV and VL have both been observed to cause CD8[+] T cell exhaustion and senescence, with VL development as an independent cause of T cell senescence in HIV-co-infected individuals[19,27,28]. While the causal impact of T cell exhaustion or senescence on VL-HIV disease chronicity remains unknown, it merits further attention as VL-HIV seems to be a primarily immune-driven disease.

It remains elusive which immunological mechanisms underlie the chronic VL disease course that develops in a major subset of the VL-HIV patients. Consequently, treating physicians are currently lacking strong prognostic markers to predict VL chronicity in HIV-co-infected individuals. They are dependent on the few associated determinants such as a persistent lack of CD4[+] T cell reconstitution to guide patient management, that exhibit variable and limited prognostic value[9,10,29,30]. Therefore, a better characterisation of the host immune response in VL-HIV disease is vital for the identification of novel determinants that predispose patients to the chronic VL form.

Thus, in this study, we longitudinally characterised the compositional and functional changes in peripheral immune cell subsets, as blood represents an easily accessible compartment for clinical care, in chronic (frequent recurrent episodes) and non-chronic (no frequent recurrent episodes) VL-HIV patients over a median of 19.5 months and compared it to long-term cured or asymptomatic *Leishmania*-infected individuals and an HIV control group. Here, we found that chronic VL-HIV patients exhibit persistently high levels of the exhaustion-associated markers PD-1 and TIGIT on their CD8[+] and CD8[−] T cells throughout the disease course. In addition, these chronic VL-HIV patients show CD4[+] T cell anergy, characterised by a lack of T cell activation at the single-cell transcriptome level, and a lack of a lymphoproliferative response at the clonal level.

## Results

### VL-HIV chronicity is not associated with ART status nor CD4 + T cell counts, but with KAtex positivity

Our study included a total of 63 participants, including 19 (30.2%) individuals with HIV, 20 (31.7%) *Leishmania*-seropositive individuals with no incident VL during our study (LS+-HIV), and 24 (38.1%) participants with HIV that developed active VL during the course of the study (VL-HIV) (described in Supplementary Table 2).

As expected in the Ethiopian context, all LS+-HIV and VL-HIV participants were male, and almost all of them (88.6%) were working as farmers or daily labourers, occupations shown to have high risk of *Leishmania* transmission (Supplementary Table 2)[31]. Although matching in most variables, the majority of our endemic HIV control group were female (57.9%) and did not work in high-risk occupations (52.6%) (Supplementary Table 2). All our participants (98.4%) except one VL-HIV patient were on ART at study inclusion with a self-reported good adherence (only one participant reported a recently missed dose). VL-HIV patients exhibited lower CD4[+] T cell counts (P < 0.001), total lymphocytes (P < 0.001), platelets (P < 0.001), and haemoglobin (P < 0.001) levels at time of overt disease than the other participant groups did at study inclusion. None of the LS+-HIV participants developed overt VL during the study duration and consisted in majority (65%) of individuals with a history of VL, confined to 0-2 prior episodes with an average of 6 years (73 months) since their previous episode, suggesting long-term VL cure (Supplementary Table 2).

Out of the 24 participants with VL-HIV (Table 1), 7 (29.2%) patients were considered non-chronic VL-HIV cases which had no prior nor recent (within <10 years of study inclusion) VL history, and 17 (70.8%) patients were chronic cases with recent VL history (ranging from 1-12 episodes

before study admission) of which most developed one or multiple additional VL episode(s) during the study period (Table 1, Fig. 1b). The VL-HIV patients were followed-up for a median time of 19.5 months (IQR 12-26.5), visualised per participant in Fig. 1b. All non-chronic VL-HIV patients were permanently living in *Leishmania* endemic areas, while 4 (23.5%) chronic VL-HIV patients were migrant workers. Non-chronic VL-HIV patients were significantly older than chronic VL-HIV patients (P = 0.022). While all chronic VL-HIV patients were KAtex-positive at active disease, all but two patients with non-chronic VL-HIV were negative for KAtex (P < 0.001). Besides age and KAtex-positivity, no significant differences in participant characteristics could be identified between participants with non-chronic or chronic VL-HIV disease (Table 1).

Finally, no significant differences or bias could be observed in participant characteristics between the 8 selected participants for single-cell sequencing, indicating a representative selection (Supplementary Table 1).

### The peripheral blood immune cell composition is not altered during VL-HIV chronicity

To study whether differences could be detected in the composition of blood immune cell subsets of chronic VL-HIV patients, we first employed multi-colour flow cytometry to investigate the CD8[+] and CD8[−] T cell (as a proxy for CD4[+] T cells) populations.

No difference was observed in the frequency of CD8[+] T cells between participant groups (Fig. 2a), however, a lower relative CD8[−] T cell frequency was observed in VL-HIV patients as compared to the HIV group (Fig. 2d). Further suggesting the non-prognostic value of CD4[+] counts for VL chronicity, no difference was observed in the CD8[+] nor CD8[−] T cell (as proxy for CD4[+] T cells) frequency between non-chronic or chronic VL-HIV patients at active disease nor at the end of treatment (Fig. 2b,e). However, a significant difference between the non-chronic and chronic VL-HIV patients was observed in the frequency of CD8[−], but not the CD8[+], T cells across the follow-up duration (Fig. 2c, f).

For a broader characterisation on differences in circulatory immune cell composition, we sequenced a total of 17.308 PBMCs across the different participant groups (n = 10, see methods), and a total of 12.822 cells to evaluate VL treatment impact in cVL-HIV and ncVL-HIV patients (n = 4, both D0 and EOT timepoints, see Fig. 1, Supplementary Table 1 for participant selection). Cell type clustering and annotation resulted in 20 (all group comparison) and 21 (longitudinal comparison) distinct clusters (Fig. 3). Although no statistical comparison could be performed due to the small sample size, the cellular composition was relatively similar across disease groups, except for CD4[+] T effector-memory (TEM) cells which were predominantly identified in the *Leishmania* co-infected participants only, suggesting an active disease response in these participants (Fig. 3). In addition, there was a visual trend for an increase in the proportion of CD4[+] T cells after VL treatment in the ncVL-HIV, but not the cVL-HIV group, suggesting a lack of CD4[+] reconstitution in chronic VL-HIV patients. Finally, depletion of CD14[+] monocytes was observed at both timepoints in cVL-HIV patients, as compared to the ncVL-HIV patients.

### Persistently high levels of TIGIT[+] CD8[+] and CD8[−] T cells, and PD-1[+] CD8[+] T cells in chronic VL-HIV patients

Since T cell exhaustion and senescence are clear hallmarks of a chronic HIV or VL infection that may lead to decreased functionality against *Leishmania*, we assessed the frequencies of CD8[+] T cells and CD8[−] T cells (as a proxy for CD4[+] T cells) that were positive for a range of exhaustion-associated (PD-1, LAG-3, TIM-3, and TIGIT) and senescence (KLRG1 and CD57) markers, and the marker's mean fluorescence intensities (MFI), across the different participant groups.

For all exhaustion-associated markers, we observed significant increases in the frequencies (except for TIGIT) and MFI (except for LAG-3) of marker-positive CD3[+]CD8[+] and CD3[+]CD8[−] T cells in participants with VL-HIV as compared to the other participant groups (Fig. 4a, d, g, j, Supplementary Fig. 3a,d,g,j, Supplementary Fig. 4a,d,g,j, and Supplementary Fig. 5a,d,g,j). For LAG-3, we only observed an increase in marker

**Table 1 | Participant socio-demographic, clinical, and bio-chemical characteristics of patients with non-chronic VL-HIV or chronic VL-HIV at various timepoints, including at active disease development (D0), at End-of-Treatment (EOT) and at six months post-treatment (Post M6; VL-HIV only)**

| | Non-chronic VL-HIV (n = 7) | Chronic VL-HIV (n = 17) | P value |
|---|---|---|---|
| Socio-demographic characteristics | | | |
| Age in years, median (IQR) | 45 (41–51) | 38 (31–41) | 0.022 |
| Male, n (%) | 7 (100) | 17 (100) | 1 |
| BMI in kg/m², median (IQR) | 15.8 (15.1–16.4) | 17 (15.8–18.6) | 0.192 |
| Literacy, n (%) | 4 (57.1) | 13 (76.5) | 0.374 |
| Permanently living in endemic area, n (%) | 7 (100) | 13 (76.5) | 0.283 |
| Occupation, n (%) | | | 0.449 |
| Daily labourer | 4 (57.1) | 11 (64.8) | |
| Farmer | 3 (42.9) | 3 (17.6) | |
| Other | 0 | 3 (17.6) | |
| Clinical characteristics and disease history | | | |
| VL history, n (%) | 2 (28.6) | 17 (100) | NA |
| Past VL episodes, median (IQR) | 0 (0–0.5) | 2 (2–8) | NA |
| Months since previous VL episode, median (IQR) | 185 (164–206) | 5 (4–10) | NA |
| Microscopically confirmed, n (%) | 5 (71.4) | 17 (100) | 0.076 |
| Parasite grading out of those microscopically confirmed, n (%) | | | 0.311 |
| +1 to +3 | 4 (80) | 7 (41.2) | |
| +4 to +6 | 1 (20) | 10 (58.8) | |
| VL treatment regime, n (%) | | | 1 |
| AmBisome + Miltefosine | 7 (100) | 16 (94.1) | |
| AmBisome + Sodium Stibogluconate | 0 (0) | 1 (5.9) | |
| On ART, n (%) | 6 (85.7) | 17 (100) | 0.292 |
| ART treatment regime, n (%) | | | 0.849 |
| 3TC + TDF + EFV | 6 (85.7) | 10 (58.8) | |
| 3TC + TDF + DTG | 0 (0) | 2 (11.8) | |
| 3TC + TDF + D4T | 0 (0) | 1 (5.9) | |
| 3TC + TDF + ATV/r | 0 (0) | 1 (5.9) | |
| 3TC + AZT + ATV/r | 1 (14.3) | 1 (5.9) | |
| 3TC + AZT + NVP | 0 (0) | 2 (11.8) | |
| Concomitant diseases, n (%) | 2 (28.6) | 4 (23.5) | 1 |
| Typhoid fever | 0 (0) | 3 (17.6) | |
| Amoebiasis | 0 (0) | 1 (5.9) | |
| Pneumonia | 1 (14.3) | 0 (0) | |
| Pulmonary tuberculosis | 1 (14.3) | 0 (0) | |

**Table 1 (continued) | Participant socio-demographic, clinical, and biochemical characteristics of patients with non-chronic VL-HIV or chronic VL-HIV at various timepoints, including at active disease development (D0), at End-of-Treatment (EOT) and at six months post-treatment (Post M6; VL-HIV only)**

| | Non-chronic VL-HIV (n = 7) | Chronic VL-HIV (n = 17) | P value |
|---|---|---|---|
| Laboratory markers at D0 | | | |
| rK39 RDT positivity, n (%) | 7 (100) | 16 (94.1) | 1 |
| rK39 ELISA positivity, n (%) | 6 (85.7) | 15 (88.2) | 1 |
| KAtex positivity, n (%) | 2 (28.6) | 17 (100) | <0.001 |
| DAT positivity, n (%) | 5 (71.4) | 17 (100) | 0.076 |
| *Leishmania* PCR positivity, n (%) | 7 (100) | 17 (100) | 1 |
| CD4 count, median (IQR); (cells/µl) | 66 (52–157) | 54 (36–113) | 0.547 |
| Lymphocytes, median (IQR); (×10³/µl) | 0.46 (0.26–1) | 0.56 (0.33–0.74) | 1 |
| Platelets, median (IQR); (×10³/µl) | 124 (92–130) | 112 (77–149) | 0.973 |
| Haemoglobin, median (IQR); (g/dL) | 9.5 (8.8–9.9) | 9.4 (7.7–10.6) | 1 |
| Laboratory markers at EOT | | | |
| Lymphocytes, median (IQR); (×10³/µl) | 0.81 (0.44–1.24) | 0.9 (0.4–1.27) | 0.891 |
| Platelets, median (IQR); (×10³/µl) | 216 (167–262) | 145 (108–169) | 0.077 |
| Haemoglobin, median (IQR); (g/dL) | 9.5 (8.8–10.6) | 10.1 (8.5–11.2) | 0.616 |
| Laboratory markers at Post M6 | | | |
| CD4 count, median (IQR); (cells/µl) | 219 (100–438) | 82 (61–104) | 0.091 |
| Lymphocytes, median (IQR); (×10³/µl) | 0.89 (0.51–1.22) | 0.5 (0.34–0.82) | 0.462 |
| Platelets, median (IQR); (×10³/µl) | 184 (94–188) | 116 (96–129) | 0.291 |
| Haemoglobin, median (IQR); (g/dL) | 11.1 (9.8–11.4) | 10.3 (9.8–11.9) | 0.892 |

MFI between the VL-HIV and HIV groups, but not between the VL-HIV and LS+-HIV groups, on both of the T cell populations (Supplementary Fig. 4d and Supplementary Fig. 5d). This consistent pattern of exhaustion-marker positivity in the VL-HIV group in contrast to the other groups could also be detected when comparing the frequency of $CD8^+$ and $CD8^-$ T cells that are double-positive for any combination of the exhaustion-associated markers PD-1, TIM-3, and LAG-3, as the VL-HIV group had a significantly higher frequency for all combinations (Supplementary Fig. 6a,d,g,j and S7a,d), suggesting a more exhausted compartment than an activated one.

At both the active disease and end-of-treatment timepoints, we observed a higher frequency and MFI of PD-1 on $CD8^+$ T cells of chronic VL-HIV patients than in non-chronic VL-HIV patients (Fig. 4b,e). For the $CD8^-$ T cell population, we only observed a higher frequency of PD-1 in chronic VL-HIV patients at the end-of-treatment timepoint, but not at active disease nor by MFI (Supplementary Fig. 3b, e). We observed both a higher frequency and a higher MFI of TIGIT on both the $CD8^+$ and $CD8^-$ T cell populations of chronic VL-HIV patients than in non-chronic VL-HIV patients (Fig. 4h, k and Supplementary Fig. 3h,k). Furthermore, these increases in the frequency/MFI of TIGIT on both T cell populations persisted across the entire study duration (Fig. 4i, l and Supplementary Fig. 3i,l). The frequency and MFI of PD-1 on $CD8^+$ T cells also remained persistently higher in the chronic VL-HIV group than in the non-chronic VL-HIV group across the study follow-up, but for the $CD8^-$ T cells this was only the case for the frequency and not the MFI (Fig. 4c, f and Supplementary Fig. 3c,f). In contrast to TIGIT and PD-1, we did not observe any difference in the frequency nor the MFIs of LAG-3 nor TIM-3 between the cVL-HIV and ncVL-HIV patients at any timepoint (Supplementary Fig. 4b, c, e, f, h, i, k, l and Supplementary Fig. 5b, c, e, f, h, i, k, l). Furthermore, when evaluating the frequency of T cells double positive for combinations of PD-1, TIM-3, and LAG-3, we observed no difference between the non-chronic and chronic VL-HIV patients at any timepoint (Supplementary Fig. 6b, c, e, f, h, i, k, l and Supplementary Fig. 7b, c, e, f).

For the senescence-associated markers, we observed an increased MFI of KLRG1 on $CD8^+$ T cells in VL-HIV patients in comparison to

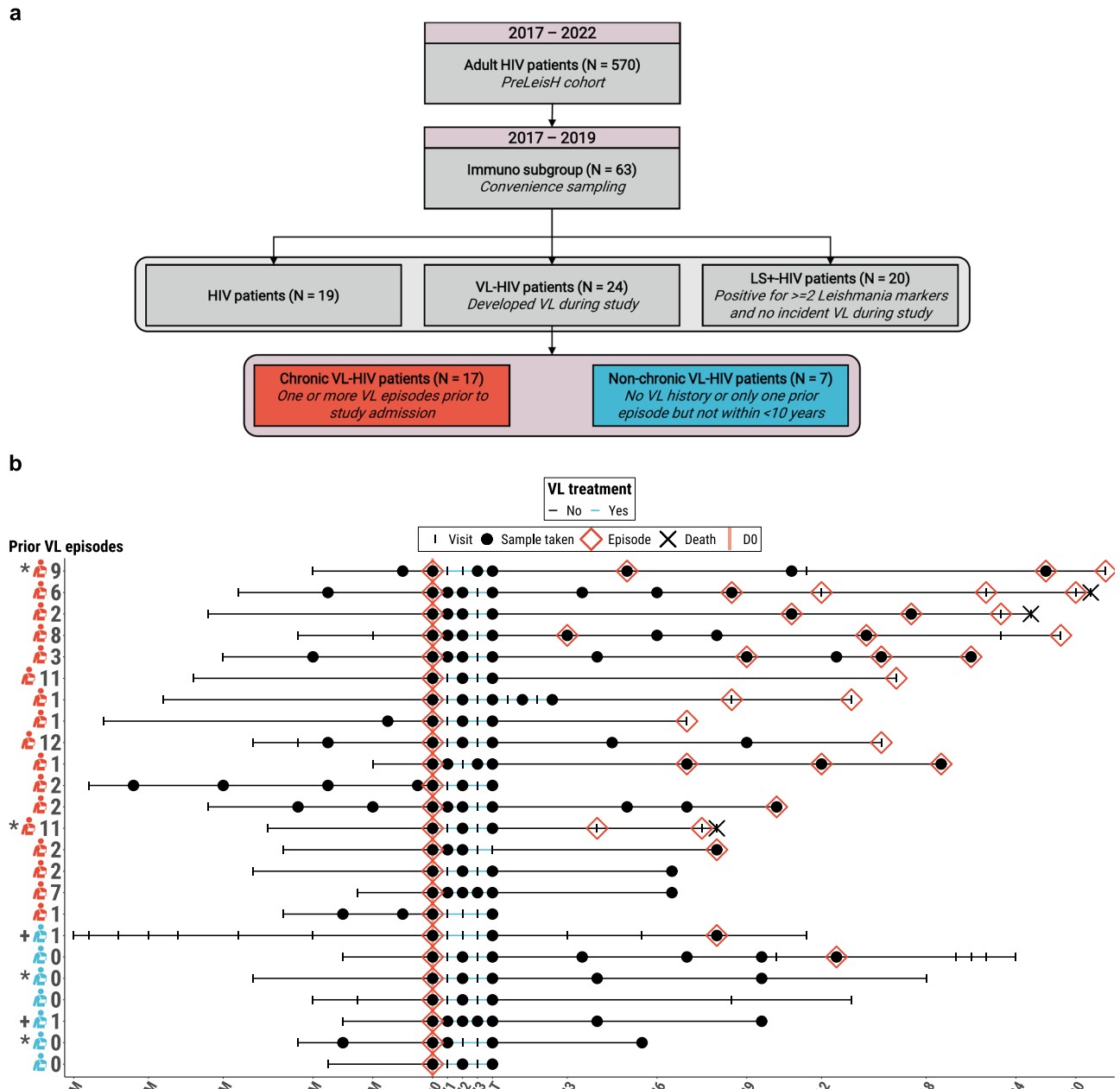

**Fig. 1 | Study design and patient stratification. a** Flow chart and selection of study participants. HIV: human immunodeficiency virus, VL-HIV: visceral leishmaniasis-HIV patients, LS+-HIV: *Leishmania*-seropositive HIV co-infected individuals. **b** VL-HIV patients follow-up monitoring timelines. The depicted months of follow-up were calculated backwards and onwards from the time of active VL development (D0) to standardise timepoints between individuals. On the left axis, the number of VL episodes prior to study inclusion are shown beside the red (Chronic VL-HIV) and blue (Non-chronic VL-HIV) patient icons. Stripe indices indicate visits while black dots indicate that a blood sample was taken at that visit. VL disease episodes are represented by red diamond symbols. Deaths are indicated by a black cross. Blue lines indicate the treatment periods. Individuals indicated with a plus sign (+) are non-chronic VL-HIV patients with one prior VL episode more than 10 years before study inclusion. Individuals indicated with an asterisk (*) were included in the single-cell immune profiling. D0 Day 0 (active disease development); W Week, EOT End of Treatment, M Month.

the other participant groups, and an increased MFI of KLRG1 on CD8⁻ T cells in the VL-HIV patients in comparison to the HIV group but not the LS+-HIV group (Supplementary Fig. 8a, d). Except for a slight increase in the MFI of KLRG1 on CD8⁻ T cells in the chronic VL-HIV group at EOT ($p = 0.049$), we did not observe any other difference in the MFI of KLRG1 on CD8⁺ and CD8⁻ T cells between the ncVL-HIV and cVL-HIV groups at any timepoint nor across the study duration (Supplementary Fig. 8b, c, e, f, h, i, k, l). In addition, we observed a decrease in the MFI of CD57 on CD8⁺ T cells in the VL-HIV group

compared to the LS+-HIV group (Supplementary Fig. 8g), but no other difference was found in the MFI of CD57 between any group at any timepoint.

Together, these findings indicate a marked increase in the exhaustion-associated markers TIGIT and PD-1 on CD8⁺ (and to a lower extend CD8⁻) T cells of chronic VL-HIV patients, which are persistently present during and in between active VL episodes and even during intermittent cure. In addition, the lack of differences in T cell senescence-associated markers could indicate a reversible nature.

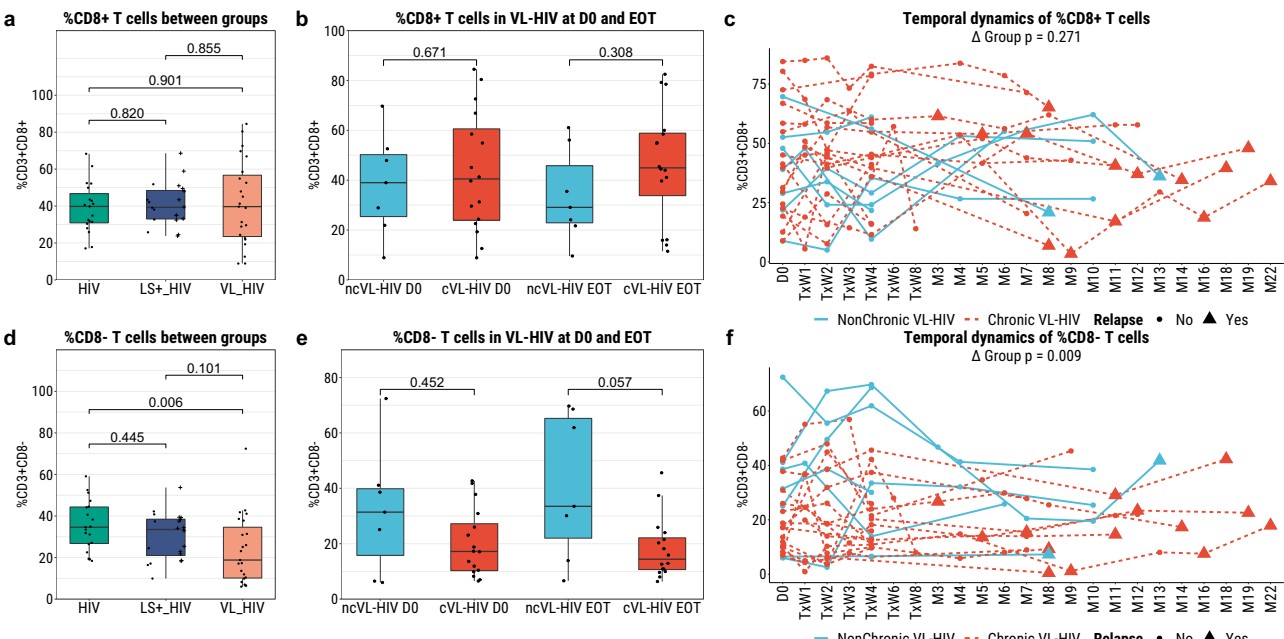

**Fig. 2 | The cellular composition of PBMCs isolated from participants as measured by flow cytometry. a**, **d** Cross-sectional analysis of the frequencies of CD8$^+$ T cells (CD3$^+$CD8$^+$) and CD8$^-$ T cells (CD3$^+$CD8$^-$), respectively, at active disease development for the VL-HIV group (D0; $n = 23$) and at study recruitment for the HIV ($n = 19$), and *Leishmania*-seropositive HIV-coinfected (LS+-HIV; $n = 20$) groups. In the latter group, the plus sign (+) indicates those *Leishmania*-seropositive

individuals with a history of VL. **b**, **e** Comparison of the frequencies of CD8$^+$ T cells (CD3$^+$CD8$^+$) and CD8$^-$ T cells (CD3$^+$CD8$^-$), respectively, for non-chronic (blue; $n = 7$) or chronic (red; $n = 16$) VL-HIV patients only, at active disease development (D0) and at the End-of-Treatment (EOT). **c**, **f** Longitudinal profiling of the cellular frequencies from time of VL development, for non-chronic (blue; $n = 7$) or chronic (red; $n = 17$) VL-HIV patients only.

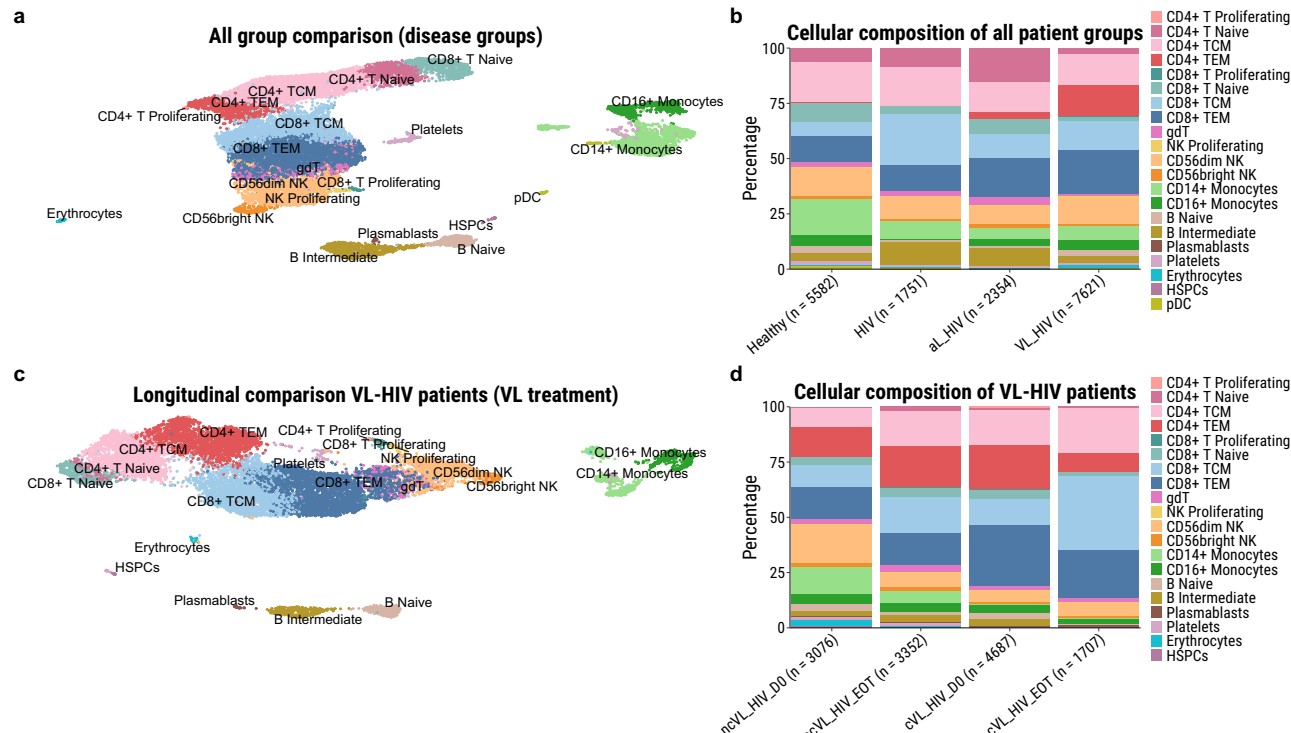

**Fig. 3 | The cellular composition of PBMCs isolated from participants as measured by single-cell RNA sequencing. a** UMAP representation of 17.308 primary PBMC derived from two representative cases of healthy endemic controls, individuals with HIV, individuals with AL-HIV (*Leishmania*-seropositive individuals with no history of VL), and four VL-HIV patients at active disease development. **b** Cellular proportions of inferred immune cell types in the PBMC fractions of the

different participant groups at D0, n = number of cells per patient group. **c** UMAP representation of 12.822 primary PBMC derived from two non-chronic (and long-term cured) and two chronic VL-HIV patients, all sampled at D0 and EOT. **d** Cellular composition of inferred immune cell types in VL-HIV participant PBMC fractions across the different timepoints of the disease course, n = number of cells per participant group and timepoint.

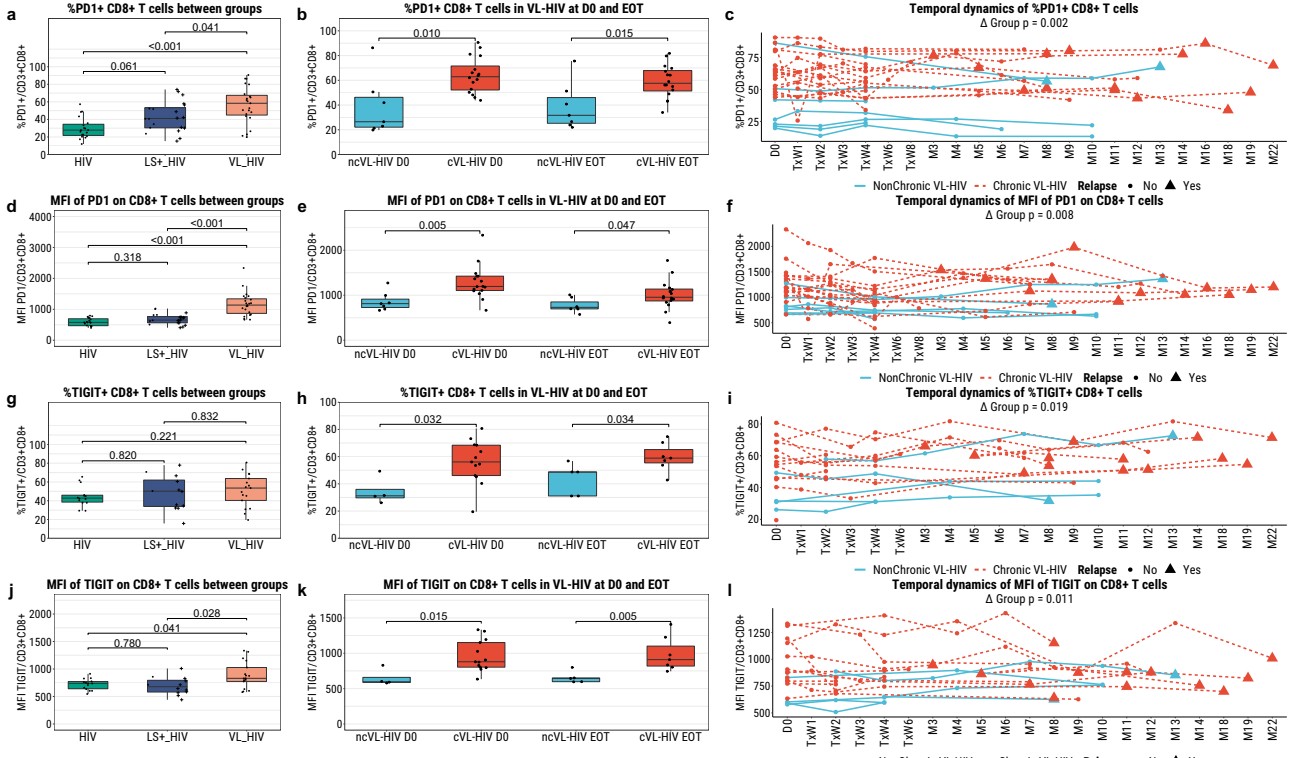

**Fig. 4 | The PD-1⁺ and TIGIT⁺ CD8⁺ (CD3⁺CD8⁺) T cell fractions of PBMCs isolated from the different participant groups as measured by flow cytometry. a, d, g, j** Cross-sectional profiling of the frequency of PD-1⁺ CD8⁺ T cells, the MFI of PD-1 on CD8⁺ T cells, the frequency of TIGIT⁺ CD8⁺ T cells, and the MFI of TIGIT on CD8⁺ T cells, respectively, between the VL-HIV (PD-1 *n* = 23, TIGIT *n* = 17), the *Leishmania*-seropositive HIV (PD-1 *n* = 20, TIGIT *n* = 12) and the HIV-only (PD-1 *n* = 19, TIGIT *n* = 14) groups, using a Benjamini-Hochberg corrected pairwise Mann-Whitney U test to test for statistical differences. The plus sign (*) indicates those *Leishmania*-seropositive individuals with a history of VL.

**b, e, h, k** Comparison between chronic (PD-1 *n* = 16 at both timepoints, TIGIT *n* = 13 at D0 and *n* = 7 at EOT) and non-chronic VL-HIV patients (PD-1 *n* = 7 at both timepoints, TIGIT *n* = 4 at D0 and *n* = 5 at EOT) at the active disease development (D0) and End-of-Treatment (EOT) timepoints using a BH-corrected Mann-Whitney U to test for statistical differences. **c, f, i, l** Longitudinal characterisation of the frequencies and MFIs of the same cellular subsets, for non-chronic (PD-1 *n* = 7, TIGIT *n* = 6) or chronic VL-HIV (PD-1 *n* = 17, TIGIT *n* = 16) patients only, using linear mixed-effects models (see Methods).

## Lower functionality of TIGIT⁺ CD8⁺ and CD8⁻ T cells suggests exhaustion in chronic VL-HIV patients

Next, we questioned whether the general immune capacity of cVL-HIV patients is decreased, and whether the higher frequency of TIGIT-expressing T cells in these patients is accompanied with an impaired functionality of these cells.

With regard to the first, we first looked at differences in the functionality of the general T cell compartments. Here, we observed lower levels of IFN-γ on the CD8⁺ and CD8⁻ T cell populations in VL-HIV patients compared to the LS+-HIV group, but not participants with HIV (Supplementary Fig. 9a, d). However, we did not observe any difference between non-chronic and chronic VL-HIV patients at any timepoint nor across the study duration (Supplementary Fig. 9b, c, e, f). Similarly, while we observed a higher level of CD107α⁺ CD8⁺ T cells in VL-HIV patients compared to the other participant groups, we did not observe any difference between ncVL-HIV and cVL-HIV patients (Supplementary Fig. 9g–i). This suggests no difference in the overall functional capacity of the T cell compartment between chronic and non-chronic VL-HIV patients

Next, we studied whether TIGIT expression was accompanied with a decreased cell functionality suggestive of T cell exhaustion that may underlie VL chronicity. We observed a lower frequency of IFN-γ⁺TIGIT⁻ CD8⁺ T cells in chronic VL-HIV patients compared to non-chronic VL-HIV patients at D0, but not at EOT, although it did persist across the study duration (Fig. 5 a, b). Similarly, we observed a higher frequency of IFN-γ⁻TIGIT⁺ CD8⁺ T cells in cVL-HIV patients than in ncVL-HIV patients at D0, but not at EOT nor across the study duration (Fig. 5 c, d). We did not observe any difference in the frequency of IFN-γ out of TIGIT⁺ nor TIGIT⁻

CD8⁺ T cells at any timepoint (Fig. 5e–h). Unlike the CD8⁺ T cell population, the frequency of IFN-γ⁺TIGIT⁻ CD8⁻ T cells was lower in cVL-HIV patients than in ncVL-HIV patients at D0, EOT, and across the study duration (Supplementary Fig. 10 a, b). Concurrently, the frequency of IFN-γ⁻TIGIT⁺ CD8⁻ T cells was higher in cVL-HIV than in ncVL-HIV patients at D0, EOT, and across the study duration (Supplementary Fig. 10 c, d). In addition, the frequency of IFN-γ⁺ cells out of the TIGIT⁻CD8⁻ T cell population was higher in ncVL-HIV than cVL-HIV patients across the study duration, but not at D0 nor at EOT (Supplementary Fig. 10 g, h).

## Pronounced CD4⁺ and CD8⁺ T cell treatment unresponsiveness in chronic VL-HIV patients at the single-cell transcriptional level

To better characterise a potential impaired CD4⁺ and CD8⁺ T cell functionality underlying VL chronicity, we studied the transcriptional activity at the single T-cell level before and after VL treatment in two chronic and two non-chronic cases in detail (see participants indicated with * in Fig. 1b).

In CD4⁺ T cells, the 'Interferon gamma response', 'T cell activation'-related, and 'antigen processing and presentation' pathways were highly upregulated in ncVL-HIV patients after treatment compared to active disease (Fig. 6a, Supplementary Fig. 11, and Supplementary Fig. 12a). In addition, the CD4⁺ T cells of the non-chronic VL-HIV patients have a lower expression of *TIGIT*, but not *PDCD1* (the gene encoding for PD-1), after parasitological treatment, further confirming our findings on the flow cytometry level (Fig. 5a). This is in sharp contrast to the cVL-HIV patients after parasitological treatment, which showed a consistent downregulation of these pathways instead (Fig. 6b and Supplementary Fig. 12b). This pattern is similar for CD8⁺ T cells, although less prominent (Fig. 6c, d and

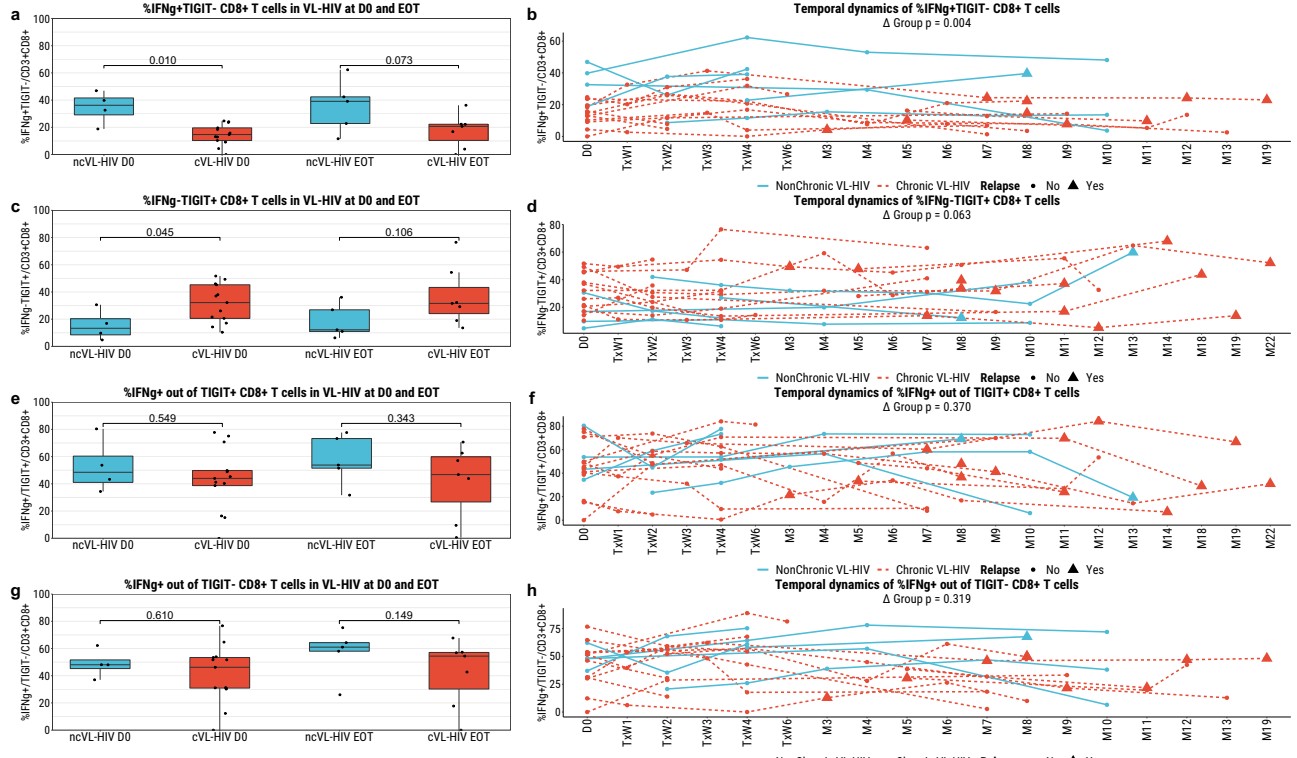

**Fig. 5 | Interferon-γ positivity in the TIGIT$^+$ and TIGIT$^-$ CD8$^+$ (CD3$^+$CD8$^+$) T cell fractions of PBMCs isolated from the different participant groups as measured by flow cytometry. a, c, e, g** Comparison of the frequencies of IFN-γ$^+$TIGIT$^-$ cells out of all CD8$^+$ T cells, IFN-γ$^-$TIGIT$^+$ cells out of all CD8$^+$ T cells, IFN-γ$^+$ cells out of TIGIT$^+$CD8$^+$ T cells, and IFN-γ$^+$ cells out of TIGIT$^-$CD8$^+$ T cells, respectively, between the chronic ($n$ = 13 at D0, $n$ = 7 at EOT) and non-chronic ($n$ = 4 at D0, $n$ = 5 at EOT) VL-HIV patients at the active disease development (D0) and End-of-Treatment (EOT) timepoints using a BH-corrected Mann-Whitney U to test for statistical differences. **b, d, f, h** Longitudinal characterisation of the frequencies of the same cellular subsets, for non-chronic or chronic VL-HIV patients, using linear mixed-effects models (see Methods).

Supplementary Fig. 12c, d). These findings suggest a generalised lack of T cell responsiveness in the chronic VL-HIV patients during parasitological treatment. This pattern is also apparent when comparing the differentially expressed genes between the non-chronic and the chronic VL-HIV groups at both timepoints, with non-chronic VL-HIV patients generally showing an increased expression of T cell activation -and antigen presentation-related genes after parasitological treatment, compared to chronic VL-HIV patients (Supplementary Fig. 12a–d). Furthermore, non-chronic VL-HIV patients had a lower expression of *TIGIT* in CD4$^+$ (at EOT only) and CD8$^+$ T cells compared to chronic VL-HIV patients (Supplementary Fig. 12a–d). Similarly, non-chronic VL-HIV patients had a lower expression of *PDCD1* in CD4$^+$ T cells compared to chronic VL-HIV patients at both timepoints (Supplementary Fig. 12a, b).

Together, these findings confirm an apparent T cell exhaustion status in chronic VL-HIV patients that was suggested on the protein-level by flow cytometry, which may underlie the inability to clear parasites during and after treatment.

**Lack of lymphoproliferative response in chronic VL-HIV patients during VL treatment**

Lastly, we evaluated the proliferative response or clonal expansion of T cells during VL treatment as a proxy marker for the specific response against *Leishmania*.

When comparing the clonal expansion of the top 30 clonotypes in the ncVL-HIV and cVL-HIV patients after treatment, we observed a marked oligoclonal expansion in ncVL-HIV patients (Fig. 7a), but no clonal expansions in those with cVL-HIV (Fig. 7b). The oligoclonal expansion in ncVL-HIV patients occurred exclusively in both the effector-memory CD4$^+$ and CD8$^+$ T cell subsets, in contrast to a less expanded and restricted CD8$^+$ T cell repertoire in those with cVL-HIV (Fig. 7c, d). The clonotypic

expansion in ncVL-HIV patients is consistent with a *Leishmania*-specific response. The lack of any lymphoproliferative responses after parasitological treatment in cVL-HIV patients further strengthens the observation of parasite tolerance and persistent CD4$^+$ T cell anergy.

## Discussion

In the search for underlying biomarkers of VL disease chronicity in HIV co-infected individuals, we carried out an in-depth and longitudinal evaluation of peripheral blood mononuclear cells in participants with non-chronic and chronic VL-HIV disease living in a remote area in NW-Ethiopia. Out of a large panel of T cell exhaustion- and senescence-associated markers, we demonstrated high levels of TIGIT and PD-1 on both CD8$^+$ and/or CD8$^-$ T cells, that persisted during antileishmanial treatment and across multiple relapse episodes, as marked immunological determinants of a chronic VL disease course in HIV co-infected individuals. In addition, we observed a pronounced CD4$^+$ T cell unresponsiveness or anergy that stably persisted overtime at the transcriptional, protein, and T cell clonotype level in VL-HIV patients that experienced recurrent or chronic VL disease. Most interestingly, this discriminatory signature could already be observed at time of active VL development, suggesting predictive value for VL relapse development and disease chronicity.

Our observation that the majority (80%) of VL-HIV patients experience recurrent VL disease, with VL relapses developing mostly after 6 months from end of treatment, indicates the high need for prognostic and/or predictive markers of VL disease chronicity. Besides comprehensively studying T cell characteristics in a unique cohort, to the best of our knowledge, this is the first study to have a long-term longitudinal screening that showed the early generation and persistence of T cell markers towards chronicity[10,29,32,33]. Our findings on increased levels of PD-1 on CD8$^+$ and CD8$^-$ T cells in chronic VL-HIV patients are in line with observations in a

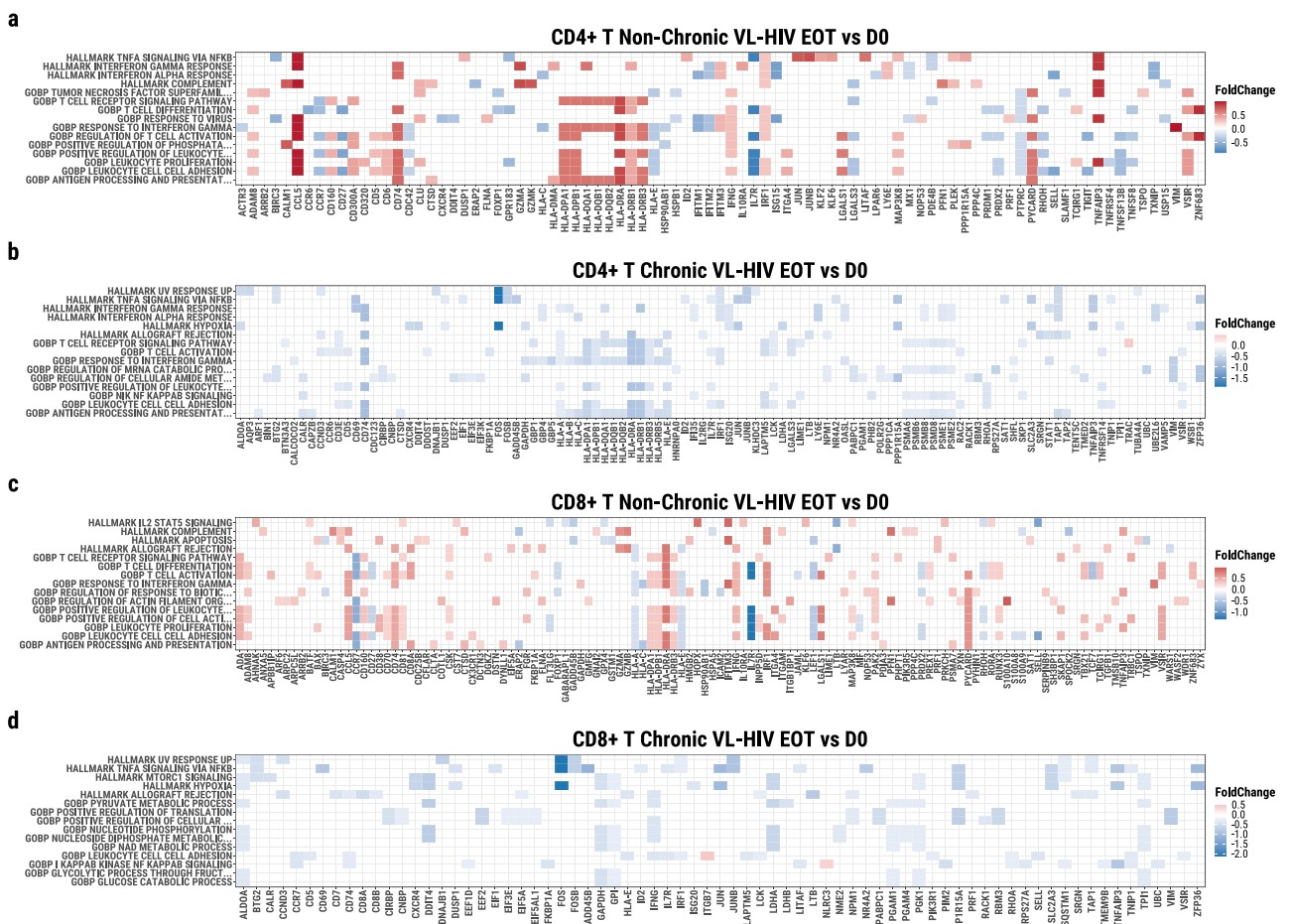

**Fig. 6 | Pathway overrepresentation visualised as a heatmap performed on the differentially expressed genes between. a** CD4$^+$ T cells of non-chronic VL-HIV at End-of-Treatment (EOT) versus active disease (D0), **b** CD4$^+$ T cells of chronic VL-HIV at EOT versus D0, **c** CD8$^+$ T cells of non-chronic VL-HIV patients at EOT versus D0, **d** CD8$^+$ T cells of chronic VL-HIV patients at EOT versus D0. N = 2 each group and timepoint.

similar cohort, as Takele et al. showed higher PD-1 expression on both CD4$^+$ and CD8$^+$ T cells in VL-HIV patients that subsequently experienced VL relapse than in those that did not[32]. More recently, these authors also observed that both primary and recurrent VL-HIV patients exhibited higher PD-1 expression on CD4$^+$ and CD8$^+$ T cells than controls, and that the PD-1 level of primary VL-HIV patients dropped after treatment while those of the recurrent VL-HIV patients did not[34]. Together, our shared findings strongly point towards increased levels of PD-1 as an underlying determinant of a chronic VL disease course with recurrent VL relapses in HIV-co-infected individuals. In line, we additionally identified TIGIT on CD8$^+$ and CD8$^-$ T cells to be a robust immunological determinant underlying VL chronicity in VL-HIV co-infection. Unfortunately, due to flow cytometry panel limitations, we could not include TIGIT and PD-1 in the same panel, preventing us from performing any co-expression analyses for these two markers. Nonetheless, our findings suggest that PD-1 and TIGIT may have strong predictive value for recurrent relapse and a chronic VL disease course in HIV-co-infected individuals.

As TIGIT and PD-1 are inhibitory receptors often expressed on T cells upon activation, expression of these molecules is often either associated with early T cell activation and higher T cell functionality, or with a state of T cell exhaustion that has limited T cell functionality[35,36]. Although we did not observe a difference in the general T cell functionality on the flow cytometry level (based on IFN-γ and CD107α) between chronic and non-chronic VL-HIV patients, we showed that TIGIT$^+$ T cells are less performant, indicating an exhausted state, evidenced by a higher level of IFN-γ$^-$TIGIT$^+$ T cells and a lower level of IFN-γ$^+$TIGIT$^-$ T cells in chronic VL-HIV patients. This may suggest a decreased IFN-γ functionality of TIGIT$^+$ T cells in chronic VL-

HIV patients and can be indicative of T cell exhaustion although exhausted T cells can often still retain some functionality[37,38]. Unfortunately, due to flow cytometry panel limitations, we could not confirm these findings for PD-1 T cells as well. Despite this, concurrent to our findings on TIGIT, Takele et al. observed a correlation between the failure to restore antigen-specific production of IFN-γ (in a whole blood stimulation assay) and the high expression of PD-1 on CD4$^+$ T cells in VL-HIV patients[32]. Thus, in contrast to VL patients who may also present with higher exhaustion markers but still produce an antigen-specific IFN-γ response in a whole blood stimulation assay, VL-HIV patients present with both higher PD-1 levels and a complete lack of antigen-specific IFN-γ response (at T cell and whole blood level)[39]. Together, these findings strengthen the hypothesis that increased levels of TIGIT and PD-1 may be reflective of increased T cell exhaustion or even anergy in chronic VL-HIV. At the single-cell transcriptome level, which unlike the flow cytometry-level, is unrestricted by marker selection, we did observe a trend for general functional differences suggestive of T cell exhaustion. Further suggestive of T cell exhaustion, on the single-cell T cell clonal level, we also observed a lack of T cell expansion after treatment in cVL-HIV patients, but not in those with ncVL-HIV, primarily in the CD4$^+$ T cell population. This lack of a lymphoproliferative response in recurrent VL-HIV disease has also been observed in vitro using an SLA stimulation cellular proliferation test[33]. This suggests the CD4$^+$ T cells of VL-HIV patients are not only functionally impaired, but may be anergic and unable to respond to parasite stimuli. Whether this is due to defective antigen recognition, or defective co-stimulatory signalling, is not yet fully understood. Together, these findings strongly demonstrate increased T cell exhaustion-associated markers and concurrent functional

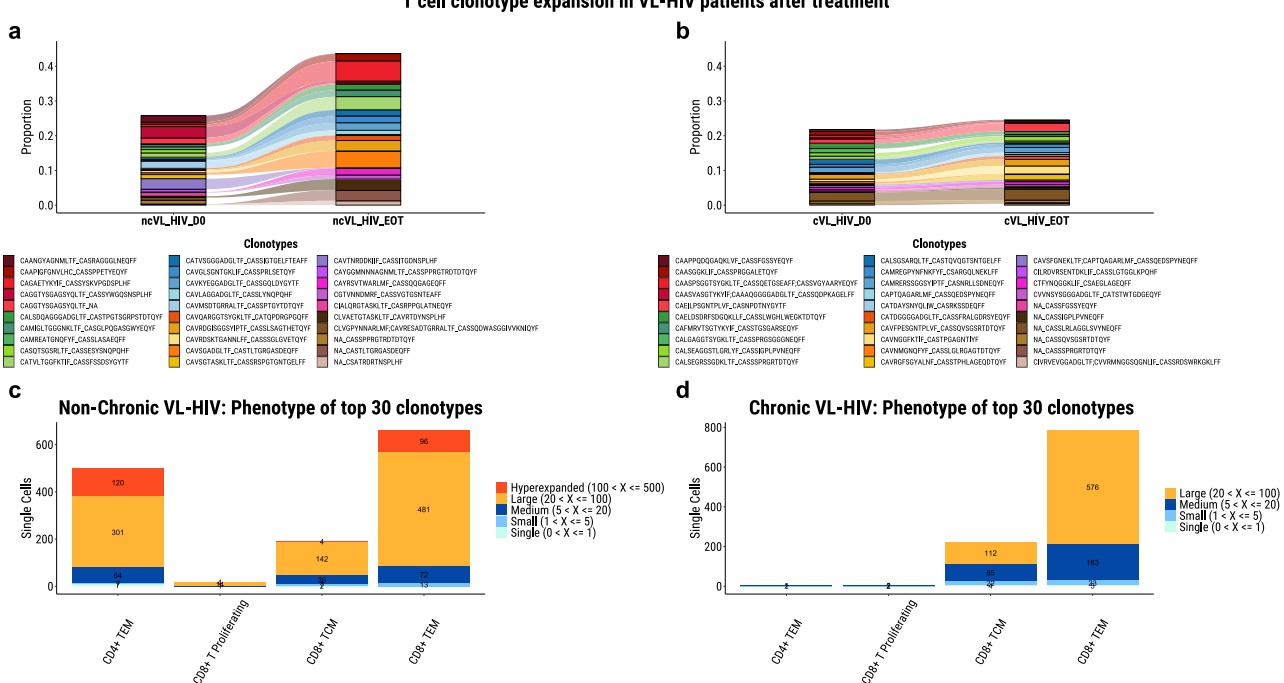

**Fig. 7 | The proportions of the top 30 T cell clonotypes per group, and their phenotypes.** The proportions of the top 30 T cell clonotypes per group, and their phenotypes of the (**a**) non-chronic VL-HIV and (**b**) chronic VL-HIV groups, at the start and end of VL treatment. **c** Phenotypes of the top 30 clonotypes of the non-chronic VL-HIV patients and **d** Phenotypes of the top 30 clonotypes of the chronic VL-HIV patients. *n* = 2 of each condition.

impairment of T cells of VL-HIV patients, especially those that experience chronic VL disease, suggesting a role for host-directed immunotherapy as adjuvant next to the traditional parasite-targeting therapies.

Next to a high PD-1 and TIGIT expression on pan T cells, the VL-HIV patients also showed increased levels of LAG-3 and TIM-3, although these levels did not distinguish between non-chronic and chronic VL-HIV. The co-expression of multiple immune checkpoint proteins on the same cell, in addition to more detailed assessments of the *Leishmania*-specific functional and proliferative response, is required to robustly define T cell exhaustion, and define the eligibility of patients for immune checkpoint blockage therapy. Similar to cancer research, blocking only the PD-1 pathway could have suboptimal effects when other immune checkpoint pathways are equally activated but not inhibited[40]. For instance, the high T cell exhaustion levels could be reversed with TIGIT/PD-1 co-blockade using the currently approved immune checkpoint inhibitors in cancer research[4,41]. While PD-1 and TIGIT as inhibitory molecules exert their inhibitory function primarily on different stimulatory targets (TIGIT targets CD226, while PD-1 primarily targets CD28), they can co-inhibit CD226, and inhibiting both is required to restore full CD226 signalling activity[35]. Although evidence on PD-1 blockade in human VL is still scarce, in canine VL, blockade of the PD-1 pathway with such an immune checkpoint inhibitor led to the restoration of the T cell effector function, reversing the exhaustion[42]. Similarly, in experimental VL, PD-1 blockade led to a reduced parasite load[43]. In HIV infection alone, the co-blockade of TIGIT and PD-1 led to greater restoration of the viral-specific CD8+ T cell response than single blockade by either, further suggesting the potential of co-blockade[44]. Especially in difficult-to-cure VL-HIV patients, the associated high toxicity and variable cure rates with such immune checkpoint inhibitors may be within the acceptable range[45].

The reversible nature of T cell exhaustion is vital for such an immunomodulatory strategy. Accordingly, we did not observe overt irreversible T cell senescence in VL-HIV patients nor cVL-HIV patients. While a previous study demonstrated higher T cell senescence levels in VL-HIV patients as compared to healthy individuals, this was irrespective of the chronicity, with no observed difference between recurrent and non-recurrent VL-HIV[46].

More in-depth studies targeting other markers of cellular senescence, such as telomere length, are crucial to discriminate between reversible T cell exhaustion and irreversible T cell senescence processes, to disentangle the pathways for future immunotherapeutic approaches.

One of the major limitations of our study is the lack of a VL group without HIV co-infection. This would have allowed us to better characterise T cell exhaustion or anergy in VL-HIV patients versus VL patients. A minor limitation of our study is that the majority of our participants with HIV are female, compared to an exclusively male population in the other groups. While this does more accurately reflect the actual population distribution of HIV and *Leishmania* infection, it may have confounded the comparison with the predominantly male VL-HIV group. Another caveat is that more than half (65%) of our participants with asymptomatic *Leishmania* and HIV co-infection are long-term cured individuals that reported VL history with a low number (<2) of VL episodes. However, both true asymptomatic and long-term cured participant groups represent populations able to successfully control the parasite, whether it was before or after disease. This LS+-HIV group has similar levels of T cell exhaustion-associated markers and cytokine expression as HIV controls, indicating *Leishmania* infection by itself or prior VL history does not lead to higher levels of T-cell exhaustion or anergy, and further confirmed its specificity to those with a chronic disease course. Although some prior studies stimulate with soluble *Leishmania* antigen, which could possibly provide more information on the VL-specific immune response, we stimulated the T cell functionality panel with a leucocyte activation cocktail to assess whether the T cells were still capable of sufficient cytokine production[32–34]. As such, we wanted to evaluate the maximum capacity of these cells to produce a given cytokine, and get a general idea of the host immune state in a complex co-infection rather than evaluating a *Leishmania*-specific response, given a potential synergistic role of HIV. A final limitations is that we only capture the transition of non-chronic to chronic disease stage for two of the non-chronic individuals, complicating any inferences. Thus, larger longitudinal analyses of non-chronic VL-HIV patients are essential to better understand the role of these T cell markers in the shift to a chronic VL phenotype.

In this work, we refrained from predictive modelling due to sample size limitations and class imbalance. However, Takele et al. showed that the combination of CD4+ T cell counts, whole blood IFN-γ levels after SLA stimulation, and PD-1 levels on CD4+ T cells, is able to better predict relapse development at the end of treatment than CD4+ T cell counts alone[32]. Although their model achieved high performance, the authors did not use an external test/validation set or cross-validation for performance assessment and could have risked overfitting[47,48]. Nevertheless, these findings make a compelling case to look beyond the CD4+ T cell count in VL-HIV outcome prediction. The early markers associated with CD8+ T cell exhaustion and CD4+ T cell anergy we identified may help clinical research as surrogate endpoints. If they are translated to more point-of-care amenable assays, they can help guide the clinical decision making process, which may include treatment optimisation or extension, and the initiation of secondary prophylaxis in high risk individuals. This translation is encouraged by our finding of a generalised and reduced T cell functionality or increased T cell exhaustion in the periphery instead of a *Leishmania*-specific finding, eliminating the need for complex measurements requiring cell stimulation with *Leishmania* antigens.

In conclusion, we consistently observed higher levels of the T cell exhaustion-associated markers PD-1 and TIGIT in chronic VL-HIV cases on ART, and showed that the cells of these patients may be functionally impaired too, suggestive of T cell exhaustion or even T cell anergy. The prognostic value of these markers should be further validated and incorporated into clinical algorithms to allow prediction of VL relapse and disease chronicity, and guide the use of secondary prophylaxis. Based on our data and previous data, we argue that chronic VL-HIV patients require a restoration of their T cell response by immunomodulatory therapy together with antileishmanial treatment to enable parasite clearance. We believe such an immunochemotherapy approach is necessary to break the recurrent VL disease episode cycle that the majority of VL-HIV patients experience.

## Methods
### Ethics statement
The study protocol was approved by the Ethiopian National Research Ethics Review Committee, the University of Gondar Institutional Review Board, the Institute of Tropical Medicine Antwerp Institutional Review Board, the Médecins Sans Frontiers Ethics Review Board and the Antwerp University Hospital Ethics Committee. All participants provided written informed consent and the study was carried out in accordance with international guidelines (Helsinki Declaration, Good Clinical Practices and local regulations). All ethical regulations relevant to human research participants were followed.

### Study population and design
This work was embedded in the prospective PreLeisH cohort study in North-West Ethiopia (clinicaltrials.gov NCT03013673), in which 570 adult individuals in HIV care at the study site (and free of VL at time of enrolment) were followed up for up to 24 months between October 2017 and November 2020 at three to six months intervals to monitor *Leishmania* infection and further VL development at the Médecins Sans Frontières (MSF)-supported Abdurafi Health Centre. Participants that developed active VL were hospitalised, received free treatment according to routine care, and entered a follow-up phase to monitor treatment outcome and long-term relapse-free survival. This follow-up phase included two scheduled visits at 6 and 12 months after treatment end, and at any suspected VL relapse episode until the last patient's last visit occurred (study end).

In this substudy, we included the first 24 participants that developed active VL during the study period between 2017 and 2019 (excluding those lacking blood sampling) (Fig. 1a). These 24 patients were stratified in non-chronic patients (ncVL-HIV; *n* = 7), that included five individuals with no VL history and two with one prior VL episode more than 10 years (with a median of 185 (IQR 164-206) months) before enrolment in our study, and chronic patients (cVL-HIV; *n* = 17) with one or more recent VL episodes all within 3 years (with a median of 5 (IQR 4-10) months) prior to study

recruitment. As control groups, we included HIV-positive participants with no VL history and no asymptomatic *Leishmania* infection over the complete study period (HIV; *n* = 19; based on consecutive and confirmatory sampling); and a mixture of asymptomatic *Leishmania*-infected and long-term VL cured HIV-positive participants that tested positive for at least 2 or more *Leishmania* markers (rK39 RDT, rK39 ELISA, DAT, KATEX, PCR) at consecutive timepoints during the study period (*Leishmania*-seropositive-HIV, LS+-HIV; *n* = 20; consecutive sampling).

All 63 participants underwent blood sampling with 10 mL Sodium-Heparin CPT™ Mononuclear Cell Preparation tubes at each timepoint (missing samples visualised in Fig. 1b), and subsequently PBMCs were isolated and cryopreserved in a freezing solution (50% FBS, 40% RPMI, 10% DMSO) within 4–6 h after collection. Flow cytometry was performed on all participants and all available timepoints of these patients. Furthermore, we sampled two individuals of each of the described participant groups (HIV, LS+-HIV without VL history (AL-HIV), ncVL-HIV, cVL-HIV), in addition to two anonymised healthy endemic controls, for unbiased processing with single-cell 5' RNA and TCR sequencing (participant characteristics described in Supplementary Table 1). For this, participant samples were chosen based on the highest cell viabilities. In addition, for the VL-HIV patients, the two most representative patients of the ncVL-HIV and cVL-HIV groups were chosen based on their VL chronicity (number of VL episodes prior to our study and the number of VL episodes during our study). For the VL-HIV patients, samples at the time of active VL disease development (D0) and the end of parasitological treatment (D28) were included, while for the remaining participant groups the sampling at time of recruitment was included (grand total of 12 samples).

### Flow cytometry
Thawed PBMCs were washed and subjected to two separate antibody panels (Miltenyi Biotec, USA), according to cell availability. The CD8+ T-cell exhaustion-targeted panel consisted of anti-CD3-VioGreen, anti-CD8-VioBlue, anti-CD57-APC-Vio770, anti-LAG-3-APC, anti-KLRG1-PE-Vio770, anti-PD-1-PE, anti-TIM-3-VioBright-FITC, and 7AAD for viability. The CD8+ T-cell functionality-targeted panel consisted of anti-CD3-VioGreen, anti-CD8-VioBlue, anti-IFNy-Vio667, anti-CD95-PE-Vio770, anti-TIGIT-PE, anti-CD107α-FITC, and FVS780 for viability. For the CD8+ T cell functionality-targeted panel, cells were thawed, washed and pre-stimulated with 1 µl of Leukocyte Activation Cocktail with GolgiPlug (LAC, BD Biosciences, Belgium) per 300.000 cells, for 4 h in a humidified incubator at 5% $CO_2$ and 37 °C. In addition to this, anti-CD107α antibody was immediately added to the cells with LAC (with GolgiPlug), and after 1 hour, GolgiStop was added for the remaining 3 h of incubation. Afterwards, cells were fixed and permeabilised before intracellular co-staining using the FOXP3/Transcription Factor Staining Buffer Set (eBioscience, USA) according to the manufacturer's instructions. After fixation/permeabilisation, cells were washed and immediately acquired on a FACSVerse flow cytometer (BD Biosciences, San Jose, CA), including prior FMO controls and single stainings for each parameter. The gating strategy is outlined in Supplementary Figs. 1, 2. Analysis was performed using FlowJo v8.5 and the cellular subsets were expressed as frequencies of parent, and mean fluorescence intensities of each marker were exported as well. Any sample with a viability less than 25% was excluded from further analysis. This threshold was chosen by correlation analyses between the sample viability and all markers, and no correlation could be observed above a 25% threshold.

### Single-cell 5' RNA and T cell receptor sequencing
Thawed PBMCs were used for single-cell 5' RNA and TCR sequencing using the 5' Chromium Single Cell Immune Profiling v1 solution from 10X Genomics, according to the manufacturer's instructions. Thawed PBMCs were directly loaded on the 10x Genomics A Chip Kit for cellular barcoding. The resulting cell-barcoded gene expression libraries were sequenced on an Illumina NovaSeq6000 at an average read depth of approximately 130.000 reads/cell, while the V(D)J amplified libraries were sequenced on an Illumina NextSeq at an average read depth of approximately 30.000 reads/cell.

After demultiplexing the sequencing output using the *mkfastq* command of the CellRanger version 6.0.0 software, the *multi* pipeline of this software was used with default settings to map the gene expression library reads to the GRCh38 reference genome while simultaneously aligning the reads from the V(D)J library to the V(D)J-compatible GRCh38 5.0.0 reference set. Next, the resulting gene expression count matrices were analysed using the Seurat package version 4.1 in R[49]. Briefly, the data of individual samples were integrated using Seurat's *merge* command. To remove multiplets and other low quality cells, all cells expressing <800 and >3.000 genes were filtered out. In addition, for the same purpose, only cells expressing between 2.5% and 12.5% mitochondrial gene counts and above 5% ribosomal gene counts were included. The data was then log-transformed, scaled, and centred. Next, the 2.000 most variable genes were identified and used as input for a principal component analysis. The top 15 principal components were then used for clustering, and for dimension reduction and visualisation using the Uniform Manifold Approximation and Projection (UMAP) method[50]. The resulting cell clusters were annotated using an ensemble method of reference-based mapping based on the SingleR package version 1.8.0, and with a gating strategy as employed by the scGate package version 1.0.0, using their pre-defined custom models[51,52]. More specifically, the Human Primary Cell Atlas reference of the celldex R package was used for SingleR mapping.

### Differential gene expression and pathway analysis
Differentially expressed genes (DEGs) were identified using the *FindMarkers* function in Seurat with the default Wilcoxon Rank Sum test. A log fold change threshold of 0.25 was applied, and only genes with a Benjamini-Hochberg adjusted *p* value of <0.05 were considered for further analysis. The identified DEGs were subjected to over-representation analysis using the clusterProfiler package, and the top 12 enriched pathways were visualised using the enrichplot package[53]. Gene sets used include the Hallmark (H) and GO ('C5') categories from the MSigDB 7.4[54].

### T cell receptor repertoire analysis
The scRepertoire package version 1.7.2 was used to analyse the TCR data at single-cell resolution[55]. Clonotypes were defined using the unique combination of V(D)J genes and the CDR3 nucleotide sequence. To track changes in clonotype dynamics after treatment, we used the *compareClonotypes* function on the top 30 clonotypes of the ncVL-HIV and cVL-HIV groups. Finally, to look at the functionality of the top clonotypes, we used the *occupiedscRepertoire* function to look at the clonotype frequency per T cell phenotype.

### Statistics and reproducibility
For participant characteristics, continuous variables were represented as medians with interquartile ranges (IQR) and categorical data as numbers and proportions. To compare continuous variables across the three participant groups (HIV, LS+-HIV, and VL-HIV), the Kruskal-Wallis test was performed. Similarly, to compare continuous variables across the non-chronic and chronic VL-HIV groups, a Mann-Whitney U test was performed. Statistical testing for categorical variables was performed with a Fisher's exact test. To test for differences in flow cytometry markers across participant groups at study recruitment for HIV and LS+-HIV participants (D0) and at active disease development for VL-HIV patients (D0), pairwise Mann-Whitney U tests were performed between all groups. To test for differences in flow cytometry markers between non-chronic and chronic VL-HIV patients, a Mann-Whitney U test was performed at D0 and at End-of-Treatment (EOT). Subsequently, for longitudinal analyses, linear mixed-effects models were implemented, using the lmerTest R package version 3.1.3, to test for differences in marker frequencies between patients with non-chronic or chronic VL-HIV across their disease course[56]. Here, for each marker, interactions between the participant group and the measurement timepoint as fixed effects were modelled, with each individual serving as a random effect to control for repeated measures within the individual. The reported p-values were Benjamini-Hochberg adjusted to correct for multiple testing, and only adjusted *p* values lower than <0.05 were considered

significant. All tests were two-sided. For both the flow cytometry and single-cell RNA and single-cell TCR sequencing assays, the sample sizes are included in figure legends (n), and these sample sizes indicate the number of patients. For the single-cell analyses, the number of cells are mentioned in the text. Experiments were not repeated, but for the flow cytometry assay, repeated measures of the same patients were taken. For the single-cell assays, repeated measures were only taken for the non-chronic and chronic VL-HIV patients.

### Reporting summary
Further information on research design is available in the Nature Portfolio Reporting Summary linked to this article.

## Data availability
Processed single-cell RNA and T cell receptor sequencing data has been deposited on Zenodo at https://doi.org/10.5281/zenodo.10887674[57]. Raw sequencing data has been deposited on NCBI's Sequence Read Archive under the accession code https://www.ncbi.nlm.nih.gov/bioproject/PRJNA1093393. Flow cytometry source data is also available on request, but values used to create the figures have been provided in Supplementary Data File 1 and on the Zenodo repository[57]. Patient characteristics data will remain restricted due to privacy and ethics concerns, however, summary statistics have been provided in the manuscript.

## Code availability
All code is available on GitHub (https://github.com/BioinformaNicks/PreLeisH_Immuno) and Zenodo (https://doi.org/10.5281/zenodo.10887674).

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

## Acknowledgements
This work was supported by the Research Foundation Flanders (FWO) fellowships to N.D.V. and W.A., the Institute for Tropical Medicine Antwerp's SOFI programme supported by the Department of Economy, Science and Innovation of the Flemish Government, and the Belgian Directorate General for Development Cooperation under the ITM-DGDC framework agreement FA-III & FAIV. The funders had no role in study design, data collection and analysis, decision to publish, or preparation of the manuscript. The computational resources used for this work were provided by the VSC (Flemish Supercomputer Center) at the University of Antwerp. Our gratitude goes to all participants involved in this study, and all the Leishmaniasis Treatment and Research Center and MSF Abdurafi staff members who were actively involved in the care of these participants. The authors also thank the Drugs for Neglected Disease Initiative and the University of Gondar for supporting the Leishmaniasis Research and Treatment Center. The authors want to thank Lieselotte Cnops, and the entire MSF Abdurafi and BoH Abdurafi teams, with special thanks to Daniel, Mercy, Fikadu, Hamid, Esthetie, and Janice. Finally, N.d.V. wants to thank his dearly beloved late fiancée Sofie Van de Poel for her eternal love and support.

## Author contributions
W.A., J.v.G., F.V., and E.D. designed and conceptualised the study. T.B., D.M., K.S., and A.K. entered the data into the clinical study database. H.L. and T.B. curated the data after entry. M.K., R.M., and A.Y. performed *Leishmania* positivity testing. H.B. performed the isolation of peripheral mononuclear cells from blood. W.H., A.V.I.M., and J.P. performed the flow cytometry experiments. W.A., I.M., and M.A.D. performed the single-cell RNA and TCR sequencing experiments. N.d.V. and A.M.R. did the formal data analysis. N.d.V. created all figures except for the flow cytometry gating strategies which T.T.P. and J.P. created. N.d.V. and W.A. wrote the manuscript. N.d.V., A.M.R., A.V.I.M., J.P., T.T.P., H.L., M.A.D., B.C., P.M., S.v.H., J.v.G., and W.A. reviewed and edited the manuscript. F.V., S.v.H., H.B., K.R., J.v.G., R.M., E.D., and W.A. coordinated the clinical study. B.C., K.L., P.M., and W.A. supervised the data analysis and manuscript preparation. W.A., J.v.G. and E.D. acquired the funding for the study. All authors have read and agreed to the published version of the manuscript.

## Competing interests
K.L. and P.M. hold shares in ImmuneWatch™, an immunoinformatics company. ImmuneWatch™ had no role in study design, data collection and analysis, decision to publish, or preparation of the manuscript. The other authors declare no conflict of interest.

## Additional information

[1]Clinical Immunology Unit, Department of Clinical Sciences, Institute of Tropical Medicine, 2000 Antwerp, Belgium. [2]Adrem Data Lab, Department of Computer Science, University of Antwerp, 2020 Antwerp, Belgium. [3]Department of Medical Oncology, University Hospital Heidelberg, National Center for Tumor Diseases (NCT) Heidelberg, 69120 Heidelberg, Germany. [4]Department of Microbiology, Aggeu Magalhães Institute—FIOCRUZ/PE, Recife, Brazil. [5]Département de Pathologie et Microbiologie, Faculté de Médecine Vétérinaire, Université de Montréal, Saint-Hyacinthe, QC, Canada. [6]Department of Immunology and Molecular Biology, Faculty of Biomedical Sciences, University of Gondar, Gondar, Ethiopia. [7]Leishmaniasis Research and Treatment Centre, University of Gondar, Gondar, Ethiopia. [8]Molecular Parasitology Unit, Department of Biomedical Sciences, Institute of Tropical Medicine, 2000 Antwerp, Belgium. [9]Clinical Trial Unit, Department of Clinical Sciences, Institute of Tropical Medicine, 2000 Antwerp, Belgium. [10]National Centre for Epidemiology and Population Health, The Australian National University, Canberra 2601, Australia. [11]The Kirby Institute, University of New South Wales, Sydney 2052, Australia. [12]Unit of Neglected Tropical Diseases, Department of Clinical Sciences, Institute of Tropical Medicine, 2000 Antwerp, Belgium. [13]Médecins Sans Frontières, Abdurafi, Ethiopia. [14]Médecins Sans Frontières, Amsterdam, The Netherlands. ✉e-mail: wadriaensen@itg.be

