## [Peer Review File · Communications Biology]

Reviewers' comments:

Reviewer #1 (Remarks to the Author):

The study by de Vrij et al. followed cohorts of HIV and VL-HIV patients in Ethiopia and measured T cell intracellular cytokine production in response to polyclonal activation, as well as single cell RNA and T cell receptor sequencing. Their results show that the majority of VL-HIV patients in their study had a previous history of VL and that these individuals are characterized by higher proportions of PD1+ and TIGIT+ CD4+ T cells and PD-1+ CD8+ T cells; by marked CD4+ and CD8+ T cell unresponsiveness at the single-cell transcriptional level; as well as by an absence of lymphoproliferative response at the clonotype level. They also showed that in VL-HIV individuals who relapsed, there was lower percentages of CD4+ T cells producing IFN- γ in response to polyclonal activation.

The paper is unnecessary long, the data are poorly organised and some of the conclusions are not supported by the results. The number of patients in one of the cohorts (primary VL-HIV) is very low (n=7), and for some of the time points, only two patients were analysed. The description of the cohorts, particularly the VL-HIV cohorts, is completely inadequate. The definition of "post-asymptomatic" is dubious as this characterized successfully treated VL patients, i.e. HIV patients who had VL and were treated with anti-leishmanial treatment. Furthermore, in the primary VL-HIV cohort, the authors use two distinct groups of patients: VL-HIV individuals who already had 1 episode of VL, and those who have had no previous VL, but tested positive by serological tests or PCR. Several studies, some published by the authors of this paper, have shown that a previous VL history in HIV patients is a risk factor for further relapse. The authors also chose those individuals from the primary VL-HIV group who had no VL relapses; and incidentally, those with some of the highest number of previous relapses in the chronic VL/HIV group, to perform their single cell RNA sequencing.

1. Line 44: PD1 and TIGIT are one of many markers of T cell exhaustion and anergy. T cells cannot be characterized as anergic or exhausted using only one marker, and indeed, PD-1 can be expressed by activated T cells. The abstract needs to be rephrased.
2. Introduction: our knowledge about the mechanisms accounting for VL relapses in VL-HIV individuals is still limited, however, quite a few studies in Africa and in Brazil have detailed some of the potential mechanisms. This is poorly described in the introduction and needs to be developed further. The introduction is in general quite vague, with little detail on what has already been published in HIV and VL-HIV patients.
3. Line 77: the authors use a reference that mainly describes Th1 responses in experimental models, but not in human, a more appropriate reference needs to be used in the context of a human study.
4. Line 84: what is the rationale for analysing the role of NK cells?
5. Line 90: a study was published in 2021 that identified markers that can be used to predict VL relapse, this is not cited in the introduction
6. Line 112: study population. The authors use indiscriminately VL/HIV individuals who already had 1 episode of VL, and those who didn't, but tested positive by serological tests or PCR, even though a previous history of VL in VL/HIV patients is a known risk factor for further relapse. Unless the authors can

show compelling evidence that these two groups are immunologically comparable, they should be treated as separate groups.

7. Lines 114-115: "HIV patients were followed up between October 2017 and November 2020". This is for a period of 37 months. However, the timeline in Fig 1B shows patients followed from 20 months before to 30 months after they developed VL. This doesn't make sense.

8. Lines 119-120: There are discrepancies between the timeline showed in Fig 1B and the text on line 119-120: "visits at 6 and 12 months after the end of VL treatment". However, the timeline in Fig 1B shows a follow-up for up to 30 months.

9. Line 133: 4-6 hours is a long time for whole blood or purified PBMC to stay on the bench, especially if this was done in Aburafi, where the temperatures are high. Did the authors test the impact of this long wait on survival and activation markers? There is an unusually high number of dead cells (Figure S3). If the PBMCs were purified in Aburafi, did the author have a -80oC freezer to keep the cells frozen, what about liquid nitrogen?

10. Figure 1A: the authors have 2 patients who had a prior episode of VL "not within 10 years" (blue) and 2 "within 10 years" (red). How can the authors justify that these patients are immunologically different?

11. Figure 1A: The blue group is considered as "asymptomatic", whereas patients have had VL or have never had VL in the past. How can the authors justify that these patients are immunologically different? It looks like an attempt to increase the number of individuals the AL-HIV group from 5 to 7.

12. Line 158: why did the authors use a polyclonal activation, and not an antigen-specific stimulation? The latter will only provide information about the maximal capacity of a given cells to produce a cytokine.

13. How many PBMC were stimulated and how many events were acquired?

14. Table 1: The area where the study took place has been shown to have many seasonal labour migrants. Are the participants of this study migrant workers or individuals living in the endemic areas around Aburafi?

15. Table 1: The authors are following an HIV+ cohort, but not all individuals are on ART? Why? For the 2 individuals not on ART, did they start ART? What ART drugs were used? Were the HIV patients compliant? If so, how was this assessed?

16. Table 1: What drugs were used to treat the VL patients?

17. Table 2: What are the concomitant diseases affecting these individuals?

18. CD4 were not measured at EoT?

19. Lines 225: this had already been shown previously by several groups, included some of the authors of this paper, these tables and results should be presented as supplementary data

20. Line 238: "As expected..." why is it expected? Please explain?

21. Lines 238-239: "the majority of HIV patients were female (57.9%) and did not work in high-risk occupations (52.6)." There is no mention of female study participants in your methodology and suddenly you talk about results from female study participants. This is misleading and confusing.

22. Lines 240-244: the definitions of asymptomatic are unsubstantiated, as being cured for >3years will not guarantee that these individuals will not relapse; nor that the immunological responses of "true" asymptomatic is similar to those of long-term cured (previous history of VL versus serologically positive but with no previous history of VL).

23. The authors describe the activation status of CD4+ T cells, but never used a CD4 marker, these cells

need to be referred to as CD3+ CD8- cells in the results section.

24. The authors mentioned intracellular IL-10, but there are no results.

25. Figure S1: the lymphocyte gate is across 2 populations (likely the monocytes and lymphocytes) but includes half of the lymphocyte population, this is unacceptable.

26. Using FOXP3 alone is definitively NOT enough to characterize Treg.

27. Line 134-135. It's not made at all clear how these "representative" patients were chosen. Given the tiny number of cases chosen for scRNA-seq work, the process for choosing these patients seems absolutely critical in making any claim about a broader population from these analyses.

28. Line 181-183 - Nothing is said about how the QC and filtering criteria were chosen.

29. Line 187-189 - The description of cell cluster annotation seems very inadequate - both singleR and scGate allows the use of different reference sets (singleR) and a choice of predefined or custom gating models (scGate). It should be clarified exactly what was done - presumably using some pre-computed human data in each case? We are also told nothing about how results from the two approaches were combined to produce a single annotation.

30. Line 193. Both the log-fold change and the significance cut-off employed seem very surprisingly loose. The log-2 fold change representing only a 20% change in expression. Similarly, the BH procedure doesn't actually control the type I error rate (and so probably shouldn't be called a p-value, but a false discovery rate) but this implies that 5% of the findings are expected to be false positives. Given the very small number of patients involved here, I have concerns about how likely the gene expression changes identified are to be generalisable.

31. Throughout the section on lines 298-315: nothing is said about the magnitude or statistical significance of differences in cell composition, and the methods section also tell us nothing about the statistical procedures used to support the statements made here.

32. Lines 369-370 "The proportion of Th17 did not differ between pVL-HIV and cVL-HIV patients at any timepoint (Fig. 2D-E)." Fig. 2D-E shows CD8+ T cells not Th17.

33. Line 421. It's not at all clear why a 'lack of responsiveness' would result in an actual REDUCTION of these markers with treatment, rather than just the lack of an effect. That seems unexpected and really needs some explanation.

34. Section lines 402-437. As far as we are told, (line 195) the top 12 enriched pathways were considered, and nothing is said about how 'enriched' these pathways need to be in differentially expressed genes before they are reported. We are told nothing about the numbers of genes and the magnitude of changes in pathways reported here, so it's impossible to evaluate these results: there is a basic lack of detail about the results here, and just showing some heatmaps does not give enough information.

35. Line 435: This problem is particularly acute when we need to evaluate claims like this: "While this could indicate that treatment has few consistent effects on the expression of genes of these patients, this could also be partly explained by the observed depletion of CD16+ monocytes within this patient group" - as it's impossible to work out from the results presented here whether this apparent difference between groups is likely to be a genuine difference, or due to differences in the power to detect expression changes between the groups.

36. Line 438 - it's not clear what the clonotype expansion analysis is intended to show - or precisely what the claim is - beyond the fact that there is less increase in CD4+ and CD8+ T-cells following

treatment in the cVL group than pVL group, which was already observed (e.g. fig 2). From the figures, it looks like all clonotypes expand, but to a lesser extent in chronic than primary patients. That would just seem to be the expectation, given that we know cell numbers recover less in chronic patients. Are the authors claiming that the dynamics of the most common clonotypes are different to others? If so, that should be clarified, and some kind of statistical analysis would seem warranted.

Reviewer #2 (Remarks to the Author):

Vrij et al.'s focus on chronic visceral leishmaniasis (VL) in HIV-infected populations sheds light on an under-represented population with an unmet need for better treatment and predictive immunological markers. Vrij et al's longitudinal cohort of people with coinfection of VL and HIV allows for comparison of those with asymptomatic and chronic VL to better understand immunological determinants of chronic VL. Starting with basic cellular composition of PBMCs and moving logically through exhaustion/senescence markers, transcriptional analysis and T cell clonotypes of CD4+ and CD8+ T cells Vrij et al claims that loss of CD4+ T cells does not predict chronicity but that chronicity is linked with exhausted CD8+ T cells and anergic CD4+ T cells. The human populations studied are well described in the tables, the progression of the story makes logical sense and the authors tie the conclusions back to populations suffering from coinfection/how treatment could be improved. Methods section has sufficient detail to repeat experiments if needed.

A few comments:

1. Consider the use of 'people first' language, e.g. saying 'participant/individual' not 'patient' to help reduce stigma around those living with HIV
2. The plots as seen in, for example, fig 2a/d/g would be easier to understand if simplified (e.g. no need to show dots, box AND zig-zag plot) and include in key reminder of what AL, VL etc. mean.
3. NK cells are mentioned briefly in introduction and figure 2 but are not followed up for further analysis - were these cells looked at for any of the other readouts?
4. Loss of CD14+ monocytes is touched upon in figure 3 and in regard to figure S9 - is there a rationale for their loss in cVL-HIV?
5. The IFN γ -producing CD4+ T cells analysis only shows a difference between those VL-HIV +/- episode-free survival on follow up (rather than pVL-HIV vs. cVL-HIV as analysed for the remainder of the manuscript). Was this +/- episode free survival stratification used elsewhere in the manuscript? Or is there any other functional analysis of pVL-HIV vs. cVL-HIV beyond IFN γ for CD4+ T cells?
6. Main concern: Was any analysis done of multi-exhaustion markers on the same cell (e.g. PD1+TIGIT+)? There is concern that conclusions drawn about exhaustion status are based on expression of single

markers that can also be used as markers for activation, e.g. PD-1. Even on ART, PLWH can have higher T cell activation than HIV-naive individuals and no HIV- cohorts are used here as controls. Additionally, some markers, like TIGIT, have conflicting evidence suggesting their expression is actually linked to higher function (Blazkova 2021). Co-expression analyses and further functional analyses of the samples would be needed to substantiate their claims about exhaustion.

Reviewer #3 (Remarks to the Author):

Vrij et al characterised the compositional and functional changes in peripheral immune cell subsets, in primary and chronic Visceral leishmaniasis HIV patients. They observed persistent T cell exhaustion in chronic VL-HIV cases on ART and state that they identified early markers of CD8+ T cell exhaustion and CD4+ T cell anergy that may significantly help clinical research as surrogate endpoints. While this body of work is of interest for translational utility, this manuscript requires some work to improve clarity for the reader and also strengthening of the data provided.

Comments-

- 1- Figures 2,4 and 5 are quite busy and difficult to interpret. The authors should show longitudinal data of pVL_HIV vs cVL-HIV from the same timepoints and statically compare the data at D0, W1, W2, EOT then M3-6 and M9-12 as samples we taken form these timepoints. The temporal data can be moves to supplemental data.
2. Why is the data in figure 4 labelled CD8-? Are these Cd3+CD4+ T cells or are they CD3+ CD8-? If the later is the case then these cells will be contaminated with CD3-8-4- innate like cells (NKT and gamma delta cells) and this will change the interpretation of the data.
3. Functional data with interferon gamma expression in CD4 and CD8 T cells should be included as a part of the main figures as this confirms the exhausted/anergic state of T cells during chronic VL.
4. What is the HIV viral load of these patient groups? Do they differ between primary and chronic VL?
5. From the TCR data, were the authors able to annotate VL specific TCRs from the scRNAseq data? It would be of interest if these VL specific T cells are more exhausted or anergic compared to VL-specific clones from Primary VL patients. This will further strengthen the data.

Reviewer comments

Reviewer #1 (Remarks to the Author):

The study by de Vrij et al. followed cohorts of HIV and VL-HIV patients in Ethiopia and measured T cell intracellular cytokine production in response to polyclonal activation, as well as single cell RNA and T cell receptor sequencing. Their results show that the majority of VL-HIV patients in their study had a previous history of VL and that these individuals are characterized by higher proportions of PD1+ and TIGIT+ CD4+ T cells and PD-1+ CD8+ T cells; by marked CD4+ and CD8+ T cell unresponsiveness at the single-cell transcriptional level; as well as by an absence of lymphoproliferative response at the clonotype level. They also showed that in VL-HIV individuals who relapsed, there was lower percentages of CD4+ T cells producing IFN- γ in response to polyclonal activation. The paper is unnecessary long, the data are poorly organised and some of the conclusions are not supported by the results.

We thank the reviewer for the detailed review and suggestions. We have now significantly focused the paper on T cells only, reformulated conclusions, and have reduced the length of the manuscript to make it as concise as possible. We hope these adaptations (more explained below) have led to better organized data and clarified anything that was not clear in the original manuscript.

The number of patients in one of the cohorts (primary VL-HIV) is very low (n=7), and for some of the time points, only two patients were analysed.

We like to point out that given that this was a longitudinal study in a hard-to-reach population (e.g. it can take patients several days to travel from their home to reach the health centre), follow-up study visits were often missed (even more so due to the COVID-19 pandemic near the end of the study). This restriction in available timepoints only affected the longitudinal analyses, in which an appropriate statistical analysis method was applied (i.e. linear mixed effect models) that takes these missing values into account.

The description of the cohorts, particularly the VL-HIV cohorts, is completely inadequate. The definition of "post-asymptomatic" is dubious as this characterized successfully treated VL patients, i.e. HIV patients who had VL and were treated with anti-leishmanial treatment. Furthermore, in the primary VL-HIV cohort, the authors use two distinct groups of patients: VL-HIV individuals who already had 1 episode of VL, and those who have had no previous VL, but tested positive by serological tests or PCR. Several studies, some published by the authors of this paper, have shown that a previous VL history in HIV patients is a risk factor for further relapse.

We have removed the 'post-asymptomatic' term which can indeed be dubiously interpreted. The description of the cohorts has now been better clarified, and the 'primary' nomenclature has been changed to 'non-chronic' to account for the two patients with a VL episode that occurred more than 10 years ago. We agree that VL history is a risk factor for relapse, although mostly for subsequent VL relapse within 1-2 years, but argue that the VL episodes that occurred more than 10 years ago in these two patients did not jeopardize our objective (i.e. compare immune state between chronic and non-chronic VL-HIV patients), and likely suggests the presence of a new *Leishmania* infection rather than a recurrent VL episode. We defined chronic patients as those who undergo frequent recurrent episodes, while non-chronic patients were patients that were able to limit the infection (like the 2 patient with past VL >10 years ago) or were primarily infected.

The authors also chose those individuals from the primary VL-HIV group who had no VL relapses; and incidentally, those with some of the highest number of previous relapses in the chronic VL/HIV group, to perform their single cell RNA sequencing.

We would like to clarify that these individuals were indeed deliberately chosen as they were the most representative cases of the non-chronic VL-HIV and chronic VL-HIV groups, and therefore most interesting to conduct the more in-depth analyses on, in order to better define immunological signatures of chronic VL-HIV.

1. Line 44: PD1 and TIGIT are one of many markers of T cell exhaustion and anergy. T cells cannot be characterized as anergic or exhausted using only one marker, and indeed, PD-1 can be expressed by activated T cells. The abstract needs to be rephrased.

We agree with the reviewer's comment and have better structured the paper to build up the evidence for this statement, and during revisions we have more carefully worded any other claims in the manuscript accordingly. For example, we now describe PD1 and TIGIT rather as exhaustion-associated markers, or we just simply mention higher levels of PD1 and TIGIT without categorizing these as associated with any functionality or lack thereof.

2. Introduction: our knowledge about the mechanisms accounting for VL relapses in VL-HIV individuals is still limited, however, quite a few studies in Africa and in Brazil have detailed some of the potential mechanisms. This is poorly described in the introduction and needs to be developed further. The introduction is in general quite vague, with little detail on what has already been published in HIV and VL-HIV patients.

Potential underlying mechanisms for VL relapses were deliberately only briefly listed in the introduction with references provided for broader scoping, to keep the paper focused (and shorter) on our key messages that are not directly related to any mechanisms. It is not our intention to provide a comprehensive overview of all potential general mechanisms for VL relapse.

3. Line 77: the authors use a reference that mainly describes Th1 responses in experimental models, but not in human, a more appropriate reference needs to be used in the context of a human study.

We thank the reviewer for pointing this out, but we do not know which specific reference the reviewer means. In the sentence "A Th2 polarised immune response is permissive for parasite replication, while protective immunity to VL has been linked to a CD4+ T helper 1 (Th1) polarised immune response with high interferon gamma (IFN- γ) production 24-26.", we listed three different references of which two (24-25) mention Th1 responses in human VL disease. We agree with the reviewer that reference 26 is not specific enough, and have removed this reference from the manuscript. Instead, we have included a new reference instead that specifically describes Th1 responses in human VL: Khalil, E.A., Ayed, N.B., Musa, A.M., Ibrahim, M.E., Mukhtar, M.M., Zijlstra, E.E., Elhassan, I.M., Smith, P.G., Kieny, P.M., Ghalib, H.W., Zicker, F., Modabber, F. & Elhassan, A.M. Dichotomy of protective cellular immune responses to human visceral leishmaniasis. Clin Exp Immunol 140, 349-353 (2005).

4. Line 84: what is the rationale for analysing the role of NK cells?

To improve the focus of the paper, we have deleted all paragraphs on NK cells.

5. Line 90: a study was published in 2021 that identified markers that can be used to predict VL relapse, this is not cited in the introduction.

We thank the reviewer for this comment, however no specific DOI or reference was cited by the reviewer for the proposed study. We agree that several papers were published in 2021 on potential markers to predict VL relapse in immunocompetent (qPCR, etc.) and immunocompromised individuals. If the reviewer is willing to provide the reference to the suggested manuscript, we will consider to cite it in an appropriate section.

6. Line 112: study population. The authors use indiscriminately VL/HIV individuals who already had 1 episode of VL, and those who didn't, but tested positive by serological tests or PCR, even though a previous history of VL in VL/HIV patients is a known risk factor for further relapse. Unless the authors can show compelling evidence that these two groups are immunologically comparable, they should be treated as separate groups.

As outlined above, the description of the cohorts has been clarified and group names were changed accordingly to clarify the goal of the comparison between chronic and non-chronic VL-HIV patients.

7. Lines 114-115: "HIV patients were followed up between October 2017 and November 2020". This is for a period of 37 months. However, the timeline in Fig 1B shows patients followed from 20 months before to 30 months after they developed VL. This doesn't make sense.

Not all patients were followed 20 months before and 30 months after they developed VL (see Fig 1B). The plot axis range from a maximum of 20 months before to a maximum of 30 months after the patients developed VL (D0). However, these patients were recruited at different times between October 2017 and November 2020, developed VL at different times during this period, and were followed-up after disease development for different periods of time. The individual lines of a patient display the follow-up period length, and as evidenced by plot Fig 1B the maximum follow-up time of a patient was around 33 months. To clarify this, we have included an additional sentence in the Figure legend: "The depicted months of follow-up were calculated backwards and onwards from the time of active VL development (D0) to standardize timepoints between patients".

8. Lines 119-120: There are discrepancies between the timeline showed in Fig 1B and the text on line 119-120: "visits at 6 and 12 months after the end of VL treatment". However, the timeline in Fig 1B shows a follow-up for up to 30 months.

The full sentence in lines 119-120 states "This follow-up phase lasted until study end and included visits at 6 and 12 months after the end of VL treatment, and at any suspected VL disease episode." Any follow-up after the 12 months was therefore due to a suspected VL disease episode (represented as unscheduled visits or VL relapse visits).

9. Line 133: 4-6 hours is a long time for whole blood or purified PBMC to stay on the bench, especially if this was done in Abdurafi, where the temperatures are high. Did the authors test the impact of this long wait on survival and activation markers? There is an unusually high number of dead cells (Figure S3).

If the PBMCs were purified in Abdurafi, did the author have a -80oC freezer to keep the cells frozen, what about liquid nitrogen?

We would like to clarify that we followed a "standardized operating procedure" (SOP) for the isolation of PBMCs from whole blood that follows those used globally. In this SOP it states that collected venous blood can be kept for a maximum of 4h on room temperature i.e. 20-25°C before processing. The PBMC isolation itself takes a maximum of 2 hours, and isolated venous blood samples were kept on the bench in the air conditioning-controlled laboratory in Abdurafi, where we also performed the

PBMC isolation. After PBMC isolation, samples were kept in a cool cell at -80°C overnight and stored in LN2 the next day, according to cell cryopreservation guidelines. This is commonly done in our other studies in Europe and Africa, and no correlation between this waiting time and viability or T cell activation has been observed in the past.

We agree that the sample shown in Figure S3 had a relatively high number of dead cells for PBMCs as compared to the samples of the other panels. However, in contrast to these other panels, this panel was stimulated with LAC, BD GolgiPlug, and BD GolgiStop. The first panel in Figure S1 was not stimulated, and the second panel in Figure S2 was stimulated with LAC and BD GolgiPlug only. We expect this to be the reason for the relatively lower viability of this panel. Nonetheless, we now set a viability threshold of >25%, and we further analysed whether any of the markers were impacted by the lower viability and there does not seem to be a correlation between viability and the expression of the markers included in our study.

10. Figure 1A: the authors have 2 patients who had a prior episode of VL "not within 10 years" (blue) and 2 "within 10 years" (red). How can the authors justify that these patients are immunologically different?

The prior episodes of the two patients in the "non-chronic VL-HIV" cohort who had a prior episode of VL "not within 10 years" (depicted in blue with a "+" in Figure 1B) took place in 1999 and 2007. These prior episodes occurred 227 and 143 months, respectively, before the patient's active VL development within this study (red diamond symbol).

Meanwhile, the prior episodes of the four patients in the "chronic VL-HIV" cohort who had one prior episode "within 10 years" (depicted in red with "1" for the number of prior episodes) all took place between 2017 and 2019. For these 4 "chronic VL-HIV" patients, the prior episodes occurred 4, 7, 10 or 15 months before the patient's active VL development within this study (red diamond symbol).

We believe that patients with the prior episodes of 4, 7, 10 or 15 months before the patient's active VL development differ in disease course compared to the patients with the prior episodes of 227 and 143 months before, and are therefore included in the "chronic VL-HIV" cohort to study whether there are indeed immunological differences.

11. Figure 1A: The blue group is considered as "asymptomatic", whereas patients have had VL or have never had VL in the past. How can the authors justify that these patients are immunologically different? It looks like an attempt to increase the number of individuals the AL-HIV group from 5 to 7.

We assume the reviewer intended to refer to the former "primary VL-HIV" group (n = 7) and the 2 patients with past VL >10 years ago, and not the asymptomatic group (AL-HIV) that consists of 20 patients. The "non-chronic VL-HIV" group (formerly known as "primary VL-HIV") is not considered "asymptomatic" but is expected to be immunologically different from the "chronic VL-HIV" group.

12. Line 158: why did the authors use a polyclonal activation, and not an antigen-specific stimulation? The latter will only provide information about the maximal capacity of a given cells to produce a cytokine.

We hypothesized that there would be general anergy in the T cells of VL-HIV patients, warranting the use of a polyclonal activation to see whether the T cells of VL-HIV were indeed no longer capable of sufficient cytokine production. As such, we wanted to evaluate the maximum capacity of the cells to

produce a given cytokine, and get a general idea of the full host immune state in a complex co-infection rather than evaluating only a *Leishmania*-antigen specific response, given that there may also be a potential synergistic role of HIV. Moreover, as there is no clarity yet on immunodominant antigens for *Leishmania*, an antigen-specific stimulation was not preferred.

3. How many PBMC were stimulated and how many events were acquired?

Between 25.000 and 100.000 events were acquired in the flow cytometry experiments. The number of cells in the single-cell experiments is mentioned in the manuscript.

14. Table 1: The area where the study took place has been shown to have many seasonal labour migrants. Are the participants of this study migrant workers or individuals living in the endemic areas around Adburafi?

The majority (93.8%) of the participants of this study were individuals living in the endemic areas around Adburafi. Two participants were migrant workers from Gondar, Amhara, and another two participants were migrant workers from Humera, Tigray.

15. Table 1: The authors are following an HIV+ cohort, but not all individuals are on ART? Why? For the 2 individuals not on ART, did they start ART? What ART drugs were used? Were the HIV patients compliant? If so, how was this assessed?

As evidenced in Table 1 and in accordance with the treat-all strategy currently applied in Ethiopia, all patients but one (62 out of 63 participants) were on ART at the start of the study. For this one patient, ART was initiated at the recruitment visit. Patient ART regimen information has been added to Table 1. As described in the results section, all patients were ART compliant. Compliance was assessed by self-reporting at each visit (i.e. by questioning the average number of missed doses per month).

16. Table 1: What drugs were used to treat the VL patients?

VL patients are treated with the combination therapy with AmbiSome and Miltefosine, or AmbiSome and Sodium Stibogluconate. These details were included in Table 1 which delves deeper into the characteristics of the VL-HIV patients.

17. Table 2: What are the concomitant diseases affecting these individuals?

These were amebiasis, typhoid fever, pneumonia, and pulmonary tuberculosis. More specifically, there were three cases of typhoid fever and one case of amebiasis in the chronic VL-HIV group, and one case of pneumonia and one case of pulmonary tuberculosis in the non-chronic VL-HIV group. This is now mentioned in the table.

18. CD4 were not measured at EoT?

CD4 counts were routinely measured every 6 months, and additional time points were unfortunately not included in the study protocol.

19. Lines 225: this had already been shown previously by several groups, included some of the authors of this paper, these tables and results should be presented as supplementary data.

We agree that this validates previous findings and have, as recommended, moved these details to Supplementary Table 1. We have decided to maintain Table 2 in the main manuscript as it describes the characteristics of our specific study groups with the main comparison "chronic vs non-chronic VL-HIV". We find that this placement is crucial for the further interpretation of results.

20. Line 238: "As expected..." why is it expected? Please explain?

We thank the reviewer for pointing out this misplaced piece of a sentence. This was originally supposed to be in front of the previous sentence. We have revised the manuscript accordingly by swapping this between sentences. It is now:

"As expected in the Ethiopian context, all AL-HIV and VL-HIV participants were male, and almost all of them (88.6%) were working as farmers or daily labourers, occupations shown to have high risk of Leishmania transmission (Table S1) 40. Although matching in most variables, the majority of our endemic HIV control group were female (57.9%) and did not work in high-risk occupations (52.6%) (Table S1)."

21. Lines 238-239: "the majority of HIV patients were female (57.9%) and did not work in high-risk occupations (52.6)." There is no mention of female study participants in your methodology and suddenly you talk about results from female study participants. This is misleading and confusing.

We recruited patients in a biological sex-agnostic manner, and these are hence referred to "participants" in general in the methods section

22. Lines 240-244: the definitions of asymptomatic are unsubstantiated, as being cured for >3years will not guarantee that these individuals will not relapse; nor that the immunological responses of "true" asymptomatic is similar to those of long-term cured (previous history of VL versus serologically positive but with no previous history of VL).

We thank the reviewer for noticing this. Asymptomatic infection in HIV patients was defined in this study as followed: having positivity on any *Leishmania* marker (rK39 RDT and/or ELISA, DAT, KAtex, and/or PCR) and no VL disease development throughout our study duration. This forms an interesting control group (AL-HIV, independent of prior VL history) for our overt VL (both chronic and non-chronic VL-HIV) study cohort. We agree with the reviewer and have not made any claims on the similarity in immunological response between those with or without previous history of VL. We simply remarked on the fact that 'none of the AL-HIV participants developed overt VL during the study duration' and that this thus would be an interesting control group to compare against. Nonetheless, we fully agree with the reviewer that a comparison between 'true' asymptomatics versus long-term cured (prior history of VL long before the study) would be very interesting, but we think this is not within the scope of this paper (focused on VL chronicity in HIV co-infection), as our manuscript is already lengthy and complex enough. This separate research question will rather be incorporated in a separate study specific for this question, Including additional experiments and analyses only performed for these two groups.

23. The authors describe the activation status of CD4+ T cells, but never used a CD4 marker, these cells need to be referred to as CD3+ CD8- cells in the results section.

This is a valid point made by the reviewer and we have revised accordingly.

24. The authors mentioned intracellular IL-10, but there are no results.

We thank the reviewer for this observation and have removed any mentioning of IL-10 in the manuscript.

25. Figure S1: the lymphocyte gate is across 2 populations (likely the monocytes and lymphocytes) but includes half of the lymphocyte population, this is unacceptable.

We thank the reviewer for this observation. This was a gating error that affected a small number of samples in that panel. We have updated all the gatings for every sample and have reanalysed and readjusted in the manuscript accordingly. We have also included new gating strategy figures.

26. Using FOXP3 alone is definitively NOT enough to characterize Treg.

We thank the reviewer for their comment, and we have now opted to remove any mention of FOXP3⁺ CD4⁺ T cells or Tregs to improve the focus of the manuscript, and to shorten the paper in accordance with the reviewer's previous suggestions.

27. Line 134-135. It's not made at all clear how these "representative" patients were chosen. Given the tiny number of cases chosen for scRNA-seq work, the process for choosing these patients seems absolutely critical in making any claim about a broader population from these analyses.

Based on cell viability and patient characteristics, we have chosen the most representative cases to study the differences of each group with scRNA-seq. The characteristics of the selected patients are described in Supplementary Table 2.

28. Line 181-183 - Nothing is said about how the QC and filtering criteria were chosen.

The QC parameters were chosen based on visual inspection due to a lack of standardized guidelines. As scRNA-seq is still a relatively new method, QC and filtering criteria are still a remaining challenge in scRNA-seq and lack standardization. However, we followed the recommended approach of visual inspection and outlier removal (Luecken and Theis, 2019, <https://doi.org/10.15252/msb.20188746>).

29. Line 187-189 - The description of cell cluster annotation seems very inadequate - both singleR and scGate allows the use of different reference sets (singleR) and a choice of predefined or custom gating models (scGate). It should be clarified exactly what was done - presumably using some pre-computed human data in each case? We are also told nothing about how results from the two approaches were combined to produce a single annotation.

We thank the reviewer for these valid points. To clarify, we have used the Human Primary Cell Atlas reference from the *celldex* package for SingleR, and the predefined gating models of scGate. The ensemble approach consists specifically of first using the predefined gating models of scGate for several well-known populations, and afterwards executing SingleR for validation. SingleR was then used to annotate any remaining clusters for cell populations that were not in the predefined scGate models. However, all annotations were manually curated based on between-cluster differentially expressed genes. This is an approach that is currently recommended by leading experts (Heumos et al., 2023). We have revised the manuscript with this explanation, accordingly.

30. Line 193. Both the log-fold change and the significance cut-off employed seem very surprisingly loose. The log-2 fold change representing only a 20% change in expression. Similarly, the BH procedure doesn't actually control the type I error rate (and so probably shouldn't be called a p-value, but a false discovery rate) but this implies that 5% of the findings are expected to be false positives. Given the very small number of patients involved here, I have concerns about how likely the gene expression changes identified are to be generalisable.

We thank the reviewer for their comment. The chosen log2 fold change is a Seurat default, and it is indeed somewhat inclusive. We chose the default parameter due to this small number of patients. Instead of focusing on identifying specific genes that could be associated with the disease, we wanted to investigate different pathway trends and profiles in an unbiased manner, resulting in a more inclusive approach of DEGs. We like to note that increasing the log2 fold change to 0.58 for an

estimated 50% change in expression, resulted in a similar gene expression pattern (see figure below) as the 0.25 threshold in the manuscript.

31. Throughout the section on lines 298-315: nothing is said about the magnitude or statistical significance of differences in cell composition, and the methods section also tell us nothing about the statistical procedures used to support the statements made here.

For our scRNA-seq data, we have not included statistical testing due to the low number of patients per study group. We have demonstrated the scRNA-seq trends and differences in a visual manner. As such, no hard conclusions are made and the data is further discussed in the discussion.

32. Lines 369-370 "The proportion of Th17 did not differ between pVL-HIV and cVL-HIV patients at any timepoint (Fig. 2D-E)." Fig. 2D-E shows CD8+ T cells not Th17.

We thank the reviewer for their observation. This sentence referred to former Fig. S5D-E. However, in the newly revised manuscript we have opted to remove any mentioning of Th17 to improve the focus of the manuscript, and shorten the manuscript.

33. Line 421. Its not at all clear why a 'lack of responsiveness' would result in an actual REDUCTION of these markers with treatment, rather than just the lack of an effect. That seems unexpected and really needs some explanation.

We agree with the reviewer and have removed this sentence and conclusion.

34. Section lines 402-437. As far as we are told, (line 195) the top 12 enriched pathways were considered, and nothing is said about how 'enriched' these pathways need to be in differentially expressed genes before they are reported. We are told nothing about the numbers of genes and the magnitude of changes in pathways reported here, so it's impossible to evaluate these results: there

is a basic lack of detail about the results here, and just showing some heatmaps does not give enough information.

We agree with the reviewer and have added the pathway enrichment scores and number of genes in supplementary figure 11 to provide more information to evaluate the results with.

35. Line 435: This problem is particularly acute when we need to evaluate claims like this: "While this could indicate that treatment has few consistent effects on the expression of genes of these patients, this could also be partly explained by the observed depletion of CD16+ monocytes within this patient group" - as its impossible to work out from the results presented here whether this apparent difference between groups is likely to be a genuine difference, or due to differences in the power to detect expression changes between the groups.

We have deleted the monocytes and NK cell data in order to focus the manuscript on T cells, and have altered the discussion accordingly while taking note of the reviewer's comments.

36. Line 438 – it's not clear what the clonotype expansion analysis is intended to show - or precisely what the claim is - beyond the fact that there is less increase in CD4+ and CD8+ T-cells following treatment in the cVL group than pVL group, which was already observed (e.g. fig 2). From the figures, it looks like all clonotypes expand, but to a lesser extent in chronic than primary patients. That would just seem to be the expectation, given that we know cell numbers recover less in chronic patients. Are the authors claiming that the dynamics of the most common clonotypes are different to others? If so, that should be clarified, and some kind of statistical analysis would seem warranted.

The clonotype expansion analyses in Figure 8A indicate the expansion/proliferation of the top 30 largest expanded T cell clones (i.e. T cell receptor sequence-based clonotypes, T cells carrying the same TCR) during antileishmanial treatment in non-chronic patients, while this expansion/proliferation is not observed in chronic patients (Fig. 8B). These expanded/proliferated T cell clonotypes are represented on the UMAP in Fig 8C. This is different from Figure 2 where we provide the proportions of all CD8- and CD8+ T cells in the blood compartment. The top expanded T cell clonotypes are most likely to be involved in an active infection due to a shift in dynamics (i.e. expansion or contraction), and we can speculate that these may likely be *Leishmania*-specific T cells. In this figure, we do not see all clonotypes expanding, some contract and disappear, while some expand greatly and become dominant clones. These dominant clones are most likely to be involved in the active infection, and may be important for resolving the infection. For example, in the non-chronic patients, we see several clonotypes expanding to a proportion several folds larger than previously, becoming the dominant clones, while several other clonotypes contract or disappear from the top 30. The total proportion of the top 30 clonotypes of the non-chronic group shifts from around 0.275 to 0.425, nearly doubling in size. In contrast, this total proportion of the top 30 clonotypes of the chronic group shifts from roughly 0.225 to 0.25, showing little to no expansion in response to the active infection. This indicates anergy from stimulation, something we can not show in figure 2. In addition, when overlaying these top 30 T cell clonotypes on the UMAP of the VL-HIV patients (Fig 4C), we can deduct that the expanded T cell clonotypes are CD4+ and CD8+ Tem cells in the non-chronic group, but only CD8+ Tem cells in the chronic group. This highlights CD4+ T cell functional anergy. Due to the limited number of patients per study group for scRNA-seq, we have not included statistical analysis.

Reviewer #2 (Remarks to the Author):

Vrij et al.'s focus on chronic visceral leishmaniasis (VL) in HIV-infected populations sheds light on an under-represented population with an unmet need for better treatment and predictive immunological markers. Vrij et al's longitudinal cohort of people with coinfection of VL and HIV allows for comparison of those with asymptomatic and chronic VL to better understand immunological determinants of chronic VL. Starting with basic cellular composition of PBMCs and moving logically through exhaustion/senescence markers, transcriptional analysis and T cell clonotypes of CD4+ and CD8+ T cells Vrij et al claims that loss of CD4+ T cells does not predict chronicity but that chronicity is linked with exhausted CD8+ T cells and anergic CD4+ T cells. The human populations studied are well described in the tables, the progression of the story makes logical sense and the authors tie the conclusions back to populations suffering from coinfection/how treatment could be improved. Methods section has sufficient detail to repeat experiments if needed.

We thank the reviewer for their recognition of our work .

A few comments:

1. Consider the use of 'people first' language, e.g. saying 'participant/individual' not 'patient' to help reduce stigma around those living with HIV

We thank the reviewer for this valuable feedback, and we have revised this in the manuscript.

2. The plots as seen in, for example, fig 2a/d/g would be easier to understand if simplified (e.g. no need to show dots, box AND zig-zag plot) and include in key reminder of what AL, VL etc. mean.

We thank the reviewer for this feedback, and we have revised this in the manuscript. We now demonstrate the data via jittered dots and a boxplot. We have also clarified the abbreviations of our participant groups in the legend.

3. NK cells are mentioned briefly in introduction and figure 2 but are not followed up for further analysis - were these cells looked at for any of the other readouts?

We thank the reviewer for this question, but in order to improve the focus of the manuscript per the suggestions of other reviewers, we have removed any mention of NK cells from the manuscript.

4. Loss of CD14+ monocytes is touched upon in figure 3 and in regard to figure S9 - is there a rationale for their loss in cVL-HIV?

We do not have a clear-cut rationale for this observation, and we believe more research is needed on this. From the small number of patients, we do not feel comfortable do deduce any hypothesis. In the revised manuscript, we do not touch upon this anymore.

5. The IFNg-producing CD4+ T cells analysis only shows a difference between those VL-HIV +/- episode-free survival on follow up (rather than pVL-HIV vs. cVL-HIV as analysed for the remainder of the manuscript). Was this +/- episode free survival stratification used elsewhere in the manuscript? Or is there any other functional analysis of pVL-HIV vs. cVL-HIV beyond IFNg for CD4+ T cells?

We have removed this episode-free survival stratification analysis entirely in the newly revised manuscript to improve focus and shorten the manuscript, and because we did not fully support this stratification.

6. Main concern: Was any analysis done of multi-exhaustion markers on the same cell (e.g. PD1+TIGIT+)? There is concern that conclusions drawn about exhaustion status are based on expression of single markers that can also be used as markers for activation, e.g. PD-1. Even on ART, PLWH can have higher T cell activation than HIV-naive individuals and no HIV- cohorts are used here as controls. Additionally, some markers, like TIGIT, have conflicting evidence suggesting their expression is actually linked to higher function (Blazkova 2021). Co-expression analyses and further functional analyses of the samples would be needed to substantiate their claims about exhaustion.

We thank the reviewer for their valuable comment and have performed a multi-exhaustion marker analysis, as suggested. T cell exhaustion was assessed by co-expression of PD1+LAG3+, PD1+TIM3+ and TIM3+LAG3 on T cells, all of which remained inconclusive. TIGIT was, unfortunately, not included within the same flow cytometric staining panel as PD1. However, we have assessed the TIGIT+IFN γ - and TIGIT-IFN γ + co-expression levels to explore whether the high level of TIGIT on patient T cells is more associated with T cell activation rather than T cell inhibition, and show marked differences between the chronic and non-chronic VL-HIV patients (Figure 5). In addition, we showed functional exhaustion at the single-cell level and a lack of lymphoproliferative response at the single-cell T cell clonal level, all indicative of T cell exhaustion (Figure 6 and 7).

Reviewer #3 (Remarks to the Author):

Vrij et al characterised the compositional and functional changes in peripheral immune cell subsets, in primary and chronic Visceral leishmaniasis HIV patients. They observed persistent T cell exhaustion in chronic VL-HIV cases on ART and state that they identified early markers of CD8+ T cell exhaustion and CD4+ T cell anergy that may significantly help clinical research as surrogate endpoints. While this body of work is of interest for translational utility, this manuscript requires some work to improve clarity for the reader and also strengthening of the data provided.

We thank the reviewer for their recognition of our work, and the fact that the reviewer thinks there is translational utility in our work. We have incorporated their feedback to improve clarity and strengthen our findings.

Comments-

1- Figures 2,4 and 5 are quite busy and difficult to interpret. The authors should show longitudinal data of pVL_HIV vs cVL-HIV from the same timepoints and statically compare the data at D0, W1, W2, EOT then M3-6 and M9-12 as samples we taken form these timepoints. The temporal data can be moves to supplemental data.

We thank the reviewer for this suggestion. We agree that these figures are quite busy, and therefore difficult to interpret. As such, we have removed the density layer on the figure and show only jittered points and a boxplot. In addition, we have opted to statistically compare the data at D0 and at EOT and performed this consistently throughout the manuscript. Other timepoints were not included for cross-sectional comparisons as there were either a significant amount of missing timepoints, or time points were not comparable due to unscheduled visits outside of the recommended visit window for that time point. To compensate for this study limitation, as well as to be more statistically rigid by analysing longitudinal data in a longitudinal manner and thus limiting the number of tests, we performed longitudinal analyses in order to assess temporal dynamic patterns. In order to make the figures more interpretable, we have also removed the relapse rate comparisons. In summary, we have made the figures less busy and easier to interpret, and have included EOT in a cross-sectional comparison in addition to D0. Other timepoints were instead included for the temporal analysis because this is statistically more rigid.

2. Why is the data in figure 4 labelled CD8-? Are these Cd3+CD4+ T cells or are they CD3+ CD8-? If the later is the case then these cells will be contaminated with CD3-8-4- innate like cells (NKT and gamma delta cells) and this will change the interpretation of the data.

These are indeed CD3+CD8- rather than CD3+CD4+, due to flow cytometry panel limitations. We agree with the reviewer that this cell population will contain contamination of different cell populations. To avoid confusion about this, we have referred to these cells as CD3+CD8- cells in the manuscript, and have put analyses on this cellular subset in the supplementary data, opting to instead focus on CD3+CD8+ T cells on the flow cytometry level.

3. Functional data with interferon gamma expression in CD4 and CD8 T cells should be included as a part of the main figures as this confirms the exhausted/anergic state of T cells during chronic VL.

We agree with the reviewer that this should be included as additional analysis to further confirm or validate the signs of an exhausted/anergic state during chronic VL in HIV co-infected individuals. As

such, we have included this data in the manuscript. More specifically, we have analysed the co-expression of TIGIT+IFN γ - and TIGIT-IFN γ + in CD8+ and CD8- T cell populations. This is now shown in Figure 5. In addition, the single-cell transcriptome level exhibited in Figure 6 also shows this exhausted state at a functional level.

4. What is the HIV viral load of these patient groups? Do they differ between primary and chronic VL?

HIV viral load data has now been included in Table 1 of the manuscript. These do not differ between the 'chronic' VL-HIV and 'non-chronic' VL-HIV (the former 'Primary' group) groups. We should, however, clarify that there is a substantial number of missing viral load data for both groups. While we can probably rule out a large confounding effect of HIV viral load, it remains to be seen whether it is a contributing factor to VL chronicity in HIV co-infected individuals.

5. From the TCR data, were the authors able to annotate VL specific TCRs from the scRNAseq data? It would be of interest if these VL specific T cells are more exhausted or anergic compared to VL-specific clones from Primary VL patients. This will further strengthen the data.

The authors do fully agree that this would be very interesting, as this is something we intend to focus on in our future leishmaniasis work as well. Unfortunately, there are currently only a small number of known T cell epitopes of *Leishmania*, and no known *Leishmania*-specific TCRs. Thus, we are not able to annotate the TCRs with specificity information.

Reviewer #1 (Remarks to the Author):

The authors have still not answered several points satisfactorily, please see our comments in bold italic below:

Reviewer comments

Reviewer #1 (Remarks to the Author):

The study by de Vrij et al. followed cohorts of HIV and VL-HIV patients in Ethiopia and measured T cell intracellular cytokine production in response to polyclonal activation, as well as single cell RNA and T cell receptor sequencing. Their results show that the majority of VL-HIV patients in their study had a previous history of VL and that these individuals are characterized by higher proportions of PD1+ and TIGIT+ CD4+ T cells and PD-1+ CD8+ T cells; by marked CD4+ and CD8+ T cell unresponsiveness at the single-cell transcriptional level; as well as by an absence of lymphoproliferative response at the clonotype level. They also showed that in VL-HIV individuals who relapsed, there was lower percentages of CD4+ T cells producing IFN- γ in response to polyclonal activation. The paper is unnecessary long, the data are poorly organised and some of the conclusions are not supported by the results.

We thank the reviewer for the detailed review and suggestions. We have now significantly focused the paper on T cells only, reformulated conclusions, and have reduced the length of the manuscript to make it as concise as possible. We hope these adaptations (more explained below) have led to better organized data and clarified anything that was not clear in the original manuscript.

The number of patients in one of the cohorts (primary VL-HIV) is very low (n=7), and for some of the time points, only two patients were analysed.

We like to point out that given that this was a longitudinal study in a hard-to-reach population (e.g. it can take patients several days to travel from their home to reach the health centre), follow-up study visits were often missed (even more so due to the COVID-19 pandemic near the end of the study). This restriction in available timepoints only affected the longitudinal analyses, in which an appropriate statistical analysis method was applied (i.e. linear mixed effect models) that takes these missing values into account.

The description of the cohorts, particularly the VL-HIV cohorts, is completely inadequate. The definition of “post-asymptomatic” is dubious as this characterized successfully treated VL patients, i.e. HIV patients who had VL and were treated with anti-leishmanial treatment. Furthermore, in the primary VL-HIV cohort, the authors use two distinct groups of patients: VL-HIV individuals who already had 1 episode of VL, and those who have had no previous VL, but tested positive by serological tests or PCR. Several studies, some published by the authors of this paper, have shown that a previous VL history in HIV patients is a risk factor for further relapse.

We have removed the ‘post-asymptomatic’ term which can indeed be dubiously interpreted. The description of the cohorts has now been better clarified, and the ‘primary’ nomenclature has been

changed to 'non-chronic' to account for the two patients with a VL episode that occurred more than 10 years ago. We agree that VL history is a risk factor for relapse, although mostly for subsequent VL relapse within 1-2 years, but argue that the VL episodes that occurred more than 10 years ago in these two patients did not jeopardize our objective (i.e. compare immune state between chronic and non-chronic VL-HIV patients), and likely suggests the presence of a new *Leishmania* infection rather than a recurrent VL episode. We defined chronic patients as those who undergo frequent recurrent episodes, while non-chronic patients were patients that were able to limit the infection (like the 2 patient with past VL >10 years ago) or were primarily infected.

The 2 patients in question were VL-HIV patients with a previous history of relapse. The authors cannot assess whether these patients were immunologically different from those who had no relapse. Furthermore, there is no evidence for the author to suggest “the presence of a new Leishmania infection”. This seems at best unproven: there is considerable evidence that VL parasites can persist for years following cure, and this is likely to have ongoing impact on immune function (e.g. [https://doi.org/10.1016/S0163-4453\(03\)00002-1](https://doi.org/10.1016/S0163-4453(03)00002-1); doi:10.1016/S0140-6736(86)91725-3). We don't really know how common this is, as most of this literature is based on disease re-emerging in non-endemic locations, and so is necessarily rather anecdotal – it's certainly the case that disease relapse over 1 year after apparent cure is due to the original infection re-emerging (DOI: [10.1128/mBio.00971-21](https://doi.org/10.1128/mBio.00971-21)), so it seems naive to assume this is not the case over longer time periods. There is also evidence that many Leishmania species can reach a dormant state likely connected to this clinical persistence (<https://doi.org/10.1073/pnas.1619265114>; <https://doi.org/10.3389/fcimb.2023.1253033>). So the grouping of even 10+ year previous VL infections is a concern.

Can the authors define “non-chronic” in the text when they use it first (line 98)?

The authors also chose those individuals from the primary VL-HIV group who had no VL relapses; and incidentally, those with some of the highest number of previous relapses in the chronic VL/HIV group, to perform their single cell RNA sequencing.

We would like to clarify that these individuals were indeed deliberately chosen as they were the most representative cases of the non-chronic VL-HIV and chronic VL-HIV groups, and therefore most interesting to conduct the more in-depth analyses on, in order to better define immunological signatures of chronic VL-HIV.

If these individuals were deliberately chosen, it has to be clearly specified in the text.

1. Line 44: PD1 and TIGIT are one of many markers of T cell exhaustion and anergy. T cells cannot be characterized as anergic or exhausted using only one marker, and indeed, PD-1 can be expressed by activated T cells. The abstract needs to be rephrased.

We agree with the reviewer's comment and have better structured the paper to build up the evidence for this statement, and during revisions we have more carefully worded any other claims in the manuscript accordingly. For example, we now describe PD1 and TIGIT rather as exhaustion-associated markers, or we just simply mention higher levels of PD1 and TIGIT without categorizing these as associated with any functionality or lack thereof.

Change exhaustion on line 517 to “levels of exhaustion-associated markers”.

2. Introduction: our knowledge about the mechanisms accounting for VL relapses in VL-HIV individuals is still limited, however, quite a few studies in Africa and in Brazil have detailed some of the potential mechanisms. This is poorly described in the introduction and needs to be developed further. The introduction is in general quite vague, with little detail on what has already been published in HIV and VL-HIV patients.

Potential underlying mechanisms for VL relapses were deliberately only briefly listed in the introduction with references provided for broader scoping, to keep the paper focused (and shorter) on our key messages that are not directly related to any mechanisms. It is not our intention to provide a comprehensive overview of all potential general mechanisms for VL relapse.

The title of this paper is Persistent T cell unresponsiveness associated with chronic visceral leishmaniasis in HIV-coinfected patients” and one of the key messages of this paper is “These findings suggest PD-1 and TIGIT play a pivotal in VL disease chronicity in HIV co-infected patients”. It is therefore appropriate to briefly summarize what is already known about T cell responsiveness in this field.

Can the authors explain why the paper of Wherry et al is cited on line 75? This paper doesn’t address the loss of host immune response during leishmaniasis.

And also complete the sentence: play a pivotal... in VL disease.

3. Line 77: the authors use a reference that mainly describes Th1 responses in experimental models, but not in human, a more appropriate reference needs to be used in the context of a human study.

We thank the reviewer for pointing this out, but we do not know which specific reference the reviewer means. In the sentence “A Th2 polarised immune response is permissive for parasite replication, while protective immunity to VL has been linked to a CD4+ T helper 1 (Th1) polarised immune response with high interferon gamma (IFN- γ) production 24-26.”, we listed three different references of which two (24-25) mention Th1 responses in human VL disease. We agree with the reviewer that reference 26 is not specific enough, and have removed this reference from the manuscript. Instead, we have included a new reference instead that specifically describes Th1 responses in human VL: Khalil, E.A., Ayed, N.B., Musa, A.M., Ibrahim, M.E., Mukhtar, M.M., Zijlstra, E.E., Elhassan, I.M., Smith, P.G., Kieny, P.M., Ghalib, H.W., Zicker, F., Modabber, F. & Elhassan, A.M. Dichotomy of protective cellular immune responses to human visceral leishmaniasis. Clin Exp Immunol 140, 349-353 (2005).

The notion that Th1 is associated with healing was extensively described in experimental model, but has been largely challenged in human VL (e.g. in 10.1016/j.it.2007.07.004 and 10.1128/CVI.00143-12). Please amend the text and your reference accordingly.

4. Line 84: what is the rationale for analysing the role of NK cells?

To improve the focus of the paper, we have deleted all paragraphs on NK cells.

5. Line 90: a study was published in 2021 that identified markers that can be used to predict VL relapse, this is not cited in the introduction.

We thank the reviewer for this comment, however no specific DOI or reference was cited by the reviewer for the proposed study. We agree that several papers were published in 2021 on potential markers to predict VL relapse in immunocompetent (qPCR, etc.) and immunocompromised individuals. If the reviewer is willing to provide the reference to the suggested manuscript, we will consider to cite it in an appropriate section.

There were few papers published in 2021 that discuss potential markers of relapse, and by typing the key words: “prediction markers and visceral leishmaniasis and HIV, there are 3 papers coming up. Unless the authors have issues with accessing PubMed, it should be straight forward to find it.

6. Line 112: study population. The authors use indiscriminately VL/HIV individuals who already had 1 episode of VL, and those who didn't, but tested positive by serological tests or PCR, even though a previous history of VL in VL/HIV patients is a known risk factor for further relapse. Unless the authors can show compelling evidence that these two groups are immunologically comparable, they should be treated as separate groups.

As outlined above, the description of the cohorts has been clarified and group names were changed accordingly to clarify the goal of the comparison between chronic and non-chronic VL-HIV patients.

Please see the comments above, the authors didn't show any evidence that the two groups are immunologically different.

7. Lines 114-115: “HIV patients were followed up between October 2017 and November 2020” . This is for a period of 37 months. However, the timeline in Fig 1B shows patients followed from 20 months before to 30 months after they developed VL. This doesn't make sense.

Not all patients were followed 20 months before and 30 months after they developed VL (see Fig 1B). The plot axis range from a maximum of 20 months before to a maximum of 30 months after the patients developed VL (D0). However, these patients were recruited at different times between October 2017 and November 2020, developed VL at different times during this period, and were followed-up after disease development for different periods of time. The individual lines of a patient display the follow-up period length, and as evidenced by plot Fig 1B the maximum follow-up time of a patient was around 33 months. To clarify this, we have included an additional sentence in the Figure legend: “The depicted months of follow-up were calculated backwards and onwards from the time of active VL development (D0) to standardize timepoints between patients”.

8. Lines 119-120: There are discrepancies between the timeline showed in Fig 1B and the text on line 119-120: “visits at 6 and 12 months after the end of VL treatment” . However, the timeline in Fig 1B shows a follow-up for up to 30 months.

The full sentence in lines 119-120 states “This follow-up phase lasted until study end and included visits at 6 and 12 months after the end of VL treatment, and at any suspected VL disease episode.” Any follow-up after the 12 months was therefore due to a suspected VL disease episode (represented as unscheduled visits or VL relapse visits).

This sentence is still unclear as “unscheduled” visit are not defined. Please rephrase the legend of Figure 1b

9. Line 133: 4-6 hours is a long time for whole blood or purified PBMC to stay on the bench, especially if this was done in Abdurafi, where the temperatures are high. Did the authors test the impact of this long wait on survival and activation markers? There is an unusually high number of dead cells (Figure S3).

If the PBMCs were purified in Abdurafi, did the author have a -80oC freezer to keep the cells frozen, what about liquid nitrogen?

We would like to clarify that we followed a “standardized operating procedure” (SOP) for the isolation of PBMCs from whole blood that follows those used globally. In this SOP it states that collected venous blood can be kept for a maximum of 4h on room temperature i.e. 20-25°C before processing. The PBMC isolation itself takes a maximum of 2 hours, and isolated venous blood samples were kept on the bench in the air conditioning-controlled laboratory in Abdurafi, where we also performed the PBMC isolation. After PBMC isolation, samples were kept in a cool cell at -80°C overnight and stored in LN2 the next day, according to cell cryopreservation guidelines. This is commonly done in our other studies in Europe and Africa, and no correlation between this waiting time and viability or T cell activation has been observed in the past.

We agree that the sample shown in **Figure S3** had a relatively high number of dead cells for PBMCs as compared to the samples of the other panels. However, in contrast to these other panels, this panel was stimulated with LAC, BD GolgiPlug, and BD GolgiStop. The first panel in Figure S1 was not stimulated, and the second panel in Figure S2 was stimulated with LAC and BD GolgiPlug only. We expect this to be the reason for the relatively lower viability of this panel. Nonetheless, we now set a viability threshold of >25%, and we further analysed whether any of the markers were impacted by the lower viability and there does not seem to be a correlation between viability and the expression of the markers included in our study.

There is no dotplot in Figure S3. Can the authors clarify the “threshold of >25%”. Can they also explain how they tested if markers were impacted by the lower viability, and add this in the text?

10. Figure 1A: the authors have 2 patients who had a prior episode of VL “not within 10 years” (blue) and 2 “within 10 years” (red). How can the authors justify that these patients are immunologically different?

The prior episodes of the two patients in the “non-chronic VL-HIV” cohort who had a prior episode of VL “not within 10 years” (depicted in blue with a “+” in Figure 1B) took place in 1999 and 2007. These prior episodes occurred 227 and 143 months, respectively, before the patient’s active VL development within this study (red diamond symbol).

Meanwhile, the prior episodes of the four patients in the “chronic VL-HIV” cohort who had one prior episode “within 10 years” (depicted in red with “1” for the number of prior episodes) all took place between 2017 and 2019. For these 4 “chronic VL-HIV” patients, the prior episodes occurred 4, 7, 10 or 15 months before the patient’s active VL development within this study (red diamond symbol).

We believe that patients with the prior episodes of 4, 7, 10 or 15 months before the patient’s active VL development differ in disease course compared to the patients with the prior episodes of 227 and 143 months before, and are therefore included in the “chronic VL-HIV” cohort to study whether there are indeed immunological differences.

Please see the comments above, the authors didn’t show any evidence that the two groups are immunologically different.

11. Figure 1A: The blue group is considered as “asymptomatic” , whereas patients have had VL or have never had VL in the past. How can the authors justify that these patients are immunologically

different? It looks like an attempt to increase the number of individuals the AL-HIV group from 5 to 7.

We assume the reviewer intended to refer to the former "primary VL-HIV" group (n = 7) and the 2 patients with past VL >10 years ago, and not the asymptomatic group (AL-HIV) that consists of 20 patients. The "non-chronic VL-HIV" group (formerly known as "primary VL-HIV") is not considered "asymptomatic" but is expected to be immunologically different from the "chronic VL-HIV" group.

12. Line 158: why did the authors use a polyclonal activation, and not an antigen-specific stimulation? The latter will only provide information about the maximal capacity of a given cells to produce a cytokine.

We hypothesized that there would be general anergy in the T cells of VL-HIV patients, warranting the use of a polyclonal activation to see whether the T cells of VL-HIV were indeed no longer capable of sufficient cytokine production. As such, we wanted to evaluate the maximum capacity of the cells to produce a given cytokine, and get a general idea of the full host immune state in a complex co-infection rather than evaluating only a *Leishmania*-antigen specific response, given that there may also be a potential synergistic role of HIV. Moreover, as there is no clarity yet on immunodominant antigens for Leishmania, an antigen-specific stimulation was not preferred.

Can the authors add a sentence to this end in the discussion?

13. How many PBMC were stimulated and how many events were acquired?

Between 25.000 and 100.000 events were acquired in the flow cytometry experiments. The number of cells in the single-cell experiments in mentioned in the manuscript.

The authors didn't answer the questions: how many PBMCs were stimulated, not acquired.

14. Table 1: The area where the study took place has been shown to have many seasonal labour migrants. Are the participants of this study migrant workers or individuals living in the endemic areas around Adburafi?

The majority (93.8%) of the participants of this study were individuals living in the endemic areas around Abdurafi. Two participants were migrant workers from Gondar, Amhara, and another two participants were migrant workers from Humera, Tigray.

It is relevant to know which patients are living in the endemic area and which are migrant workers, particularly given the relatively small number of the 96 participants that are HIV-VL patients that the study largely focuses on - we need to know that these 4 patients don't represent a large proportion of the patients in any of these categories - they could represent more than 1/2 of the non-chronic VL/HIV patients. This should be added in the text of the manuscript.

15. Table 1: The authors are following an HIV+ cohort, but not all individuals are on ART? Why? For the 2 individuals not on ART, did they start ART? What ART drugs were used? Were the HIV patients compliant? If so, how was this assessed?

As evidenced in Table 1 and in accordance with the treat-all strategy currently applied in Ethiopia, all patients but one (62 out of 63 participants) were on ART at the start of the study. For this one

patient, ART was initiated at the recruitment visit. Patient ART regimen information has been added to Table 1. As described in the results section, all patients were ART compliant. Compliance was assessed by self-reporting at each visit (i.e. by questioning the average number of missed doses per month).

16. Table 1: What drugs were used to treat the VL patients?

VL patients are treated with the combination therapy with AmbiSome and Miltefosine, or AmbiSome and Sodium Stibogluconate. These details were included in Table 1 which delves deeper into the characteristics of the VL-HIV patients.

17. Table 2: What are the concomitant diseases affecting these individuals?

These were amebiasis, typhoid fever, pneumonia, and pulmonary tuberculosis. More specifically, there were three cases of typhoid fever and one case of amebiasis in the chronic VL-HIV group, and one case of pneumonia and one case of pulmonary tuberculosis in the non-chronic VL-HIV group. This is now mentioned in the table.

18. CD4 were not measured at EoT?

CD4 counts was routinely measured every 6 months, and additional time points were unfortunately not included in the study protocol.

19. Lines 225: this had already been shown previously by several groups, included some of the authors of this paper, these tables and results should be presented as supplementary data.

We agree that this validates previous findings and have, as recommended, moved these details to Supplementary Table 1. We have decided to maintain Table 2 in the main manuscript as it describes the characteristics of our specific study groups with the main comparison "chronic vs non-chronic VL-HIV". We find that this placement is crucial for the further interpretation of results.

20. Line 238: "As expected..." why is it expected? Please explain?

We thank the reviewer for pointing out this misplaced piece of a sentence. This was originally supposed to be in front of the previous sentence. We have revised the manuscript accordingly by swapping this between sentences. It is now:

"As expected in the Ethiopian context, all AL-HIV and VL-HIV participants were male, and almost all of them (88.6%) were working as farmers or daily labourers, occupations shown to have high risk of Leishmania transmission (Table S1) 40. Although matching in most variables, the majority of our endemic HIV control group were female (57.9%) and did not work in high-risk occupations (52.6%) (Table S1)."

21. Lines 238-239: "the majority of HIV patients were female (57.9%) and did not work in high-risk occupations (52.6)." There is no mention of female study participants in your methodology and suddenly you talk about results from female study participants. This is misleading and confusing.

We recruited patients in a biological sex-agnostic manner, and these are hence referred to "participants" in general in the methods section

Any study population must be described in detail, please amend the section "study population" to include the HIV population in more detail.

22. Lines 240-244: the definitions of asymptomatic are unsubstantiated, as being cured for >3years will not guarantee that these individuals will not relapse; nor that the immunological responses of “true” asymptomatic is similar to those of long-term cured (previous history of VL versus serologically positive but with no previous history of VL).

We thank the reviewer for noticing this. Asymptomatic infection in HIV patients was defined in this study as followed: having positivity on any *Leishmania* marker (rK39 RDT and/or ELISA, DAT, KAtex, and/or PCR) and no VL disease development throughout our study duration. This forms an interesting control group (AL-HIV, independent of prior VL history) for our overt VL (both chronic and non-chronic VL-HIV) study cohort. We agree with the reviewer and have not made any claims on the similarity in immunological response between those with or without previous history of VL. We simply remarked on the fact that ‘none of the AL-HIV participants developed overt VL during the study duration’ and that this thus would be an interesting control group to compare against. Nonetheless, we fully agree with the reviewer that a comparison between ‘true’ asymptomatics versus long-term cured (prior history of VL long before the study) would be very interesting, but we think this is not within the scope of this paper (focused on VL chronicity in HIV co-infection), as our manuscript is already lengthy and complex enough. This separate research question will rather be incorporated in a separate study specific for this question, Including additional experiments and analyses only performed for these two groups.

It was not suggested to compare what the authors of this manuscript called “true” asymptomatic with cured VL patients; but to define “true” asymptomatic. The sentence has been deleted in the “results” section but remains in the “discussion (line 515). The authors have not yet answered how they can postulate that “true” asymptomatic will not relapse. It is well known that there is still no gold standard to define asymptomatic infection. It is therefore not possible to state that asymptomatic “true” or not, will be able to control parasite replication and not experience VL relapses.

23. The authors describe the activation status of CD4+ T cells, but never used a CD4 marker, these cells need to be referred to as CD3+ CD8- cells in the results section.

This is a valid point made by the reviewer and we have revised accordingly.

24. The authors mentioned intracellular IL-10, but there are no results.

We thank the reviewer for this observation and have removed any mentioning of IL-10 in the manuscript.

25. Figure S1: the lymphocyte gate is across 2 populations (likely the monocytes and lymphocytes) but includes half of the lymphocyte population, this is unacceptable.

We thank the reviewer for this observation. This was a gating error that affected a small number of samples in that panel. We have updated all the gatings for every sample and have reanalysed and readjusted in the manuscript accordingly. We have also included new gating strategy figures.

26. Using FOXP3 alone is definitively NOT enough to characterize Treg.

We thank the reviewer for their comment, and we have now opted to remove any mention of FOXP3+ CD4+ T cells or Tregs to improve the focus of the manuscript, and to shorten the paper in accordance with the reviewer’s previous suggestions.

27. Line 134-135. It's not made at all clear how these "representative" patients were chosen. Given the tiny number of cases chosen for scRNA-seq work, the process for choosing these patients seems absolutely critical in making any claim about a broader population from these analyses.

Based on cell viability and patient characteristics, we have chosen the most representative cases to study the differences of each group with scRNA-seq. The characteristics of the selected patients are described in Supplementary Table 2.

It is very hard to judge how useful any of the single-cell data is given the small number of patients selected and the lack of any statistical support for many of the patterns described. This seems a critical weakness of the paper - the authors reach conclusions about 'non-chronic' vs 'chronic' VL-HIV patients in general, based on 4 patients that are not randomly-chosen representatives of any defined cohort of patients, but are selected by the investigators 'Based on cell viability and patient characteristics'. So presumably as most likely to show the hoped-for responses? Any findings that lack statistical support should come with a very visible warning that these are essentially anecdotal.

28. Line 181-183 - Nothing is said about how the QC and filtering criteria were chosen.

The QC parameters were chosen based on visual inspection due to a lack of standardized guidelines. As scRNA-seq is still a relatively new method, QC and filtering criteria are still a remaining challenge in scRNA-seq and lack standardization. However, we followed the recommended approach of visual inspection and outlier removal (Luecken and Theis, 2019, <https://doi.org/10.15252/msb.20188746>).

29. Line 187-189 - The description of cell cluster annotation seems very inadequate - both singleR and scGate allows the use of different reference sets (singleR) and a choice of predefined or custom gating models (scGate). It should be clarified exactly what was done - presumably using some pre-computed human data in each case? We are also told nothing about how results from the two approaches were combined to produce a single annotation.

We thank the reviewer for these valid points. To clarify, we have used the Human Primary Cell Atlas reference from the *celldex* package for SingleR, and the predefined gating models of scGate. The ensemble approach consists specifically of first using the predefined gating models of scGate for several well-known populations, and afterwards executing SingleR for validation. SingleR was then used to annotate any remaining clusters for cell populations that were not in the predefined scGate models. However, all annotations were manually curated based on between-cluster differentially expressed genes. This is an approach that is currently recommended by leading experts (Heumos et al., 2023). We have revised the manuscript with this explanation, accordingly.

30. Line 193. Both the log-fold change and the significance cut-off employed seem very surprisingly loose. The log-2 fold change representing only a 20% change in expression. Similarly, the BH procedure doesn't actually control the type I error rate (and so probably shouldn't be called a p-value, but a false discovery rate) but this implies that 5% of the findings are expected to be false positives. Given the very small number of patients involved here, I have concerns about how likely the gene expression changes identified are to be generalisable.

We thank the reviewer for their comment. The chosen log2 fold change is a Seurat default, and it is indeed somewhat inclusive. We chose the default parameter due to this small number of patients. Instead of focusing on identifying specific genes that could be associated with the disease, we wanted to investigate different pathway trends and profiles in an unbiased manner, resulting in a

more inclusive approach of DEGs. We like to note that increasing the log2 fold change to 0.58 for an estimated 50% increase in expression, resulted in a similar gene expression pattern (see figure below) as the 0.25 threshold in the manuscript.

31. Throughout the section on lines 298-315: nothing is said about the magnitude or statistical significance of differences in cell composition, and the methods section also tell us nothing about the statistical procedures used to support the statements made here.

For our scRNA-seq data, we have not included statistical testing due to the low number of patients per study group. We have demonstrated the scRNA-seq trends and differences in a visual manner. As such, no hard conclusions are made and the data is further discussed in the discussion.

As for point 27, it is very hard to judge how useful any of the single-cell data is given the small number of patients selected and the lack of any statistical support for many of the patterns described. This seems a critical weakness of the paper - the authors reach conclusions about 'non-chronic' vs 'chronic' VL-HIV patients in general, based on 4 patients that are not randomly-chosen representatives of any defined cohort of patients, but are selected by the investigators 'Based on cell viability and patient characteristics'. So presumably as most likely to show the hoped-for responses? Any findings that lack statistical support should come with a very visible warning that these are essentially anecdotal.

32. Lines 369-370 “The proportion of Th17 did not differ between pVL-HIV and cVL-HIV patients at any timepoint (Fig. 2D-E).” Fig. 2D-E shows CD8+ T cells not Th17.

We thank the reviewer for their observation. This sentence referred to former Fig. S5D-E. However, in the newly revised manuscript we have opted to remove any mentioning of Th17 to improve the

focus of the manuscript, and shorten the manuscript.

33. Line 421. Its not at all clear why a 'lack of responsiveness' would result in an actual REDUCTION of these markers with treatment, rather than just the lack of an effect. That seems unexpected and really needs some explanation.

We agree with the reviewer and have removed this sentence and conclusion.

34. Section lines 402-437. As far as we are told, (line 195) the top 12 enriched pathways were considered, and nothing is said about how 'enriched' these pathways need to be in differentially expressed genes before they are reported. We are told nothing about the numbers of genes and the magnitude of changes in pathways reported here, so it's impossible to evaluate these results: there is a basic lack of detail about the results here, and just showing some heatmaps does not give enough information.

We agree with the reviewer and have added the pathway enrichment scores and number of genes in supplementary figure 11 to provide more information to evaluate the results with.

35. Line 435: This problem is particularly acute when we need to evaluate claims like this: "While this could indicate that treatment has few consistent effects on the expression of genes of these patients, this could also be partly explained by the observed depletion of CD16+ monocytes within this patient group" - as its impossible to work out from the results presented here whether this apparent difference between groups is likely to be a genuine difference, or due to differences in the power to detect expression changes between the groups.

We have deleted the monocytes and NK cell data in order to focus the manuscript on T cells, and have altered the discussion accordingly while taking note of the reviewer's comments.

36. Line 438 - it's not clear what the clonotype expansion analysis is intended to show - or precisely what the claim is - beyond the fact that there is less increase in CD4+ and CD8+ T-cells following treatment in the cVL group than pVL group, which was already observed (e.g. fig 2). From the figures, it looks like all clonotypes expand, but to a lesser extent in chronic than primary patients. That would just seem to be the expectation, given that we know cell numbers recover less in chronic patients. Are the authors claiming that the dynamics of the most common clonotypes are different to others? If so, that should be clarified, and some kind of statistical analysis would seem warranted.

The clonotype expansion analyses in Figure 8A indicate the expansion/proliferation of the top 30 largest expanded T cell clones (i.e. T cell receptor sequence-based clonotypes, T cells carrying the same TCR) during antileishmanial treatment in non-chronic patients, while this expansion/proliferation is not observed in chronic patients (Fig. 8B). These expanded/proliferated T cell clonotypes are represented on the UMAP in Fig 8C. This is different from Figure 2 where we provide the proportions of all CD8- and CD8+ T cells in the blood compartment. The top expanded T cell clonotypes are most likely to be involved in an active infection due to a shift in dynamics (i.e. expansion or contraction), and we can speculate that these may likely be *Leishmania*-specific T cells. In this figure, we do not see all clonotypes expanding, some contract and disappear, while some expand greatly and become dominant clones. These dominant clones are most likely to be involved in the active infection, and may be important for resolving the infection. For example, in the non-chronic patients, we see several clonotypes expanding to a proportion several folds larger than previously, becoming the dominant clones, while several other clonotypes contract or disappear from the top 30. The total proportion of the top 30 clonotypes of the non-chronic group shifts from

around 0.275 to 0.425, nearly doubling in size. In contrast, this total proportion of the top 30 clonotypes of the chronic group shifts from roughly 0.225 to 0.25, showing little to no expansion in response to the active infection. This indicates anergy from stimulation, something we can not show in figure 2. In addition, when overlaying these top 30 T cell clonotypes on the UMAP of the VL-HIV patients (Fig 4C), we can deduce that the expanded T cell clonotypes are CD4+ and CD8+ Tem cells in the non-chronic group, but only CD8+ Tem cells in the chronic group. This highlights CD4+ T cell functional anergy. Due to the limited number of patients per study group for scRNA-seq, we have not included statistical analysis.

The interpretation of Figure 7 'clonotype expansion' results is still confusing. One concern is the arbitrary classification of 'expansion levels' in figures 7c and d - is the proposed (but not statistically tested) difference in "changes in proportions" robust to the definition of (e.g.) hyper-expanded? Also, the figures presented here are proportions, where the denominator (e.g. number of cells per subtype in different patients at different timepoints) are changing as well as the numerator. It would be very easy to see bigger changes in proportions if the denominator was small, while in a different subset (with a bigger denominator) the changes in proportion would be much smaller, even for the same absolute size of cell lineage expansion. This analysis would be much more convincing if data on actual cell numbers were shown.

Reviewer #2 (Remarks to the Author):

The authors have modified their manuscript in response to reviewer's comments. 'People first' language has been added in some places, although there is still a considerable usage of 'patient' (e.g. line 135). In regards to 'orphaned' mentions of analyses or cell types (e.g. NK cells) these have been removed allowing the focus to remain on the T cell populations described across the remainder of the manuscript; this has clarified the messaging of the paper. Figures pinpointed as being over-complicated have been simplified although the longitudinal graphs are still quite busy. The main concern over single marker analysis to signify 'exhaustion' has partly been addressed by co-expression analysis where possible within the limitation of the panel; as TIGIT single expression is still relied upon as a main overall conclusion (with IFN γ / single cell data) the paper could be strengthened by commenting on the complexities of TIGIT expression (e.g. related to lower vs. higher function in different studies). Overall the paper still has a few limitations (e.g. HIV participant controls not being completely matched to the VL-HIV participants) but the authors are honest about these in the discussion and have clarified the messaging compared to the original version.

Reviewer #3 (Remarks to the Author):

The authors have address all of my concerns.

In this document, you can find **our point-by-point rebuttal to the latest concerns of reviewer #1 in green text and outlined in a box**. As shown in this document, the majority of initial concerns by this reviewer were already addressed (unboxed, in grey and light blue), but some novel unjustified concerns remain (and many of the comments address the same unjustified concern). We hope that the editor can agree that while reviewer #1 gave some good feedback (which we addressed properly), there were also instances of inadequate interpretation, aggressive tone, moving of the goalposts, and a lack of knowledge of some matters concerning our manuscript that we believe has unjustifiably influenced the decision. In addition, we emphasize that we have addressed all remaining minor concerns of reviewer #2 (as shown below). We hope our revised manuscript and point-by-point reply are sufficient for the editor to reconsider publication of our manuscript.

To avoid a delay in research dissemination, we look forward to a timely decision.

Reviewer comments

Reviewer #1 (Remarks to the Author):

The study by de Vrij et al. followed cohorts of HIV and VL-HIV patients in Ethiopia and measured T cell intracellular cytokine production in response to polyclonal activation, as well as single cell RNA and T cell receptor sequencing. Their results show that the majority of VL-HIV patients in their study had a previous history of VL and that these individuals are characterized by higher proportions of PD1+ and TIGIT+ CD4+ T cells and PD-1+ CD8+ T cells; by marked CD4+ and CD8+ T cell unresponsiveness at the single-cell transcriptional level; as well as by an absence of lymphoproliferative response at the clonotype level. They also showed that in VL-HIV individuals who relapsed, there was lower percentages of CD4+ T cells producing IFN- γ in response to polyclonal activation. The paper is unnecessary long, the data are poorly organised and some of the conclusions are not supported by the results.

We thank the reviewer for the detailed review and suggestions. We have now significantly focused the paper on T cells only, reformulated conclusions, and have reduced the length of the manuscript to make it as concise as possible. We hope these adaptations (more explained below) have led to better organized data and clarified anything that was not clear in the original manuscript.

The number of patients in one of the cohorts (primary VL-HIV) is very low (n=7), and for some of the time points, only two patients were analysed.

We like to point out that given that this was a longitudinal study in a hard-to-reach population (e.g. it can take patients several days to travel from their home to reach the health centre), follow-up study visits were often missed (even more so due to the COVID-19 pandemic near the end of the study). This restriction in available timepoints only affected the longitudinal analyses, in which an appropriate statistical analysis method was applied (i.e. linear mixed effect models) that takes these missing values into account.

Response Reviewer:

The description of the cohorts, particularly the VL-HIV cohorts, is completely inadequate. The definition of “post-asymptomatic” is dubious as this characterized successfully treated VL patients, i.e. HIV patients who had VL and were treated with anti-leishmanial treatment. Furthermore, in the primary VL-HIV cohort, the authors use two distinct groups of patients: VL-HIV individuals who already had 1 episode of VL, and those who have had no previous VL, but tested positive by serological tests or PCR. Several studies, some published by the authors of this paper, have shown that a previous VL history in HIV patients is a risk factor for further relapse.

Response Authors:

We have removed the 'post-asymptomatic' term which can indeed be dubiously interpreted. The description of the cohorts has now been better clarified, and the 'primary' nomenclature has been changed to 'non-chronic' to account for the two patients with a VL episode that occurred more than 10 years ago. We agree that VL history is a risk factor for relapse, although mostly for subsequent VL relapse within 1-2 years, but argue that the VL episodes that occurred more than 10 years ago in these two patients did not jeopardize our objective (i.e. compare immune state between chronic and non-chronic VL-HIV patients), and likely suggests the presence of a new *Leishmania* infection rather than a recurrent VL episode. We defined chronic patients as those who undergo frequent recurrent episodes, while non-chronic patients were patients that were able to limit the infection (like the 2 patient with past VL >10 years ago) or were primarily infected.

Response Reviewer:

The 2 patients in question were VL-HIV patients with a previous history of relapse. The authors cannot assess whether these patients were immunologically different from those who had no relapse. Furthermore, there is no evidence for the author to suggest "the presence of a new Leishmania infection". This seems at best unproven: there is considerable evidence that VL parasites can persist for years following cure, and this is likely to have ongoing impact on immune function (e.g. [https://doi.org/10.1016/S0163-4453\(03\)00002-1](https://doi.org/10.1016/S0163-4453(03)00002-1); doi:10.1016/S0140-6736(86)91725-3). We don't really know how common this is, as most of this literature is based on disease re-emerging in non-endemic locations, and so is necessarily rather anecdotal – it's certainly the case that disease relapse over 1 year after apparent cure is due to the original infection re-emerging (DOI: 10.1128/mBio.00971-21), so it seems naive to assume this is not the case over longer time periods. There is also evidence that many Leishmania species can reach a dormant state likely connected to this clinical persistence (<https://doi.org/10.1073/pnas.1619265114>; <https://doi.org/10.3389/fcimb.2023.1253033>). So the grouping of even 10+ year previous VL infections is a concern. Can the authors define "non-chronic" in the text when they use it first (line 98)?

Response Authors:

We argue that this is an unjustified comment as our study objective was not about comparing whether the VL-HIV patients with a previous history of relapse were immunologically different from those that had no relapse. We emphasize that our experimental setup and primary objective focuses on comparing chronic (frequent VL episodes in a short amount of time) versus non-chronic patients (none or close to no episodes throughout the years). We understand the confusion as comparing 'relapse' and 'no relapse' was present in our initial submission, but this answer suggests an incomplete and non-thorough review of our revised manuscript.

Our revision contains the definitions of 'chronic' and 'non-chronic', which were based on the definition by Bourgeois et al. (doi: 10.1111/j.1468-1293.2010.00846.x), where they characterize patients as 'chronic' if there is a presence of relapses over a period of several years. We strongly argue that the two patients with a VL episode >10 years ago fall under the non-chronic category. Whether these latter two patients are immunologically different from 'primary VL-HIV' patients is a different question, and this is not something we argue. In addition, whether the new VL episode during our study was the presence of a new *Leishmania* infection, or a recurrence of a persisting parasite niche, is irrelevant within the context of our study objective. We agree with the reviewer that it is likely that recurrence within one year is due to the original infection re-emerging, and that this could even persist for longer – which is exactly what we wrote on L70-L71 where we cited the same study as the reviewer cites here.. Finally, we have better defined non-chronic and chronic in the text in our newly revised manuscript, per the reviewer's suggestion.

Response Reviewer:

The authors also chose those individuals from the primary VL-HIV group who had no VL relapses; and incidentally, those with some of the highest number of previous relapses in the chronic VL/HIV group, to perform their single cell RNA sequencing.

Response Authors:

We would like to clarify that these individuals were indeed deliberately chosen as they were the most representative cases of the non-chronic VL-HIV and chronic VL-HIV groups, and therefore most interesting to conduct the more in-depth analyses on, in order to better define immunological signatures of chronic VL-HIV.

Response Reviewer:

If these individuals were deliberately chosen, it has to be clearly specified in the text.

Response Authors:

This is an unjustified comment as we clearly specified on L133-L134 that these participants were chosen based on cell viability and patient characteristics.

Response Reviewer:

1. Line 44: PD1 and TIGIT are one of many markers of T cell exhaustion and anergy. T cells cannot be characterized as anergic or exhausted using only one marker, and indeed, PD-1 can be expressed by activated T cells. The abstract needs to be rephrased.

Response Authors:

We agree with the reviewer's comment and have better structured the paper to build up the evidence for this statement, and during revisions we have more carefully worded any other claims in the manuscript accordingly. For example, we now describe PD1 and TIGIT rather as exhaustion-associated markers, or we just simply mention higher levels of PD1 and TIGIT without categorizing these as associated with any functionality or lack thereof.

Response Reviewer:

Change exhaustion on line 517 to "levels of exhaustion-associated markers".

Response Author:

This is a minor comment that we have now addressed in the revised manuscript.

Response Reviewer:

2. Introduction: our knowledge about the mechanisms accounting for VL relapses in VL-HIV individuals is still limited, however, quite a few studies in Africa and in Brazil have detailed some of the potential mechanisms. This is poorly described in the introduction and needs to be developed further. The introduction is in general quite vague, with little detail on what has already been published in HIV and VL-HIV patients.

Response Authors:

Potential underlying mechanisms for VL relapses were deliberately only briefly listed in the introduction with references provided for broader scoping, to keep the paper focused (and shorter) on our key messages that are not directly related to any mechanisms. It is not our intention to provide a comprehensive overview of all potential general mechanisms for VL relapse.

Response Reviewer:

The title of this paper is *Persistent T cell unresponsiveness associated with chronic visceral leishmaniasis in HIV-coinfected patients*” and one of the key messages of this paper is “These findings suggest PD-1 and TIGIT play a pivotal in VL disease chronicity in HIV co-infected patients”. It is therefore appropriate to briefly summarize what is already known about T cell responsiveness in this field.

Can the authors explain why the paper of Wherry et al is cited on line 75? This paper doesn't address the loss of host immune response during leishmaniasis.

And also complete the sentence: play a pivotal... in VL disease.

Response Authors:

We discussed what is already known about T cell responsiveness in VL(-HIV) in the discussion section based on our findings, and feel we have already adequately discussed this in the manuscript. We lack a clear suggestion from the reviewer on what the reviewer would like to see more in the introduction section, while also stating the paper should be shorter and more focused?

As for the citation of Wherry et al., this was a minor mistake and we have removed this citation as per reviewer's suggestion. Similarly, we have completed the incomplete sentence in the abstract on L46-L47 changing 'play a pivotal in VL-HIV chronicity' to 'play a pivotal role in VL-HIV chronicity'.

Response Reviewer:

3. Line 77: the authors use a reference that mainly describes Th1 responses in experimental models, but not in human, a more appropriate reference needs to be used in the context of a human study.

Response Authors:

We thank the reviewer for pointing this out, but we do not know which specific reference the reviewer means. In the sentence “A Th2 polarised immune response is permissive for parasite replication, while protective immunity to VL has been linked to a CD4+ T helper 1 (Th1) polarised immune response with high interferon gamma (IFN- γ) production 24-26.”, we listed three different references of which two (24-25) mention Th1 responses in human VL disease. We agree with the reviewer that reference 26 is not specific enough, and have removed this reference from the manuscript. Instead, we have included a new reference instead that specifically describes Th1 responses in human VL: Khalil, E.A., Ayed, N.B., Musa, A.M., Ibrahim, M.E., Mukhtar, M.M., Zijlstra, E.E., Elhassan, I.M., Smith, P.G., Kieny, P.M., Ghalib, H.W., Zicker, F., Modabber, F. & Elhassan, A.M. Dichotomy of protective cellular immune responses to human visceral leishmaniasis. Clin Exp Immunol 140, 349-353 (2005).

Response Reviewer:

The notion that Th1 is associated with healing was extensively described in experimental model, but has been largely challenged in human VL (e.g. in 10.1016/j.it.2007.07.004 and 10.1128/CVI.00143-12). Please amend the text and your reference accordingly.

Response Authors:

We are aware of the publications challenging this notion, but in a similar manner we can also provide references confirming it. In addition, the references we added in our previously revised manuscript based on the initial concern already underscore the fact that this notion has been challenged, and the reviewer did not ask us to clarify at the time. Nevertheless, we have adapted the manuscript to mention that this originates predominantly from experimental models, but has been challenged in human VL.

4. Line 84: what is the rationale for analysing the role of NK cells?

To improve the focus of the paper, we have deleted all paragraphs on NK cells.

Response Reviewer:

5. Line 90: a study was published in 2021 that identified markers that can be used to predict VL relapse, this is not cited in the introduction.

Response Authors:

We thank the reviewer for this comment, however no specific DOI or reference was cited by the reviewer for the proposed study. We agree that several papers were published in 2021 on potential markers to predict VL relapse in immunocompetent (qPCR, etc.) and immunocompromised individuals. If the reviewer is willing to provide the reference to the suggested manuscript, we will consider to cite it in an appropriate section.

Response Reviewer:

There were few papers published in 2021 that discuss potential markers of relapse, and by typing the key words: “prediction markers and visceral leishmaniasis and HIV, there are 3 papers coming up. Unless the authors have issues with accessing PubMed, it should be straight forward to find it.

Response Authors:

We argue this is an unjustified comment. The reviewer particularly mentioned ‘a study’ in the initial comment, and we interpreted the comment to be about a specific manuscript we should have included. The aggressive tone of the reviewer on this concern could have been avoided, in particular because it originates from an unclear comment by the reviewer. The reviewer suggested we should include citations of studies on markers that can be used to predict VL, but this was already in the manuscript, namely: citation #9, #10, #30, and #31. The first two papers that come up with these keywords in Pubmed (per the reviewer’s rude recommendation) are manuscripts that we cited in the discussion, namely citation #41 and #42. This points to an incomplete review by referee #1.

Response Reviewer:

6. Line 112: study population. The authors use indiscriminately VL/HIV individuals who already had 1 episode of VL, and those who didn’t, but tested positive by serological tests or PCR, even though a previous history of VL in VL/HIV patients is a known risk factor for further relapse. Unless the authors can show compelling evidence that these two groups are immunologically comparable, they should be treated as separate groups.

Response Authors:

As outlined above, the description of the cohorts has been clarified and group names were changed accordingly to clarify the goal of the comparison between chronic and non-chronic VL-HIV patients.

Response Reviewer:

Please see the comments above, the authors didn’t show any evidence that the two groups are immunologically different.

Response Authors:

We argue this is an unjustified comment as already addressed above in the first comment.

7. Lines 114-115: “HIV patients were followed up between October 2017 and November 2020”. This is for a period of 37 months. However, the timeline in Fig 1B shows patients followed from 20 months before to 30 months after they developed VL. This doesn’t make sense.

Not all patients were followed 20 months before and 30 months after they developed VL (see Fig 1B). The plot axis range from a maximum of 20 months before to a maximum of 30 months after the patients developed VL (D0). However, these patients were recruited at different times between October 2017 and November 2020, developed VL at different times during this period, and were followed-up after disease development for different periods of time. The individual lines of a patient display the follow-up period length, and as evidenced by plot Fig 1B the maximum follow-up time of a patient was around 33

months. To clarify this, we have included an additional sentence in the Figure legend: “The depicted months of follow-up were calculated backwards and onwards from the time of active VL development (D0) to standardize timepoints between patients”.

Response Reviewer:

8. Lines 119-120: There are discrepancies between the timeline showed in Fig 1B and the text on line 119-120: “visits at 6 and 12 months after the end of VL treatment”. However, the timeline in Fig 1B shows a follow-up for up to 30 months.

Response Authors:

The full sentence in lines 119-120 states “This follow-up phase lasted until study end and included visits at 6 and 12 months after the end of VL treatment, and at any suspected VL disease episode.” Any follow-up after the 12 months was therefore due to a suspected VL disease episode (represented as unscheduled visits or VL relapse visits).

Response Reviewer:

This sentence is still unclear as “unscheduled” visit are not defined. Please rephrase the legend of Figure 1b

Response Authors:

The sentence containing “unscheduled visits” was not used in the manuscript, but only in the response to the reviewer to clarify that ‘any suspected VL disease episode’ represented either an unscheduled visit on which VL was not clinically confirmed, or a visit on which VL relapse was confirmed. Unscheduled visits are in this case just defined as a regular visit in the figure. As such, we believe this does not need further defining in the figure. We argue that the initial concern originated from the reviewer only reading and citing a part of the sentence instead of the full sentence, and that this novel concern also originated from the reviewer not reading the revised manuscript thoroughly.

Response Reviewer:

9. Line 133: 4-6 hours is a long time for whole blood or purified PBMC to stay on the bench, especially if this was done in Abdurafi, where the temperatures are high. Did the authors test the impact of this long wait on survival and activation markers? There is an unusually high number of dead cells (Figure S3).

If the PBMCs were purified in Abdurafi, did the author have a -80oC freezer to keep the cells frozen, what about liquid nitrogen?

Response Authors:

We would like to clarify that we followed a “standardized operating procedure” (SOP) for the isolation of PBMCs from whole blood that follows those used globally. In this SOP it states that collected venous blood can be kept for a maximum of 4h on room temperature i.e. 20-25°C before processing. The PBMC isolation itself takes a maximum of 2 hours, and isolated venous blood samples were kept on the bench in the air conditioning-controlled laboratory in Abdurafi, where we also performed the PBMC isolation. After PBMC isolation, samples were kept in a cool cell at -80°C overnight and stored in LN2 the next day, according to cell cryopreservation guidelines. This is commonly done in our other studies in Europe and Africa, and no correlation between this waiting time and viability or T cell activation has been observed in the past.

We agree that the sample shown in Figure S3 had a relatively high number of dead cells for PBMCs as compared to the samples of the other panels. However, in contrast to these other panels, this panel was stimulated with LAC, BD GolgiPlug, and BD GolgiStop. The first panel in Figure S1 was not stimulated, and the second panel in Figure S2 was stimulated with LAC and BD GolgiPlug only. We

expect this to be the reason for the relatively lower viability of this panel. Nonetheless, we now set a viability threshold of >25%, and we further analysed whether any of the markers were impacted by the lower viability and there does not seem to be a correlation between viability and the expression of the markers included in our study.

Response Reviewer:

There is no dotplot in Figure S3. Can the authors clarify the “threshold of >25%”. Can they also explain how they tested if markers were impacted by the lower viability, and add this in the text?

Response Authors:

The mentioned ‘dotplot in Figure S3’ was only included in Figure S3 of our initial submission, and our initial response to the reviewer specifically clarified on this figure. That Figure S3 consisted of data of a flow cytometry panel that was removed entirely in our revised manuscript, and Figure S3 now concerns other analyses. While we do understand the confusion as we did not mention the removal of this panel in our earlier response to the reviewer, we do argue the reviewer should have clearly understood this from the revised manuscript, again pointing towards an incomplete review by the reviewer.

In addition, we have now clarified the threshold of >25% viability in the text of the newly revised manuscript. Finally, we have done correlation analyses between the viability and the expression of a marker, and we noted that there was no correlation above a threshold of 25%. This has been added in the text of the newly revised manuscript.

Response Reviewer:

10. Figure 1A: the authors have 2 patients who had a prior episode of VL “not within 10 years” (blue) and 2 “within 10 years” (red). How can the authors justify that these patients are immunologically different?

Response Authors:

The prior episodes of the two patients in the “non-chronic VL-HIV” cohort who had a prior episode of VL “not within 10 years” (depicted in blue with a “+” in Figure 1B) took place in 1999 and 2007. These prior episodes occurred 227 and 143 months, respectively, before the patient’s active VL development within this study (red diamond symbol). Meanwhile, the prior episodes of the four patients in the “chronic VL-HIV” cohort who had one prior episode “within 10 years” (depicted in red with “1” for the number of prior episodes) all took place between 2017 and 2019. For these 4 “chronic VL-HIV” patients, the prior episodes occurred 4, 7, 10 or 15 months before the patient’s active VL development within this study (red diamond symbol). We believe that patients with the prior episodes of 4, 7, 10 or 15 months before the patient’s active VL development differ in disease course compared to the patients with the prior episodes of 227 and 143 months before, and are therefore included in the “chronic VL-HIV” cohort to study whether there are indeed immunological differences.

Response Reviewer:

Please see the comments above, the authors didn’t show any evidence that the two groups are immunologically different.

Response Authors:

Again, we argue this is an unjustified comment as already addressed above in the first comment.

11. Figure 1A: The blue group is considered as “asymptomatic”, whereas patients have had VL or have never had VL in the past. How can the authors justify that these patients are immunologically different? It looks like an attempt to increase the number of individuals the AL-HIV group from 5 to 7.

We assume the reviewer intended to refer to the former "primary VL-HIV" group (n = 7) and the 2 patients with past VL >10 years ago, and not the asymptomatic group (AL-HIV) that consists of 20 patients. The "non-chronic VL-HIV" group (formerly known as "primary VL-HIV") is not considered "asymptomatic" but is expected to be immunologically different from the "chronic VL-HIV" group.

Response Reviewer:

12. Line 158: why did the authors use a polyclonal activation, and not an antigen-specific stimulation? The latter will only provide information about the maximal capacity of a given cells to produce a cytokine.

Response Authors:

We hypothesized that there would be general anergy in the T cells of VL-HIV patients, warranting the use of a polyclonal activation to see whether the T cells of VL-HIV were indeed no longer capable of sufficient cytokine production. As such, we wanted to evaluate the maximum capacity of the cells to produce a given cytokine, and get a general idea of the full host immune state in a complex co-infection rather than evaluating only a *Leishmania*-antigen specific response, given that there may also be a potential synergistic role of HIV. Moreover, as there is no clarity yet on immunodominant antigens for *Leishmania*, an antigen-specific stimulation was not preferred.

Response Reviewer:

Can the authors add a sentence to this end in the discussion?

Response Authors:

We agree with the reviewer that this could be added as a small paragraph in the discussion. As such, we have provided this in a newly revised version of our manuscript.

Response Reviewer:

13. How many PBMC were stimulated and how many events were acquired?

Response Authors:

Between 25.000 and 100.000 events were acquired in the flow cytometry experiments. The number of cells in the single-cell experiments in mentioned in the manuscript.

Response Reviewer:

The authors didn't answer the questions: how many PBMCs were stimulated, not acquired.

Response Authors:

The reviewer asked for both how many PBMC were stimulated, and how many events were acquired. We did answer one of these questions in the initial response, namely: how many were acquired. For how many PBMC were stimulated, we already clearly wrote in the text of our manuscript that we stimulated with 1uL of LAC per 300.000 cells. As the number of acquired events is more important, and often the only number reported, we assumed the reviewer referred mostly to this, and thought the mentioning of stimulation with 1uL of LAC per 300.000 cells in the methods would suffice. However, we like to clarify that we typically stimulate between 100.000 to 500.000 cells per well.

Response Reviewer:

14. Table 1: The area where the study took place has been shown to have many seasonal labour migrants. Are the participants of this study migrant workers or individuals living in the endemic areas around Adburafi?

Response Authors:

The majority (93.8%) of the participants of this study were individuals living in the endemic areas around Abdurafi. Two participants were migrant workers from Gondar, Amhara, and another two participants were migrant workers from Humera, Tigray.

Response Reviewer:

It is relevant to know which patients are living in the endemic area and which are migrant workers, particularly given the relatively small number of the 96 participants that are HIV-VL patients that the study largely focuses on - we need to know that these 4 patients don't represent a large proportion of the patients in any of these categories - they could represent more than 1/2 of the non-chronic VL/HIV patients. This should be added in the text of the manuscript.

Response Authors:

All four participants that are migrant workers were from the chronic VL-HIV patient group. We have made this clearly available in Table 1 and clarified in text in a newly revised version of the manuscript.

15. Table 1: The authors are following an HIV+ cohort, but not all individuals are on ART? Why? For the 2 individuals not on ART, did they start ART? What ART drugs were used? Were the HIV patients compliant? If so, how was this assessed?

As evidenced in Table 1 and in accordance with the treat-all strategy currently applied in Ethiopia, all patients but one (62 out of 63 participants) were on ART at the start of the study. For this one patient, ART was initiated at the recruitment visit. Patient ART regimen information has been added to Table 1. As described in the results section, all patients were ART compliant. Compliance was assessed by self-reporting at each visit (i.e. by questioning the average number of missed doses per month).

16. Table 1: What drugs were used to treat the VL patients?

VL patients are treated with the combination therapy with AmbiSome and Miltefosine, or AmbiSome and Sodium Stibogluconate. These details were included in Table 1 which delves deeper into the characteristics of the VL-HIV patients.

17. Table 2: What are the concomitant diseases affecting these individuals?

These were amebiasis, typhoid fever, pneumonia, and pulmonary tuberculosis. More specifically, there were three cases of typhoid fever and one case of amebiasis in the chronic VL-HIV group, and one case of pneumonia and one case of pulmonary tuberculosis in the non-chronic VL-HIV group. This is now mentioned in the table.

18. CD4 were not measured at EoT?

CD4 counts was routinely measured every 6 months, and additional time points were unfortunately not included in the study protocol.

19. Lines 225: this had already been shown previously by several groups, included some of the authors of this paper, these tables and results should be presented as supplementary data.

We agree that this validates previous findings and have, as recommended, moved these details to Supplementary Table 1. We have decided to maintain Table 2 in the main manuscript as it describes the characteristics of our specific study groups with the main comparison "chronic vs non-chronic VL-HIV". We find that this placement is crucial for the further interpretation of results.

20. Line 238: "As expected..." why is it expected? Please explain?

We thank the reviewer for pointing out this misplaced piece of a sentence. This was originally supposed to be in front of the previous sentence. We have revised the manuscript accordingly by swapping this between sentences. It is now: "As expected in the Ethiopian context, all AL-HIV and VL-HIV participants were male, and almost all of them (88.6%) were working as farmers or daily labourers, occupations shown to have high risk of Leishmania transmission (Table S1) 40. Although matching in most variables, the majority of our endemic HIV control group were female (57.9%) and did not work in high-risk occupations (52.6%) (Table S1)."

Response Reviewer:

21. Lines 238-239: "the majority of HIV patients were female (57.9%) and did not work in high-risk occupations (52.6)." There is no mention of female study participants in your methodology and suddenly you talk about results from female study participants. This is misleading and confusing.

Response Authors:

We recruited patients in a biological sex-agnostic manner, and these are hence referred to “participants” in general in the methods section

Response Reviewer:

Any study population must be described in detail, please amend the section “study population” to include the HIV population in more detail.

Response Authors:

It is entirely unclear what details the reviewer suggests to add to the study population section. Even in our initial manuscript, we had already included a reference to our larger cohort study which was registered on the clinicaltrials.gov database (NCT03013673), and this already contains all participation criteria. Next, we provided complete summary statistics of the HIV participant group in our manuscript. We believe this is sufficient to describe the study population in detail, and that this is an unjustified concern by the reviewer.

Response Reviewer:

22. Lines 240-244: the definitions of asymptomatic are unsubstantiated, as being cured for >3years will not guarantee that these individuals will not relapse; nor that the immunological responses of “true” asymptomatic is similar to those of long-term cured (previous history of VL versus serologically positive but with no previous history of VL).

Response Authors:

We thank the reviewer for noticing this. Asymptomatic infection in HIV patients was defined in this study as followed: having positivity on any *Leishmania* marker (rK39 RDT and/or ELISA, DAT, KAtex, and/or PCR) and no VL disease development throughout our study duration. This forms an interesting control group (AL-HIV, independent of prior VL history) for our overt VL (both chronic and non-chronic VL-HIV) study cohort. We agree with the reviewer and have not made any claims on the similarity in immunological response between those with or without previous history of VL. We simply remarked on the fact that ‘none of the AL-HIV participants developed overt VL during the study duration’ and that this thus would be an interesting control group to compare against. Nonetheless, we fully agree with the reviewer that a comparison between ‘true’ asymptomatics versus long-term cured (prior history of VL long before the study) would be very interesting, but we think this is not within the scope of this paper (focused on VL chronicity in HIV co-infection), as our manuscript is already lengthy and complex enough. This separate research question will rather be incorporated in a separate study specific for this question, including additional experiments and analyses only performed for these two groups.

Response Reviewer:

It was not suggested to compare what the authors of this manuscript called “true” asymptomatic with cured VL patients; but to define “true” asymptomatic. The sentence has been deleted in the “results” section but remains in the “discussion (line 515). The authors have not yet answered how they can postulate that “true” asymptomatic will not relapse. It is well known that there is still no gold standard to define asymptomatic infection. It is therefore not possible to state that asymptomatic “true” or not, will be able to control parasite replication and not experience VL relapses.

Response Authors:

We again argue this is an unjustified response, as the reviewer incorrectly stated we were only asked to define ‘true’ asymptomatic, and that we were not asked to remark on the comparison between

'true' asymptomatic and cured VL patients (see initial response). The reviewer did in fact not ask us to define 'true' asymptomatics initially, as in the initial response the reviewer stated "the definitions of asymptomatic were unsubstantiated" and next remarked on that "being cured for >3 years will not guarantee that these individuals will not relapse nor that the immune response of 'true asymptomatics' is similar to those of long-term cured patients.", thus vaguely suggesting us to remark upon this latter issue. We argue we were mostly asked to substantiate the definition of asymptomatic, and then remark upon a comparison between 'true asymptomatics' and long-term cured VL patients, which we clearly remarked upon in our initial response.

The reviewer now raises new concerns, namely: 1) to define 'true asymptomatics', 2) to postulate how 'true asymptomatics' will not relapse, and 3) that it is impossible to state that 'true asymptomatics' will be able to control parasite replication and not experience VL relapses.

With regards to the first concern of defining 'true' asymptomatic, we mean that these are individuals that are truly infected, have not experienced any prior disease (and are thus not long-term cured), and are able to remain asymptomatic for a long period of time. With regards to #2 and #3 – these 'true asymptomatics' will not relapse because they have never experienced any prior disease. While we could discuss these semantics further, we stress for concern #2 and #3 that whether these participants would, far in the future, develop VL or not, remains irrelevant to our comparison with VL-HIV patients, and that this does not in any way affect our main comparison between non-chronic and chronic VL-HIV patients.

Furthermore, we are well aware there is no golden standard to define asymptomatic infection, and therefore have used the most stringent approach available to define them by following them longitudinally to confirm no VL development (as was clearly defined in the manuscript). Again, we believe this unjustified concern comes from a misinterpretation of the initial formulation, an unclear initial comment by the reviewer, and incomplete reading by the reviewer.

23. The authors describe the activation status of CD4+ T cells, but never used a CD4 marker, these cells need to be referred to as CD3+ CD8- cells in the results section.

This is a valid point made by the reviewer and we have revised accordingly.

24. The authors mentioned intracellular IL-10, but there are no results.

We thank the reviewer for this observation and have removed any mentioning of IL-10 in the manuscript.

25. Figure S1: the lymphocyte gate is across 2 populations (likely the monocytes and lymphocytes) but includes half of the lymphocyte population, this is unacceptable.

We thank the reviewer for this observation. This was a gating error that affected a small number of samples in that panel. We have updated all the gatings for every sample and have reanalysed and readjusted in the manuscript accordingly. We have also included new gating strategy figures.

26. Using FOXP3 alone is definitively NOT enough to characterize Treg.

We thank the reviewer for their comment, and we have now opted to remove any mention of FOXP3+ CD4+ T cells or Tregs to improve the focus of the manuscript, and to shorten the paper in accordance with the reviewer's previous suggestions.

Response Reviewer:

27. Line 134-135. It's not made at all clear how these "representative" patients were chosen. Given the tiny number of cases chosen for scRNA-seq work, the process for choosing these patients seems absolutely critical in making any claim about a broader population from these analyses.

Response Authors:

Based on cell viability and patient characteristics, we have chosen the most representative cases to study the differences of each group with scRNA-seq. The characteristics of the selected patients are described in Supplementary Table 2.

Response Reviewer:

It is very hard to judge how useful any of the single-cell data is given the small number of patients selected and the lack of any statistical support for many of the

patterns described. This seems a critical weakness of the paper - the authors reach conclusions about 'non-chronic' vs 'chronic' VL-HIV patients in general, based on 4 patients that are not randomly-chosen representatives of any defined cohort of patients, but are selected by the investigators 'Based on cell viability and patient characteristics'. So presumably as most likely to show the hoped-for responses? Any findings that lack statistical support should come with a very visible warning that these are essentially anecdotal.

Response Authors:

The reviewer raises three separate issues with these new concerns, namely: 1) small patient numbers, 2) a proposed lack of statistical support and 3) non-randomly chosen representative of defined patient cohorts selected based on cell viability and patient characteristics.

With regards to #1 - While we concede the fact that there is a small number of patients for the single-cell subset analyses, we argue that we predominantly used these subanalyses to validate and further explore findings discovered on the flow cytometry level for which we had higher and sufficient patient numbers.

With regards to #2 – This is not entirely correct. We do provide statistical support for the differential gene expression analysis (which reviewer #1 even commented on), but we indeed do not provide statistical comparisons for the cellular composition and the clonal expansion analyses at the single-cell level. A manuscript on the best practices for single-cell sequencing analyses (<https://www.nature.com/articles/s41576-023-00586-w>) points out that current methods for such a statistical analysis lack an independent benchmark, and that identified signals are often a statistical artefact. Thus, we considered against this type of unvalidated statistical approach considering the patient numbers, and instead opted to use these single-cell analyses solely as validation for the statistically significant patterns at the flow cytometry level. Overall, we reiterate that we do provide statistical support for described patterns that underlie our main findings.

With regards to concern #3 - reviewer #1 disputed our choice of samples which were based on cell viability and patient characteristics instead of randomized selection. However, a selection based on cell viability is required for any single-cell RNA experiment to ensure high quality data (and thus correct interpretation), as recommended by the 10X Genomics company on their website (e.g. <https://www.10xgenomics.com/blog/faqs-about-single-cell-sample-preparation-covering-the-basics>).

From this, together with other comments on TCR analyses, we can only conclude that reviewer #1 has an inadequate background to judge our single cell RNA/TCR analyses. Furthermore, we argue that, for these subanalyses, the deliberate patient selection we did to control for potential confounders (such as CD4 counts, a known VL relapse risk factor) and to include 'clearly defined' chronic and non-chronic patients, instead of randomized selection, follows our experimental setup and hypotheses, and was necessary to ensure a good comparison and to confirm our previous findings.

28. Line 181-183 - Nothing is said about how the QC and filtering criteria were chosen.

The QC parameters were chosen based on visual inspection due to a lack of standardized guidelines. As scRNA-seq is still a relatively new method, QC and filtering criteria are still a remaining challenge in scRNA-seq and lack standardization. However, we followed the recommended approach of visual inspection and outlier removal (Luecken and Theis, 2019, <https://doi.org/10.15252/msb.20188746>).

29. Line 187-189 - The description of cell cluster annotation seems very inadequate - both singleR and scGate allows the use of different reference sets (singleR) and a choice of predefined or custom gating models (scGate). It should be clarified exactly

what was done - presumably using some pre-computed human data in each case? We are also told nothing about how results from the two approaches were combined to produce a single annotation.

We thank the reviewer for these valid points. To clarify, we have used the Human Primary Cell Atlas reference from the *celldex* package for SingleR, and the predefined gating models of scGate. The ensemble approach consists specifically of first using the predefined gating models of scGate for several well-known populations, and afterwards executing SingleR for validation. SingleR was then used to annotate any remaining clusters for cell populations that were not in the predefined scGate models. However, all annotations were manually curated based on between-cluster differentially expressed genes. This is an approach that is currently recommended by leading experts (Heumos et al., 2023). We have revised the manuscript with this explanation, accordingly.

30. Line 193. Both the log-fold change and the significance cut-off employed seem very surprisingly loose. The log-2 fold change representing only a 20% change in expression. Similarly, the BH procedure doesn't actually control the type I error rate (and so probably shouldn't be called a p-value, but a false discovery rate) but this implies that 5% of the findings are expected to be false positives. Given the very small number of patients involved here, I have concerns about how likely the gene expression changes identified are to be generalisable.

We thank the reviewer for their comment. The chosen log2 fold change is a Seurat default, and it is indeed somewhat inclusive. We chose the default parameter due to this small number of patients. Instead of focusing on identifying specific genes that could be associated with the disease, we wanted to investigate different pathway trends and profiles in an unbiased manner, resulting in a more inclusive approach of DEGs. We like to note that increasing the log2 fold change to 0.58 for an estimated 50% change in expression, resulted in a similar gene expression pattern (see figure below) as the 0.25 threshold in the manuscript.

Response Reviewer:

31. Throughout the section on lines 298-315: nothing is said about the magnitude or statistical significance of differences in cell composition, and the methods section also tell us nothing about the statistical procedures used to support the statements made here.

Response Authors:

For our scRNA-seq data, we have not included statistical testing due to the low number of patients per study group. We have demonstrated the scRNA-seq trends and differences in a visual manner. As such, no hard conclusions are made and the data is further discussed in the discussion.

Response Reviewer:

As for point 27, it is very hard to judge how useful any of the single-cell data is given the small number of patients selected and the lack of any statistical support for many of the patterns described. This seems a critical weakness of the paper - the authors reach conclusions about 'non-chronic' vs 'chronic' VL-HIV patients in general, based on 4 patients that are not randomly-chosen representatives of any defined cohort of patients, but are selected by the investigators 'Based on cell viability and patient characteristics'. So presumably as most likely to show the hoped-for responses? Any findings that lack statistical support should come with a very visible warning that these are essentially anecdotal.

Response Authors:

We argue this is an unjustified response, and one that has already been addressed above (comment #27).

32. Lines 369-370 "The proportion of Th17 did not differ between pVL-HIV and cVL-HIV patients at any timepoint (Fig. 2D-E)." Fig. 2D-E shows CD8+ T cells not Th17.

We thank the reviewer for their observation. This sentence referred to former Fig. S5D-E. However, in the newly revised manuscript we have opted to remove any mentioning of Th17 to improve the focus of the manuscript, and shorten the manuscript.

33. Line 421. Its not at all clear why a 'lack of responsiveness' would result in an actual REDUCTION of these markers with treatment, rather than just the lack of an effect. That seems unexpected and really needs some explanation.

We agree with the reviewer and have removed this sentence and conclusion.

34. Section lines 402-437. As far as we are told, (line 195) the top 12 enriched pathways were considered, and nothing is said about how 'enriched' these pathways need to be in differentially expressed genes before they are reported. We are told

nothing about the numbers of genes and the magnitude of changes in pathways reported here, so it's impossible to evaluate these results: there is a basic lack of detail about the results here, and just showing some heatmaps does not give enough information.

We agree with the reviewer and have added the pathway enrichment scores and number of genes in supplementary figure 11 to provide more information to evaluate the results with.

35. Line 435: This problem is particularly acute when we need to evaluate claims like this: "While this could indicate that treatment has few consistent effects on the expression of genes of these patients, this could also be partly explained by the observed depletion of CD16+ monocytes within this patient group" - as its impossible to work out from the results presented here whether this apparent difference between groups is likely to be a genuine difference, or due to differences in the power to detect expression changes between the groups.

We have deleted the monocytes and NK cell data in order to focus the manuscript on T cells, and have altered the discussion accordingly while taking note of the reviewer's comments.

Response Reviewer:

36. Line 438 - it's not clear what the clonotype expansion analysis is intended to show - or precisely what the claim is - beyond the fact that there is less increase in CD4+ and CD8+ T-cells following treatment in the cVL group than pVL group, which was already observed (e.g. fig 2). From the figures, it looks like all clonotypes expand, but to a lesser extent in chronic than primary patients. That would just seem to be the expectation, given that we know cell numbers recover less in chronic patients. Are the authors claiming that the dynamics of the most common clonotypes are different to others? If so, that should be clarified, and some kind of statistical analysis would seem warranted.

Response Authors:

The clonotype expansion analyses in Figure 8A indicate the expansion/proliferation of the top 30 largest expanded T cell clones (i.e. T cell receptor sequence-based clonotypes, T cells carrying the same TCR) during antileishmanial treatment in non-chronic patients, while this expansion/proliferation is not observed in chronic patients (Fig. 8B). These expanded/proliferated T cell clonotypes are represented on the UMAP in Fig 8C. This is different from Figure 2 where we provide the proportions of all CD8- and CD8+ T cells in the blood compartment. The top expanded T cell clonotypes are most likely to be involved in an active infection due to a shift in dynamics (i.e. expansion or contraction), and we can speculate that these may likely be *Leishmania*-specific T cells. In this figure, we do not see all clonotypes expanding, some contract and disappear, while some expand greatly and become dominant clones. These dominant clones are most likely to be involved in the active infection, and may be important for resolving the infection. For example, in the non-chronic patients, we see several clonotypes expanding to a proportion several folds larger than previously, becoming the dominant clones, while several other clonotypes contract or disappear from the top 30. The total proportion of the top 30 clonotypes of the non-chronic group shifts from around 0.275 to 0.425, nearly doubling in size. In contrast, this total proportion of the top 30 clonotypes of the chronic group shifts from roughly 0.225 to 0.25, showing little to no expansion in response to the active infection. This indicates energy from stimulation, something we can not show in figure 2. In addition, when overlaying these top 30 T cell clonotypes on the UMAP of the VL-HIV patients (Fig 4C), we can deduct that the expanded T cell clonotypes are CD4+ and CD8+ Tem cells in the non-chronic group, but only CD8+ Tem cells in the chronic group. This highlights CD4+ T cell functional energy. Due to the limited number of patients per study group for scRNA-seq, we have not included statistical analysis.

Response Reviewer:

The interpretation of Figure 7 'clonotype expansion' results is still confusing. One concern is the arbitrary classification of 'expansion levels' in figures 7c and d - is the proposed (but not statistically tested) difference in "changes in proportions" robust to the definition of (e.g.) hyper-expanded? Also, the figures presented here are proportions, where the denominator (e.g. number of cells per subtype in different patients at different timepoints) are changing as well as the numerator. It would be very easy to see bigger changes in proportions if the denominator was small, while in

a different subset (with a bigger denominator) the changes in proportion would be much smaller, even for the same absolute size of cell lineage expansion. This analysis would be much more convincing if data on actual cell numbers were shown.

Response Authors:

These 'expansion levels' originated from the scRepertoire package to create this figure (<https://doi.org/10.12688/f1000research.22139.2>). We clarify, however, that these expansion levels are irrelevant to the message of figure 7c and 7d, which is to show that the top 30 clonotypes of the chronic VL-HIV patients are not CD4+ TEM but only CD8+ TCM/TEM. Any shown expansion levels are purely informative, and we do not base any conclusions off of this.

It is also clear the reviewer misunderstood these types of TCR analyses. The proportions presented in figure 7a and figure 7b are the fraction of the sampled repertoire a particular clonotype occupied, where the numerator is the number of clones of a particular clonotype (in a patient), and the denominator is the sum of all T cells of that patient (all clones of all clonotypes). This does not have to do with any subset. The analysis would, in this regard, not be more convincing if data on actual cell numbers were shown, as this does not take into account the repertoire size of a patient, which may differ across patients. Actual cell numbers are instead shown in figure 7c and 7d, as the point of these two figures are not to compare the expansion levels of clonotypes, but to show the phenotype of the top 30 clonotypes.

Reviewer #2 (Remarks to the Author):

Reviewer Response:

The authors have modified their manuscript in response to reviewer's comments. 'People first' language has been added in some places, although there is still a considerable usage of 'patient' (e.g. line 135). In regards to 'orphaned' mentions of analyses or cell types (e.g. NK cells) these have been removed allowing the focus to remain on the T cell populations described across the remainder of the manuscript; this has clarified the messaging of the paper. Figures pinpointed as being over-complicated have been simplified although the longitudinal graphs are still quite busy. The main concern over single marker analysis to signify 'exhaustion' has partly been addressed by co-expression analysis where possible within the limitation of the panel; as TIGIT single expression is still relied upon as a main overall conclusion (with IFN γ / single cell data) the paper could be strengthened by commenting on the complexities of TIGIT expression (e.g. related to lower vs. higher function in different studies). Overall the paper still has a few limitations (e.g. HIV participant controls not being completely matched to the VL-HIV participants) but the authors are honest about these in the discussion and have clarified the messaging compared to the original version.

Response Authors:

Per the reviewer's suggestion, we have replaced the usage of 'patient' with 'participant' or 'individual' in our newly revised manuscript. In addition, per the reviewer's suggestion, we have added a small paragraph in the discussion section of our newly revised manuscript, in which we comment on the lower and higher function of TIGIT (and PD-1) as shown in different studies. We believe we have sufficiently addressed the reviewer's remaining minor concerns.

REVIEWERS' COMMENTS:

Reviewer #4 (Remarks to the Author):

Major criticism of paper

- One of the major drawback of this study is stratification of subject groups (e.g. AL-HIV group having both seropositive individuals with/without VL episode). Chronic VL-HIV group has VL episode in last 3 years. One of the subject has only one VL episode prior to enrollment and one episode after the enrollment (in chronic group). This may be because of the late enrollment for HIV-treatment at study site. Hence, it is the major criticism of the paper.
- The subjects selected for TCRseq in AL-HIV group were without past history of VL (but seropositive). However, Majority of subjects from this group have past history of VL. It is a very biased approach of selection (despite the fact that authors selected samples with highest viability). The inclusion of AL-HIV with past history of VL may have a different picture. Does author consider asymptomatic individual (sero-positive but no overt disease) and past VL cases (seropositive with prior history of disease) immunologically same? Similarly, for chronic HIV-VL group, the 2 subjects selected for single cell immune profiling are quite different from rest of the subjects in that group (in terms of VL episode pre and post enrollment). This is a valid point particularly when the sample size is low.
- Another drawback is that authors showed the functionality based on the expression of TIGIT only. As reported in various previous literatures, the heterogeneity exists among the T cell population in terms of the expression of co-inhibitory markers. Authors did not consider to show the expression of other inhibitory/exhaustion markers on TIGIT+IFN- γ - T cells or TIGIT-IFN- γ + T cells among different group (HIV vs VL-HIV/AL-HIV and chronic vs non-chronic).
- In discussion section authors have stated that "another caveat is that more than half (65%) of our participants with asymptomatic Leishmania and HIV co-infection are long-term cured individuals that reported VL history with a low number (<2) of VL episodes". I do not agree with this statement. These subjects are seropositive as they have the past history of VL (anti-leishmanial antibodies persist for a longer duration, even more than 10-15 years). Hence they can't be defined as asymptomatic.

Despite all those comments, this paper is very well written and data are presented nicely (with several limitations). Authors have accepted the limitation of flow panel as well as inclusion of people with past history of VL in AL-HIV group in their discussion. Authors also have discussed about emphasizing on polyclonal activation (for functionality). They could have used leishmania antigens (soluble) to see if T cells (both CD8+ and CD4+) are functional in terms of IFN- γ production in VL-HIV group compared to other groups (chronic vs non-chronic as well) even HIV condition co-exist (as they suggested PD-1/TIGIT may have predictive value for recurrent relapse.

I do agree with authors that these groups are hard to reach population particularly for such cross-sectional/longitudinal study over a period of time. And despite limitation of the study, it deemed suitable for publication.

However, few minor comments are given below which should be addressed before publication: -

Minor comments/suggestions

- The composition of freezing solution is given as (40% FBS, 60% RPMI, 10% DMSO) on page 4. Please check and correct it.
- Of 24 VL-HIV patients, 17 were with chronic VL but in result section, author mentioned 15 (page 10). Please rectify.
- Although flow data is convincing, I am not sure why authors do not use CD4 marker (they assumed CD3+ CD8- as CD4+ T cells)?

- Page 14; Repetition of "proportion of proportion of". Please rectify.
- Authors mentioned that (in methodology section) anti-CD107a antibody was immediately added to the cells with LAC and GolgiPlug, and after 1 hour GolgiStop was added. As I understand, GolgiPlug (BrefeldinA) is one of constituent of LAC (along with PMA/Ionomycin). What was the reason to add additional Golgiplug.
- Use the term frequency rather than "proportion" for flow data throughout text.
- Is there any change in the pattern of TIGIT/IFN- γ expression in individuals with one single episode prior to study enrollment vs more than 1 VL episode (some has 11 VL episodes) in chronic HIV-VL group (this stratification may help to understand the changes in functionality over time along with exhaustion)?
- CD4 T cells from VL patients (without HIV) also exhibit exhaustion marker (see available literature) but they maintain the produce antigen IFN- γ response. However, this is not the case in HIV-VL as presented in this study. Authors need to discuss this in the manuscript (as only VL as a control is missing in the study)
- Despite increased expression of LAG-3 in VL-HIV group, author did not discuss the significance of this important check-point inhibitor (while suggesting for immune-therapy).

Reviewer comments

Reviewer #4 (Remarks to the Author):

One of the major drawback of this study is stratification of subject groups (e.g. AL-HIV group having both seropositive individuals with/without VL episode). Chronic VL-HIV group has VL episode in last 3 years. One of the subject has only one VL episode prior to enrollment and one episode after the enrollment (in chronic group). This may be because of the late enrollment for HIV-treatment at study site. Hence, it is the major criticism of the paper. The subjects selected for TCRseq in AL-HIV group were without past history of VL (but sero-positive). However, Majority of subjects from this group have past history of VL. It is a very biased approach of selection (despite the fact that authors selected samples with highest viability). The inclusion of AL-HIV with past history of VL may have a different picture. Does author consider asymptomatic individual (sero-positive but no overt disease) and past VL cases (seropositive with prior history of disease) immunologically same? Similarly, for chronic HIV-VL group, the 2 subjects selected for single cell immune profiling are quite different from rest of the subjects in that group (in terms of VL episode pre and post enrollment). This is a valid point particularly when the sample size is low. Another drawback is that authors showed the functionality based on the expression of TIGIT only. As reported in various previous literatures, the heterogeneity exists among the T cell population in terms of the expression of co-inhibitory markers. Authors did not consider to show the expression of other inhibitory/exhaustion markers on TIGIT+IFN- γ - T cells or TIGIT-IFN- γ + T cells among different group (HIV vs VL-HIV/AL-HIV and chronic vs non-chronic). In discussion section authors have stated that “another caveat is that more than half (65%) of our participants with asymptomatic Leishmania and HIV co-infection are long-term cured individuals that reported VL history with a low number (<2) of VL episodes”. I do not agree with this statement. These subjects are seropositive as they have the past history of VL (anti-leishmanial antibodies persist for a longer duration, even more than 10-15 years). Hence they can't be defined as asymptomatic. Despite all those comments, this paper is very well written and data are presented nicely (with several limitations). Authors have accepted the limitation of flow panel as well as inclusion of people with past history of VL in AL-HIV group in their discussion. Authors also have discussed about emphasizing on polyclonal activation (for functionality). They could have used leishmania antigens (soluble) to see if T cells (both CD8+ and CD4+) are functional in terms of IFN- γ production in VL-HIV group compared to other groups (chronic vs non-chronic as well) even HIV condition co-exist (as they suggested PD-1/TIGIT may have predictive value for recurrent relapse. I do agree with authors that these groups are hard to reach population particularly for such cross-sectional/longitudinal study over a period of time. And despite limitation of the study, it deemed suitable for publication.

However, few minor comments are given below which should be addressed before publication:

Firstly, we'd like to thank the reviewer for the careful and critical review, as well as for the appreciation for our manuscript, even despite some of the mentioned limitations. The limitations the reviewer mentions are valid, and have mostly been discussed in the manuscript, but we would like to address the concern that we should not consider *Leishmania* seropositive individuals with a history of disease the same as “asymptomatic participants”. We agree with this concern, and to address this, we have adopted the following strategy: 1) we have renamed the group to “Leishmania-seropositive (LS+-HIV)” which we believe is more correct, while indicating its mixed background clearly in the methods section, and 2) we indicate in all flow cytometry plots which values belong to either seropositive individuals with VL history, or those without VL history. We hope this alleviates the concern and helps readers to do a critical assessment as further stratification into two groups would not be warranted due to sample size.

Minor comments/suggestions:

The composition of freezing solution is given as (40% FBS, 60% RPMI, 10% DMSO) on page 4. Please check and correct it.

We thank the reviewer for pointing out this mistake. It should be 50% FBS, 40% RPMI, and 10% DMSO, and it has been revised accordingly.

Of 24 VL-HIV patients, 17 were with chronic VL but in result section, author mentioned 15 (page 10). Please rectify.

We thank the reviewer for pointing out this small mistake, it should be 17, and has been revised accordingly.

Although flow data is convincing, I am not sure why authors do not use CD4 marker (they assumed CD3+ CD8- as CD4+ T cells)?

The choice not to include a CD4 marker was an unfortunate decision at study start. We chose to focus on CD8+ T cell exhaustion since we expected T cell exhaustion to be a primarily CD8-targeted phenomenon. Therefore, we did not include a CD4 marker in the panels due to colour limitations, and instead opted to include more exhaustion/functionality markers in these CD8-targeted panels. Thus, CD4 T cells were studied by using CD3+CD8- cells as a proxy, despite known limitations. The CD3+CD8- data does follow with the single-cell CD4 T cell data, however.

Page 14; Repetition of “proportion of proportion of”. Please rectify.

We thank the reviewer for pointing this out, and have revised the manuscript accordingly.

Authors mentioned that (in methodology section) anti-CD107 α antibody was immediately added to the cells with LAC and GolgiPlug, and after 1 hour GolgiStop was added. As I understand, GolgiPlug (BrefeldinA) is one of constituent of LAC (along with PMA/Ionomycin). What was the reason to add additional Golgiplug.

We apologise for the confusion here. The reviewer is correct, GolgiPlug is already a constituent of LAC, and we did not add any additional GolgiPlug. To make this more clear, we have now revised the paragraph from the original:

“For the CD8+ T cell functionality-targeted panel, cells were thawed, washed and pre-stimulated with 1 μ l of Leukocyte Activation Cocktail (LAC, BD Biosciences, Belgium) per 300.000 cells, for 4 hours in a humidified incubator at 5% CO₂ and 37°C. In addition to this, anti-CD107 α antibody was immediately added to the cells with LAC and GolgiPlug, and after 1 hour, GolgiStop was added for the remaining 3 hours of incubation.”

to the revised version:

“For the CD8+ T cell functionality-targeted panel, cells were thawed, washed and pre-stimulated with 1 μ l of Leukocyte Activation Cocktail with GolgiPlug (LAC, BD Biosciences, Belgium) per 300.000 cells, for 4 hours in a humidified incubator at 5% CO₂ and 37°C. In addition to this, anti-CD107 α antibody was immediately added to the cells with LAC (with GolgiPlug), and after 1 hour, GolgiStop was added for the remaining 3 hours of incubation.”

We hope the revised version is more clear.

Use the term frequency rather than “proportion” for flow data throughout text.

We thank the reviewer for this suggestion and have revised accordingly.

Is there any change in the pattern of TIGIT/IFN- γ expression in individuals with one single episode prior to study enrollment vs more than 1 VL episode (some has 11 VL episodes) in chronic HIV-VL group (this stratification may help to understand the changes in functionality over time along with exhaustion)?

We do observe that the patients with only one single VL episode prior to study enrolment have a higher frequency of IFN γ +TIGIT- CD8+ T cells compared to the rest of the patients with multiple

episodes, and are rather intermediate between the non-chronics and chronics with multiple episodes (see figure below). As far as a trend goes, the frequency increases during treatment in the non-chronics and chronics with only one prior episode, and subsequently in the chronic patient with only one prior episode but four subsequent episodes (including the D0 episode) it does seem to decrease again towards the relapse episode. However, these inferences are not as clear-cut with this sample size.

CD4 T cells from VL patients (without HIV) also exhibit exhaustion marker (see available literature) but they maintain the produce antigen IFN- γ response. However, this is not the case in HIV-VL as presented in this study. Authors need to discuss this in the manuscript (as only VL as a control is missing in the study)

We agree with the reviewer that we need to mention this, as we do not have a VL control group. In the newly revised version, we have added the lack of a VL group as a major limitation. We have also remarked on the discrepancy between the CD4 T cells of VL-HIV patients exhibiting both higher levels of exhaustion markers and defective antigen-specific IFN- γ responses (both at T cell level and at whole blood level), while VL patients also exhibit exhaustion marker expression but are still able to elicit antigen-specific IFN- γ responses in a whole blood stimulation assay (but perhaps not at the T cell level?). Whether the IFN- γ response is still elicited by T cells of VL patients is, to the best of our knowledge, not fully known, and we believe a other cells such as NK cells could be major contributors of IFN- γ release in a whole blood stimulation assays. We have added this particular nuance in the paragraph below, and we hope the reviewer can agree with this nuance.

Specifically, we have added the following paragraphs:

“One of the major limitations of our study is the lack of a VL group without HIV co-infection. This would have allowed us to better characterise T cell exhaustion or anergy in VL-HIV patients versus VL patients.”

and

“Despite this, concurrent to our findings on TIGIT, Takele et al. observed a correlation between the failure to restore antigen-specific production of IFN- γ (in a whole blood stimulation assay) and the high expression of PD-1 on CD4+ T cells in VL-HIV patients. Thus, in contrast to VL patients who may also present with higher exhaustion markers but still produce an antigen-specific IFN- γ response in a whole blood stimulation assay, VL-HIV patients present with both higher PD-1 levels and a complete lack of antigen-specific IFN- γ response (at T cell and whole blood level).”

Despite increased expression of LAG-3 in VL-HIV group, author did not discuss the significance of this important check-point inhibitor (while suggesting for immune-therapy).

We agree with the reviewer and have added an additional section in the discussion on the importance for multi-marker investigations.

We have added this particular paragraph:

“Next to a high PD-1 and TIGIT expression on pan T cells, the VL-HIV patients also showed increased levels of LAG-3 and TIM-3, although these levels did not distinguish between non-chronic and chronic VL-HIV. The co-expression of multiple immune checkpoint proteins on the same cell, in addition to more detailed assessments of the Leishmania-specific functional and proliferative response, is required to robustly define T cell exhaustion, and define the eligibility of patients for immune checkpoint blockage therapy. Similar to cancer research, blocking only the PD-1 pathway could have suboptimal effects when other immune checkpoint pathways are equally activated but not inhibited. (<https://doi.org/10.1038/s41416-023-02181-6>).”